# TRANSFORMERS LEARN SHORTCUTS TO AUTOMATA

**Bingbin Liu**[1] [*]    **Jordan T. Ash**[2]    **Surbhi Goel**[3][†]    **Akshay Krishnamurthy**[2]    **Cyril Zhang**[2]

[1]Carnegie Mellon University    [2]Microsoft Research NYC    [3]University of Pennsylvania

## ABSTRACT

Algorithmic reasoning requires capabilities which are most naturally understood through recurrent models of computation, like the Turing machine. However, Transformer models, while lacking recurrence, are able to perform such reasoning using far fewer layers than the number of reasoning steps. This raises the question: *what solutions are these shallow and non-recurrent models finding?* We investigate this question in the setting of learning automata, discrete dynamical systems naturally suited to recurrent modeling and expressing algorithmic tasks. Our theoretical results completely characterize *shortcut solutions*, whereby a shallow Transformer with only $o(T)$ layers can exactly replicate the computation of an automaton on an input sequence of length $T$. By representing automata using the algebraic structure of their underlying transformation semigroups, we obtain $O(\log T)$-depth simulators for all automata and $O(1)$-depth simulators for all automata whose associated groups are solvable. Empirically, we perform synthetic experiments by training Transformers to simulate a wide variety of automata, and show that shortcut solutions can be learned via standard training. We further investigate the brittleness of these solutions and propose potential mitigations.

## 1    INTRODUCTION

Modern deep learning pipelines demonstrate an increasing capability to perform *combinatorial reasoning*: pretrained on large, diverse distributions of natural language, math, and code, they are nascently solving tasks which seem to require a rigid "understanding" of syntax, entailment, and state inference. How do these neural networks represent the primitives of logic and the algorithms they execute internally?

When considering this question, there is an immediate mismatch between classical sequential models of computation (e.g., Turing machines) and the Transformer architecture, which has delivered many of the recent breakthroughs in reasoning domains. If we are to think of an algorithm as a set of sequentially-executed computational rules, why would we use a *shallow*[1] *non-recurrent* network?

We study this question through the lens of finite *semiautomata*, which compute state sequences $q_1, \ldots, q_T$ from inputs $\sigma_1, \ldots, \sigma_T$ by application of a transition function $\delta$ (and initial state $q_0$):

$$q_t = \delta(q_{t-1}, \sigma_t).$$

Semiautomata are the underlying structures governing the computations realizable by *automata* (such as regular expression parsers or finite-state transducers), which are simply semiautomata equipped with mappings from states to output. Thus, one natural motivation for studying them comes from the question of whether Transformers can subsume the structures found in classical NLP pipelines. Another motivation comes from the perspective of reinforcement learning and control, where Transformers are beginning to be used as *world models*: semiautomata specify deterministic discrete-state dynamical systems.

We perform a theoretical and empirical investigation of whether (and how) non-recurrent Transformers learn semiautomata. We characterize and analyze how shallow Transformers find *shortcut*

---

[*]The majority of this work was completed while B. Liu was an intern at Microsoft Research NYC.

[†]This work was completed while S. Goel was at Microsoft Research NYC.

[1]Compared to the number of symbols it can process. For example, DistilBERT (Sanh et al., 2019) can handle thousands of tokens with 6 sequential layers.

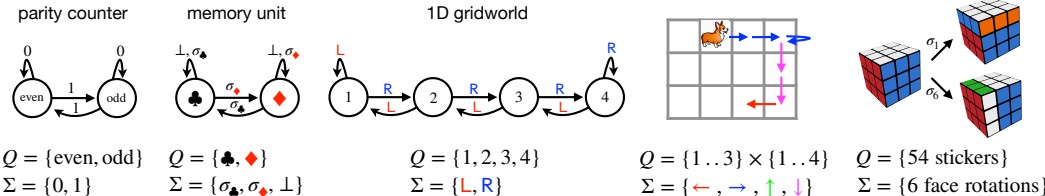

Figure 1: Various examples of semiautomata. From left to right: a mod-2 counter, a 2-state memory unit, $\mathrm{Grid}_4$, a 2-dimensional gridworld constructible via a direct product $\mathrm{Grid}_3 \times \mathrm{Grid}_4$, and a Rubik's Cube, whose transformation semigroup is a very large non-abelian group.

*solutions*, which correctly and efficiently simulate the transition dynamics of semiautomata with far fewer sequential computations than required for iteratively inferring each state $q_t$.

**Our contributions.** Our theoretical results provide structural guarantees for the representability of semiautomata by shallow, non-recurrent Transformers. In particular, we show that:

- *Shortcut* solutions, with depth logarithmic in the sequence length, always exist (Theorem 1).
- *Constant*-depth shortcuts exist for *solvable* semiautomata (Theorem 2). There *do not* exist constant-depth shortcuts for non-solvable semiautomata, unless $\mathsf{TC}^0 = \mathsf{NC}^1$ (Theorem 4).
- For a natural class of semiautomata corresponding to path integration in a "gridworld" with boundaries, we show that there are *even shorter* shortcuts (Theorem 3), beyond those guaranteed by the general structure theorems above.

We accompany these theoretical findings with an extensive set of experiments:

- *End-to-end learnability of shortcuts via SGD* (Section 4). The theory shows that shortcut solutions exist; is the non-convexity of the optimization problem an obstruction to learning them in practice? For a variety of semiautomaton simulation problems, we find empirically that there is no such obstruction. Shallow non-recurrent Transformers are able to learn shortcuts which generalize near-perfectly in-distribution.
- *More challenging settings* (Section 5). We compare non-recurrent and recurrent models in the presence of additional considerations: out-of-distribution generalization (including to unseen sequence lengths) and limited supervision. This reveals the brittleness of non-recurrent models, in line with prior "spurious representation" notions of shortcuts in deep learning. Toward mitigating these drawbacks and obtaining the best of both worlds, we show that with *recency-biased scratchpad training*, Transformers can be guided to learn the robust recurrent solutions.

## 1.1 RELATED WORK

**Emergent reasoning in neural sequence models.** Neural sequence models, both recurrent (Wu et al., 2016; Peters et al., 2018; Howard & Ruder, 2018) and non-recurrent (Vaswani et al., 2017; Devlin et al., 2018), have ushered in an era of broadly-applicable and (with pretraining) sample-efficient *natural language understanding*. Building on this, large-scale non-recurrent Transformer models have demonstrated capabilities in program synthesis, mathematical reasoning, and in-context multi-task adaptation. A nascent frontier is to leverage neural dynamics models, again both recurrent (Hafner et al., 2019) and non-recurrent (Chen et al., 2021a; Janner et al., 2021), for decision making. At the highest level, the present work seeks to idealize and understand the mechanisms behind which deep learning solves tasks requiring combinatorial and algorithmic reasoning.

**Computational models of neural networks.** In light of the above, it is empirically evident that neural networks are successfully learning circuits which generalize on some combinatorial tasks. Many efforts in the theory and empirical science of deep learning are dedicated towards the rigorous analysis of this phenomenon. Various perspectives map self-attention to bounded-complexity circuits (Hahn, 2020; Elhage et al., 2021; Merrill et al., 2021; Edelman et al., 2022), declarative programs (Weiss et al., 2021), and Turing machines (Dehghani et al., 2019). The research program of *BERTology* (Clark et al., 2019; Vig, 2019; Tenney et al., 2019) interprets trained models in terms of known linguistic and symbolic primitives.

The most relevant theoretical work to ours is (Barrington & Thérien, 1988), which acts as a "Rosetta Stone" between classical circuit complexity and semigroup theory. The core technical ideas for

Theorems 1 (NC$^1$ prefix sum), 2 (Krohn-Rhodes), and 4 (Barrington) are inspired by the results and discussions therein. In the language of circuit complexity, our work establishes that shallow, non-recurrent Transformers can efficiently represent all of the constructions involved in the (simple) NC$^1$ and (significantly more complex) ACC$^0$ solutions to sequential multiplication in semigroups. On the other hand, the shorter shortcut from Theorem 3 carefully leverages self-attention to improve upon these results; we were unable to find an analogous refinement in the circuit complexity literature.

**Synthetic combinatorial tasks.** Our problem setting of simulating finite-state semiautomata unifies the settings of several recent investigations of whether (and how) Transformers learn bounded-depth Dyck languages (Yao et al., 2021), parities (Anil et al., 2022), adders (Nogueira et al., 2021; Nanda & Lieberum, 2022), regular languages (Bhattamishra et al., 2020), and sparse logical predicates (Edelman et al., 2022; Barak et al., 2022). Zhang et al. (2022) empirically analyze the behavior and inner workings of Transformers on random-access group operations and note "shortcuts" (which skip over explicit program execution) similar to those we study. We provide an expanded discussion of related work in Appendix A.5.

## 2 PRELIMINARIES

### 2.1 SEMIAUTOMATA AND THEIR ALGEBRAIC STRUCTURE

A *semiautomaton* $\mathcal{A} := (Q, \Sigma, \delta)$ consists of a set of states $Q$, an input alphabet $\Sigma$, and a transition function $\delta : Q \times \Sigma \to Q$. In this work, $Q$ and $\Sigma$ will always be finite sets. For all positive integers $T$ and a starting state $q_0 \in Q$, $\mathcal{A}$ defines a map from input sequences $(\sigma_1, \ldots, \sigma_T) \in \Sigma^T$ to state sequences $(q_1, \ldots, q_T) \in Q^T$: $q_t := \delta(q_{t-1}, \sigma_t)$ for $t = 1, \ldots, T$. This is a deterministic Markov model, in the sense that at time $t$, the future states $q_{t+1}, \ldots, q_T$ only depend on the current state $q_t$ and the future inputs $\sigma_{t+1}, \ldots, \sigma_T$.

We define the task of *simulation*: given a semiautomaton $\mathcal{A}$, starting state $q_0$, and input sequence $(\sigma_1, \ldots, \sigma_T)$, output the state trajectory $\mathcal{A}_{T,q_0}(\sigma_1, \ldots, \sigma_T) := (q_1, \ldots, q_T)$. Let $f : \Sigma^T \to Q^T$ be a function (which in general can depend on $\mathcal{A}, T, q_0$). We will say that $f$ *simulates* $\mathcal{A}_{T,q_0}$ if $f(\sigma_{1:T}) = \mathcal{A}_{T,q_0}(\sigma_{1:T})$ for all input sequences $\sigma_{1:T}$. Finally, for a positive integer $T$, we say that a function class $\mathcal{F}$ of functions from $\Sigma^T \to Q^T$ is said to simulate $\mathcal{A}$ at length $T$ if, for each $q_0 \in Q$, there is a function in $\mathcal{F}$ which simulates $(\mathcal{A}, T, q_0)$.

Every semiautomaton induces a *transformation semigroup* $\mathcal{T}(\mathcal{A})$ of functions $\rho : Q \to Q$ under composition, generated by the per-input-symbol state mappings $\delta(\cdot, \sigma) : Q \to Q$. When $\mathcal{T}(\mathcal{A})$ contains the identity function, it is called a *transformation monoid*. When all of the functions are invertible, $\mathcal{T}(\mathcal{A})$ is a *permutation group*. See Figure 1 for some examples which appear both in our theory and experiments; additional background (including a self-contained tutorial on the relevant concepts in finite group and semigroup theory) is provided in Appendix A.2. An elementary but interesting example is a parity counter (Figure 1, left): the state is a bit, and the inputs are {"toggle the bit", "do nothing"}; the transformation semigroup is $C_2$, the cyclic group of order 2. Parity has been studied in previous synthetic experiments (Zhang et al., 2022; Anil et al., 2022).

### 2.2 RECURRENT AND NON-RECURRENT NEURAL SEQUENCE MODELS

A *sequence-to-sequence neural network* of length $T$ and dimension $d$ is a function $f_{\mathrm{nn}} : \mathbb{R}^{T \times d} \times \Theta \to \mathbb{R}^{T \times d}$, with *trainable parameters* $\theta \in \Theta$. Equipped with an *encoding layer* $E : \Sigma \to \mathbb{R}^d$ and *decoding layer* $W : \mathbb{R}^d \to Q$ (applied position-wise), the function $(W \circ f_{\mathrm{nn}} \circ E) : \Sigma^T \to Q^T$ has the same input and output types as $\mathcal{A}_{T,q_0}$. This work will investigate when the functions defined by neural networks can simulate semiautomata.

A *recurrent neural network* (RNN) is a sequence-to-sequence neural network defined by iterated composition of a *recurrent unit* $g : \mathbb{R}^d \times \mathbb{R}^d \times \Theta \to \mathbb{R}^d$. For a given initial hidden state $h_0 \in \mathbb{R}^d$, and input sequence $u_1, \ldots, u_T \in \mathbb{R}^d$, it produces an output hidden state sequence

$$h_t := g(h_{t-1}; u_t; \theta), \qquad t = 1, \ldots, T.$$

Thus, for any fixed $\theta$, an RNN defines a semiautomaton with infinitely many states and inputs: $Q = \Sigma = \mathbb{R}^d$. Thus, as long as $g$ can represent $\delta$, RNNs can simulate all semiautomata. In this sense, the computational models of RNNs and semiautomata naturally coincide.

An $L$-layer *Transformer* is another sequence-to-sequence network, consisting of alternating self-attention blocks and feedforward MLP blocks

$$f_{\mathrm{tf}} := (\mathrm{id} + f_{\mathrm{mlp}}^{(L)}) \circ (\mathrm{id} + f_{\mathrm{attn}}^{(L)}) \circ (\mathrm{id} + f_{\mathrm{mlp}}^{(L-1)}) \circ ... \circ (\mathrm{id} + f_{\mathrm{attn}}^{(1)}) \circ (\mathrm{id} + P).$$

Briefly, an attention layer performs $\ell_1$-normalized mixing operations across positions $t$, while a constant-layer MLP block performs position-wise function approximation (with no mixing between positions); $\mathrm{id}$ denotes the identity function (residual connections), and $P$ encodes the position $t$.[2] We use fairly standard positional encodings in both theory and experiments. Importantly, the standard Transformer *is* convolutional (in that the weights in $f_{\mathrm{attn}}$ and $f_{\mathrm{mlp}}$ are shared across positions $t$), but is *non-recurrent*: parameters are not shared across blocks.

All architectures have a notion of *computational depth* $D$ (succinctly, depth) when processing inputs of length $T$, which is the longest path in the computational graph. For RNNs, this is $\Theta(T)$ while an $L$-layer Transformer (with constant-layer MLPs) has depth $\Theta(L)$. For Transformers, since they coincide up to constant factors, we use depth and number of layers interchangeably. We will also track the *layers* $L$, *embedding dimension* $d$, *attention width* (the largest number of parallel attention head outputs), and *MLP width* (the largest number of parallel hidden activations in the MLP blocks).[3]

## 3    THEORY: SHORTCUTS ABOUND

A $T$-layer Transformer can trivially simulate a semiautomaton at length $T$ *sequentially*: like an RNN, the $t$-th layer can implement (an embedding of) the state transition $q_{t-1} \mapsto q_t$. Yet, Transformers succeed in practice with long contexts ($\geq 10^3$) and fewer layers (as few as 6). A natural theoretical question is that of *representability*: can Transformers efficiently simulate semiautomata with parallel *shortcut* solutions, whose depths are much smaller than the sequence length $T$?

**Definition 1** (Shortcut solution). Let $\mathcal{A}$ be a semiautomaton. Suppose that for every $T \geq 1$, there is sequence-to-sequence neural network $f_T$ which simulates $\mathcal{A}$ at length $T$. We call this sequence $\{f_T\}_{T \geq 1}$ a *shortcut solution* to the problem of simulating $\mathcal{A}$ if its depth $D$ satisfies $D \leq o(T)$.

By this definition, shortcuts are quite general and some are less interesting than others. For example, it is always possible to construct a constant-depth neural network which memorizes all $|\Sigma|^T$ values of $\mathcal{A}_{T,q_0}$, but these networks must be exceptionally wide. We could also "fast-forward" state simulation, letting each of (say) $\sqrt{T}$ layers simulate $\sqrt{T}$ consecutive state transitions, but, without exploiting the structure of the semiautomaton, this would require width $\Omega(2^{\sqrt{T}})$. To rule out these cases and focus on interesting shortcuts for Transformers, we want the other size parameters (attention and MLP width) to be small: say, scaling polynomially with $T$, or even dependent only on $|Q|, |\Sigma|$. To construct such shortcuts, we need ideas beyond explicit iteration of state transitions.

### 3.1    SEMIAUTOMATA ADMIT SHALLOW PARALLEL SHORTCUTS

We begin by noting that polynomial-width shortcuts *always* exist. This may be counterintuitive if we restrict ourselves to viewing a network's intermediate activations as representations of states $q_t$. When we instead view them as encoding *state transformations* $\delta(\cdot, \sigma) : Q \to Q$ and their compositions, a divide-and-conquer construction is evident (see Figure 2a), detailed in Appendix C.2:

**Theorem 1** (Simulation is parallelizable; informal). *Transformers can simulate all semiautomata* $\mathcal{A} = (Q, \Sigma, \delta)$ *at length* $T$*, with depth* $O(\log T)$*, embedding dimension* $O(|Q|)$*, attention width* $O(|Q|)$*, and MLP width* $O(|Q|^2)$*.*

If we assume that an attention head can only select a constant number of indices, Theorem 1 is unimprovable: the receptive field of a sublogarithmic-depth Transformer is not large enough. However, it is known in theory and practice that soft-attention heads are capable of *attending broadly*, representing certain non-sparse dependencies (Clark et al., 2019; Yao et al., 2021). Thus, we can ask a more challenging question: *can the dense operations of attention enable even shallower shortcuts?*

---

[2]We omit layer normalization. This discrepancy is superficial; see the discussion in Appendix A.4.

[3]Full statements and proofs also track $\infty$-*weight norms* (the largest absolute value of any parameter) and *bit precision* of each floating-point computation. We defer precise definitions and discussion to Appendix A.4.

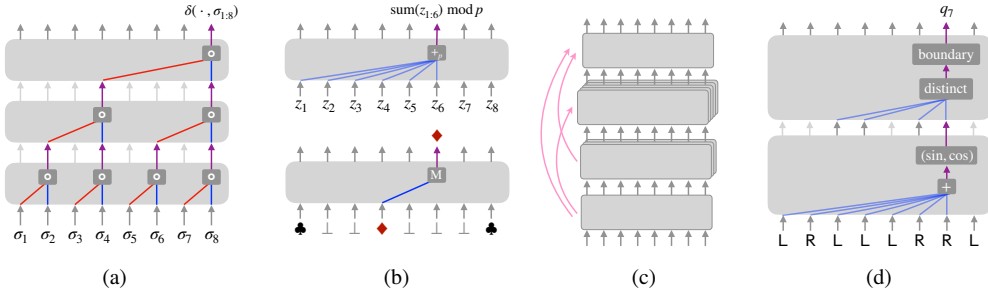

Figure 2: Intuitions for the theoretical constructions. *(a)* Divide-and-conquer function composition yields logarithmic-depth shortcuts (Theorem 1). *(b)* The two "atoms" of the constant-depth Krohn-Rhodes decomposition (Theorem 2) of a solvable semiautomaton: *modular addition* and sequentially resettable *memory*. *(c)* Information flow of the *cascade product*, which is used to glue these atoms together, and easily implemented with residual connections. *(d)* An even shorter shortcut solution for gridworld simulation (Theorem 3; see Appendix C.4).

The key to resolving this question comes from Krohn-Rhodes theory, which gives us tools to reason about the structure of *arbitrary* semiautomata and their transformation semigroups. A landmark result (Krohn & Rhodes, 1965), a vast generalization of the uniqueness of prime factorizations for integers, shows that to simulate any semiautomaton, we only need to handle two types of elementary objects: simple groups, and a *memory unit* (Figure 1b). When the Krohn-Rhodes decomposition contains no non-abelian groups (we call such a semiautomaton *solvable*[4]), there exist constant-depth circuits for simulation, which we manifest as neural networks.

It turns out that positional weight sharing (a.k.a. "width-1 convolutions"), non-recurrence, and self-attention are particularly well-suited for efficiently representing the Krohn-Rhodes decomposition of a semiautomaton: uniform-sum attention heads perform abelian group operations, proximity-based selection heads implement memory units, and the rest of the architecture (MLPs and residual connections) implements the *cascade product* (Definition 4) which combines these atomic operations. Overall, we conclude:

**Theorem 2** (Transformer Krohn-Rhodes; informal). *Transformers can simulate all solvable semiautomata* $\mathcal{A} = (Q, \Sigma, \delta)$*, with depth* $O(|Q|^2 \log |Q|)$*, embedding dimension* $2^{O(|Q| \log |Q|)}$*, attention width* $2^{O(|Q| \log |Q|)}$*, and MLP width* $|Q|^{O(2^{|Q|})} + 2^{O(|Q| \log |Q|)} \cdot T$*.*[5]

It is quite counterintuitive[6] that as $T \to \infty$, no additional depth is needed for such a large class of problems. We provide background and details (including the definition and implementation of this notion of semigroup product) in Appendices A.2 and C.3. In Figure 2b and 2c, we illustrate the three key ingredients: efficient implementations of the two atoms (modular counting and memory lookups), and the procedure for gluing them together (building a *transformation cascade*).

**What does each layer do?** The construction in Theorem 1 recursively composes functions, as opposed to the naive solution of directly emulating states. Theorem 2 takes a very different approach: it relies on the *holonomy decomposition* variant of the Krohn-Rhodes theorem (Eilenberg, 1974). Rather than simulating $q_t$ or composing functions, the computational paths correspond to a $|Q|$-level tree of nested coarsenings of the semiautomaton's dynamics: *"which subset of states could $q_t$ be in right now?"* Within each level of this tree, the network must implement (generally non-commutative) group operations. This can be done with $O(|Q| \log |Q|)$ layers, by leveraging the *Jordan-Hölder decompositions* and the *universal embedding theorem* (Krasner & Kaloujnine, 1951).

**Can we get even shallower shortcuts?** Finally, we show that on a natural class of problems, the computational model of self-attention leads to further fine-grained improvements over the guarantees of Krohn-Rhodes theory. Motivated by the application of Transformers in modeling environment

---

[4]See Definition 6. Among the solvable groups are the dihedral groups $D_{2n}$, the permutation groups $S_n, A_n$ for $n \le 4$, the quaternion group $Q_8$, all groups of order $< 120$ except $A_5$, and all groups of odd order.

[5]Perhaps surprisingly, the only place where a width of $T$ is used is to implement a mod-$n$ gate. This dependence can be removed entirely if we allow for periodic activation functions such as $x \mapsto \sin(x)$.

[6]From the back cover of Rhodes et al. (2010): the underlying theorem launched a theory which *"reveals deep and unexpected connections between algebra (semigroups) and areas of science and engineering"*.

dynamics, we consider the semiautomaton $\mathrm{Grid}_n$ corresponding to a "gridworld": $n$ states on a line, with input symbols "move left if possible" and "move right if possible" (see Figure 1, middle). We show that self-attention enables an extremely concise solution, with depth independent of both $T$ and $|Q| = n$:

**Theorem 3** (Depth-2 shortcut for gridworld; informal). *For all positive integers $n, T$, Transformers can simulate $\mathrm{Grid}_n$ at length $T$, with depth 2,[7] embedding dimension $O(1)$, attention width $O(n)$, and MLP width $O(T)$.[8]*

The proof builds a concise parallel *nearest boundary detector*, and can be found in Appendix C.4. We note that this particular setting is known to be an extremal case for the holonomy construction in Krohn-Rhodes theory (Maler (2010) discusses this, calling it the *elevator automaton*). It would be interesting to generalize our improvement and characterize the class of problems for which self-attention affords $O(1)$ instead of poly($|Q|$)-depth solutions.

**Aren't neural networks universal function approximators?** Sufficiently wide neural networks with sufficiently expressive nonlinearities can fit arbitrary functions (Hornik et al., 1989; Cybenko, 1989). However, if we constrain complexity measures such as depth and width, one cannot hope to apply universality directly. It is true that one can take the discrete circuit constructions in (Barrington & Thérien, 1988), "compile" every gate to a constant-depth network, and recover shortcut solutions with $o(T)$ depth and poly($T$) width. However, our constructions go far beyond black-box reductions– the roles of self-attention and positional parameter sharing allow for such efficient constructions that no parameter count depends on $T$ (except the MLP width, which is removable with a periodic activation function). Furthermore, the constructions are so simple and natural that they are corroborated by the preliminary "reverse engineering" investigation in Section 4.

### 3.2 LOWER BOUNDS

Can Theorem 2 be improved to handle non-solvable semiautomata? (Equivalently: can Theorem 1 be improved to constant depth?) It turns out that as a consequence of a classic result in circuit complexity (Barrington, 1986), this question is equivalent to the major open question of $\mathsf{TC}^0 \overset{?}{=} \mathsf{NC}^1$ (thus: conjecturally, no). Unless these complexity classes collapse, Theorems 1 and 2 are optimal. In summary, simulating non-solvable semiautomata with constant depth is provably hard:

**Theorem 4** (Transformer Barrington). *Let $\mathcal{A}$ be a non-solvable semiautomaton. Then, for sufficiently large $T$, no $O(\log T)$-precision Transformer with depth independent of $T$ and width polynomial in $T$ can continuously simulate $\mathcal{A}$ at length $T$, unless $\mathsf{TC}^0 = \mathsf{NC}^1$.*

This is proven in Appendix C.5. The smallest example of a non-solvable semiautomaton is the one on $|Q| = 5$ states, whose transitions generate $A_5$ (all of the even permutations).

Finally, we note that although our width bounds might be improvable, an exponential-in-$|Q|$ number of hypotheses (and hence a network with poly($|Q|$) parameters) is unavoidable if one wishes to learn an arbitrary $|Q|$-state semiautomaton from data: there are $|Q|^{|Q| \cdot |\Sigma|}$ of them, which generate $|Q|^{\Omega(|Q|^2)}$ distinct semigroups (Kleitman et al., 1976). If we wish to study how machine learning models can efficiently identify large algebraic structures, we will need finer-grained inductive biases to specify which semiautomata to prefer, a direction for future work.

## 4 EXPERIMENTS: CAN SGD FIND THE SHORTCUTS?

Our theorems are limited to *representability*: concise shallow solutions exist, but whether gradient-based local search (i.e., standard training) finds them is another matter entirely. For example, embedded within the problem of learning to simulate the 2-state parity semiautomaton is a well-known non-convex optimization problem (Daniely & Malach, 2020; Edelman et al., 2022; Nichani et al., 2022). In general, even detecting whether $\mathcal{T}(\mathcal{A})$ contains a cycle is PSPACE-hard (Cho & Huynh, 1991). Theoretically understanding how the training dynamics of deep learning transcend the worst-case hardness of non-convex optimization is a major frontier of research, that we do not attempt to

---

[7]This requires max-pooling. If we do not use max-pooling, we can instead use an MLP with width $2^{O(n)}$ and depth $O(1)$, or width $O(n)$ and depth $O(\log n)$.

[8]As with Theorem 2, the width can be reduced to $O(n)$ if we employ periodic activation functions.

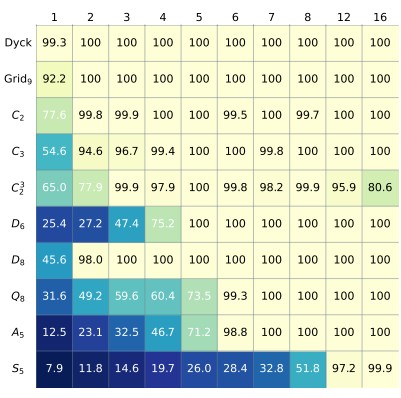

| | 1 | 2 | 3 | 4 | 5 | 6 | 7 | 8 | 12 | 16 |
|---|---|---|---|---|---|---|---|---|---|---|
| Dyck | 99.3 | 100 | 100 | 100 | 100 | 100 | 100 | 100 | 100 | 100 |
| Grid$_9$ | 92.2 | 100 | 100 | 100 | 100 | 100 | 100 | 100 | 100 | 100 |
| $C_2$ | 77.6 | 99.8 | 99.9 | 100 | 100 | 99.5 | 100 | 99.7 | 100 | 100 |
| $C_3$ | 54.6 | 94.6 | 96.7 | 99.4 | 100 | 100 | 99.8 | 100 | 100 | 100 |
| $C_2^3$ | 65.0 | 77.9 | 99.9 | 97.9 | 100 | 99.8 | 98.2 | 99.9 | 95.9 | 80.6 |
| $D_6$ | 25.4 | 27.2 | 47.4 | 75.2 | 100 | 100 | 100 | 100 | 100 | 100 |
| $D_8$ | 45.6 | 98.0 | 100 | 100 | 100 | 100 | 100 | 100 | 100 | 100 |
| $Q_8$ | 31.6 | 49.2 | 59.6 | 60.4 | 73.5 | 99.3 | 100 | 100 | 100 | 100 |
| $A_5$ | 12.5 | 23.1 | 32.5 | 46.7 | 71.2 | 98.8 | 100 | 100 | 100 | 100 |
| $S_5$ | 7.9 | 11.8 | 14.6 | 19.7 | 26.0 | 28.4 | 32.8 | 51.8 | 97.2 | 99.9 |

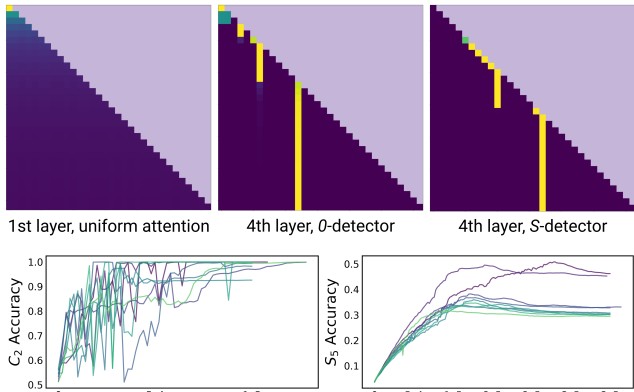

(a) Accuracy across tasks (rows) and network depths (columns).

(b) Attention heatmaps ($\text{Grid}_8$); unstable training ($C_2$ and $S_5$).

Figure 3: Overview of the empirical results in Section 4, on in-distribution learnability of shortcuts by standard Transformer training. *(a)* Truncated table of results (in-distribution accuracy); rows specify semiautomaton simulation problems, and columns specify network depth. *(b)* Attention heads implement a *nearest boundary detector (top)*; training is highly unstable *(bottom)*.

address here. Instead, we approach the question of optimization through an empirical lens. Our primary goal is to understand if gradient-based training can find shortcut solutions at all, rather than whether such training is stable. Accordingly, unless otherwise noted, we report the performance of the *best* model among 20 replicates; the *median* performance is provided in Appendix B.

For a selection of 19 semiautomata corresponding to various groups and semigroups, we train shallow Transformers to output their state sequences given random inputs. Specifically, we apply GPT-2-like models (Radford et al., 2019) with 1-16 layers on freshly-sampled sequences of length $T = 100$.[9] Strikingly, we obtain positive results ($> 99\%$ in-distribution accuracy) for all of them, including ones which generate the non-solvable groups $A_5$ and $S_5$.[10] Figure 3a gives a selection of our full results (in Appendix B.1). We find that more complex semiautomata (corresponding to non-abelian groups) require deeper networks to learn, in agreement with our theoretical constructions.

**Which shallow solutions are learned?** Our theoretical results identify shortcut solutions which follow multiple, mutually incompatible paradigms. In general, we do not attempt a full investigation of *mechanistic interpretability* of the trained models. As preliminary evidence, we visualize some of the attention patterns in Figure 3b *(top)* within successfully-trained models, finding attention heads which perform *flat summations* (with uniform attention) and *conditional resets*.

**Optimization quirks.** Although sufficiently deep networks find the solutions with non-negligible probability, the training dynamics are unstable; Figure 3b *(bottom)* shows some training curves, exhibiting high variance, negative progress, or accuracy that decays with continued training. In the same vein as the "synthetic reasoning tasks" introduced by Zhang et al. (2022), we hope that semiautomaton simulation will be useful as a clean, nontrivial testbed (with multiple difficulty knobs) for debugging and improving training algorithms, and perhaps the neural architectures themselves.

## 5 FURTHER EXPERIMENTS: MORE CHALLENGING SETTINGS

For a wide family of algebraic structures, we have proven that the function class of shallow non-recurrent networks *subsumes* deeper finite-state recurrent models. Furthermore, the experiments in Section 4 have shown that despite the non-convexity of the optimization problem, standard training works: Transformers can learn shortcuts to semiautomaton simulation, end-to-end. While encouraging, the experiments in Section 4 are idealized in several ways, and it is natural to ask if Transformers perform similarly in more challenging semiautomaton simulation scenarios. Towards answering this

---

[9]Using freshly-sampled data ensures that the model cannot achieve good performance by brute-force memorization in a number of training steps we could ever execute computationally (for sufficiently large $T$ such as $T = 100$), since there are an exponential number of sequences.

[10]Explanations on why certain groups are harder to learn are provided in Appendix B.1.1.

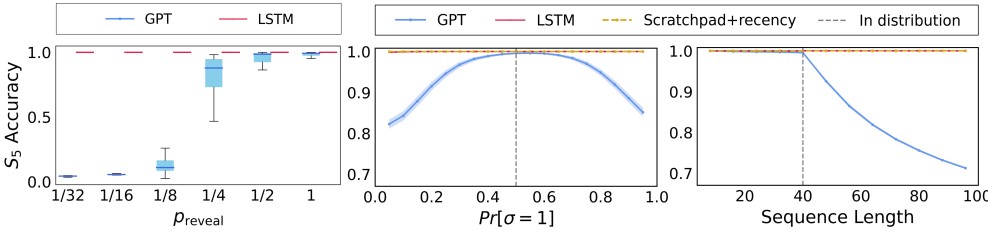

| Task | $\text{Dyck}_{4,8}$ | $\text{Grid}_9$ | $S_5$ | $C_4$ | $D_8$ | $(\texttt{abab})^\star$ |
|---|---|---|---|---|---|---|
| **Observation** | stack top | $\mathbb{1}_{\text{boundary}}$ | $\pi_{1:t}(1)$ | $\mathbb{1}_{0 \bmod 4}$ | location | accept |
| **Accuracy** | 100.0 | 99.7 | 98.0 | 99.7 | 99.8 | 100.0 |

(a) Accuracies with indirect supervision. LSTM gets 100% on all tasks.

(b) Varying $p_{\text{reveal}}$ (log spacing).    (c) $C_2$ (parity): accuracy at different $Pr[\sigma = 1]$ and $T$.

Figure 4: Overview of the empirical results in Section 5. *(a)* Learning in the latent-state setting, with various observation maps $\varphi(q_t)$. *(b)* Learning from incomplete state sequences: final accuracy vs. position-wise probability of a hidden token. *(c)* OOD generalization: Transformers fail to generalize to different distributions and lengths.

question, in this section, we consider some challenges that may arise in practice and an associated set of experimental results; further details are deferred to Appendix B.2.

### 5.1 INCOMPLETE AND INDIRECT SUPERVISION

**Automata are partially observable semiautomata.** Consider the case of *partial observability*. For any semiautomaton $\mathcal{A} = (Q, \Sigma, \delta)$ and a (generally non-invertible) *observation* function $\varphi : Q \to \tilde{Q}$, we can define the problem of predicting $\tilde{q}_t := \varphi(q_t)$. If we can only obtain observations $\tilde{q}_t$ (i.e., the state is latent), this fully captures the problem of learning a finite-state *automaton* from data. The results in this paper have shown that this is equivalent to the fully-observable case in terms of *representation*. However, the *learning* problem can be much harder; indeed, this may account for Bhattamishra et al. (2020)'s negative results on learning regular languages with constant-depth Transformers. Note that this also captures autoregressive next-token prediction tasks induced by distributions (e.g., generating Dyck languages (Yao et al., 2021)) where the sequence's continuations depend on a latent semiautomaton's state (e.g., the current stack for Dyck). Despite these potential challenges, we find that Transformers are able to find a solution with good in-distribution performance for all partially observable settings we consider; see Figure 4(a).

**Learning from incomplete state sequences.** Next, we consider the setting which is identical to that described in Section 4, but each state $q_t$ is randomly revealed from the training data with some probability $0 \leq p_{\text{reveal}} \leq 1$. As with partial observability, this does not affect representation issues, but can make learning/optimization much harder. Figure 4b shows the accuracy of $S_5$ for models trained on length 100 sequences for various $p_{\text{reveal}}$. It can be seen that Transformers may be unable to find good solutions when the labels become sparser, whereas LSTM's performance stays robust across all choices of $p_{\text{reveal}}$.

### 5.2 OUT-OF-DISTRIBUTION SHORTCOMINGS OF SHORTCUT SOLUTIONS

The theoretical construction of modular counters (Lemma 6) suggests a possible failure mode: if attention performs prefix addition and the MLP computes the sum modulo $n$, the MLP could fail on sums unseen during training. This suggests that if the distribution over $\sigma_{1:T}$ shifts between training and testing (but the semiautomaton remains the same), a non-recurrent shortcut solution might map inputs into an intermediate latent variable space (like the sum) which fails to generalize.

Indeed, we observe that with the same models which obtain the positive in-distribution results in Section 4, accuracy degrades as distribution shift increases; see Figure 4(c) *(left)*, where the performance drops as the probability of seeing input $\sigma = 1$ deviates from the training distribution $(Pr[\sigma = 1] = 0.5)$. From the viewpoint of mechanistic interpretation, this is further (but not absolutely conclusive) evidence that with standard training, Transformers learn implementations similar to those predicted by the theory. We provide details and further empirical evidence in Section B.2.3.

More ambitiously, we could try to use these models to extrapolate to longer sequence lengths $T$ than those seen in the training data. Promoting this difficult desideratum of *length generalization* is an intricate problem in its own right; see Yao et al. (2021); Anil et al. (2022) for more experiments similar to ours. Figure 4(c) *(right)* shows the performance on sequences of various lengths, where Transformer's accuracy drops sharply as we move to lengths unseen during training. In contrast, LSTM performs perfectly in both out-of-distribution scenarios. Details are deferred to Section B.2.4.

**Shortcuts as "unintended" solutions.** Throughout the deep learning literature, the term *shortcut* is often used in a statistical sense to connote undesired (i.e., misleading, spurious, or overfitting) learned representations (Geirhos et al., 2020; Robinson et al., 2021). The experiments in this section show why our circuit-depth shortcuts are statistical shortcuts. Specifically, we have identified a problem with learning relaxations to sequential state simulation: the models may "hallucinate" statistically suboptimal latent variables. The positive results in Sections 3 and 4 suggest that this may only be robustly diagnosable via out-of-distribution evaluation.

Finally, we empirically show that this flaw is circumventable. Using a combination of *scratchpad* (a.k.a. "chain-of-thought") (Nye et al., 2021; Wei et al., 2022) and recency bias (Press et al., 2022), we demonstrate that Transformers can be guided towards learning recurrent (depth-$T$) solutions that generalize out-of-distribution and to longer sequence lengths (Figure 4(c), yellow curves).

**Computational-statistical tradeoffs.** The experiments in this section highlight a statistical price for learning shortcuts to semiautomaton simulation. On the other hand, the shallowness of these shortcuts is computationally appealing: leveraging parallel computation, they enjoy much lower latency ($O(\log T)$ or $O(1)$, compared to $O(T)$), in both training and inference. Whether the best of both worlds is attainable is an interesting avenue for future work.

## 6 CONCLUSIONS AND FUTURE WORK

We have conducted a theoretical and empirical analysis of how shallow Transformers can learn *shortcut solutions* to the problem of simulating the transitions of semiautomata (and thus, the algebraic structures which underlie regular expressions, finite-state transducers, and deterministic MDPs). Using tools from semigroup theory and circuit complexity, we have constructed explicit logarithmic-depth and constant-depth shortcuts for semiautomaton simulation. Experimentally, we have shown that gradient-based optimization finds shortcut solutions which generalize near-perfectly in-distribution (Section 4), but are brittle out-of-distribution (Section 5). We hope that these results shed new light on the power and limitations of applying shallow non-recurrent models, even when the dynamics we wish to represent are deep and recurrent.

**Beyond Transformers?** The theory and experiments in this work are specialized to the Transformer architecture, to provide a concrete and timely setting. However, we note that the underlying themes (continuous arithmetic circuits; parameter sharing across input positions and/or iterated function compositions; local vs. global computational units) are not specialized to any particular neural architecture[11], nor the field of deep learning at all. The question of *"which architectures are even more natural for learning discrete automata, while being optimizable by gradient-based search?"* is extremely open-ended. We believe that the themes of sufficient depth and recurrent vs. non-recurrent function composition are relevant to the study of other (and future) deep learning methods.

**Future topics.** In terms of theory, we have only scratched the surface of the possible interplay between neural architectures and classical ideas from the complexity theories of circuits and automata. One salient direction is to generalize the shorter shortcut constructions in Theorem 3. Also, we have made no attempt to treat *stochastic* environments, which would fully capture probabilistic Markov models and MDPs. Section 5 alludes to a landscape of algorithm design challenges in the presence of distribution shift and limited supervision. The latter (i.e., latent state inference) is known to lead to worst-case computational hardness (Papadimitriou & Tsitsiklis, 1987), but yields powerful empirical tools when tractable. Towards fully understanding and leveraging the circumstances which allow learning algorithms to decode and simulate $q_t$, there is much work to be done.

---

[11]In fact, the divide-and-conquer construction of Theorem 1 is almost recurrent with $\log(T)$ depth, and resembles WaveNet-like hierarchical pooling (van den Oord et al., 2016; Larsson et al., 2016), more than Transformers.

ACKNOWLEDGEMENTS

We are very grateful to Abhishek Shetty for helpful discussions about circuit complexity. We also thank Ashwini Pokle for thoughtful comments and suggestions towards improving clarity and readability.

REPRODUCIBILITY STATEMENT

Complete proofs of the theoretical results are provided in Appendix C, with a self-contained tutorial of relevant group-theoretic concepts in Appendix A.2. For the empirical results, all our datasets are derived from synthetic distributions, which are clearly described in Appendix B.1 and B.2. The architectures, implementations (with references to popular base repositories), and hyperparameters (including training procedure) are documented in Appendix B.3. We intend to release our code as open source prior to publication.

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

# Appendix

## Table of Contents

## A ADDITIONAL BACKGROUND AND NOTATION

### A.1 NOTATION

Below, we list our notational conventions for indices, vectors, matrices, and functions.

- For a natural number $n$, $[n]$ denotes the *index set* $\{1, 2, \ldots, n\}$.
- For a vector $v \in \mathbb{R}^n$ and $i \in [n]$, $v_i$ denotes the $i$-th entry. When $v$ is an expression, we use $[v]_i$ for clarity. Vectors can be instantiated by square brackets (like $[1\ 2\ 3] \in \mathbb{R}^3$). They can be indexed by slices: $v_{a:b}$ denotes $[v_a\ v_{a+1}\ \ldots\ v_b]$. We adhere to the convention that all vectors are column vectors.
- For a matrix $M \in \mathbb{R}^{m \times n}, i \in [m], j \in [n]$: $M_{ij}$ (or $[M]_{ij}$) denotes the $(i, j)$-th entry. $M_{i,:}$ and $M_{:,j}$ denote the $i$-th row and $j$-th column, respectively. Importantly, we note the convention that this "slice" notation converts all vectors into column vectors.
- When the first dimension of a matrix $M \in \mathbb{R}^{T \times d}$ is to be interpreted as a sequence length, we will implicitly convert a sequence of vectors $v_1, \ldots, v_T \in \mathbb{R}^d$ into a matrix $(v_1, \ldots, v_T) \in \mathbb{R}^{T \times d}$ whose *rows* the vectors $v_t$. This is an arbitrary choice (compared to concatenating columns and obtaining a matrix in $\mathbb{R}^{d \times T}$), selected to adhere to previously standardized notation for the Transformer.
- We will sometimes index vectors and matrices with named indices (such as $\perp$ for padding tokens) instead of integers, for clarity.
- $e_i$ denotes the $i$-th elementary (one-hot) unit vector. Likewise as above, we sometimes use non-integer indices (e.g. $e_\perp$).
- For vectors $u, v \in \mathbb{R}^d$, $\langle u, v \rangle = u^\top v$ both denote the inner product.
- For a function $f : X \times Y \to Z$ and all $y \in Y$, we will let $f(\cdot, y) : X \to Z$ denote the restriction of $f$ to $y$ (and similarly for other restrictions). This appears in the per-input state transition functions $\delta(\cdot, \sigma) : Q \to Q$, as well as the functions represented by neural networks for a particular choice of weights.
- For functions $f, g$, $f \circ g$ denotes composition: $(f \circ g)(x) := f(g(x))$. When we compose neural networks $f : X \times \Theta_f \to Y, g : Y \times \Theta_g \to Z$ with parameter spaces $\Theta_f, \Theta_g$, we will use $f \circ g : X \times (\Theta_f \times \Theta_g) \to Z$ to indicate the composition $f(g(x; \theta_g)\theta_f)$.
- $(\cdot)_+ : \mathbb{R} \to \mathbb{R}$ denotes the ReLU (a.k.a. positive part) function: $(x)_+ = \max\{x, 0\}$. In function compositions, we use $\sigma$ to denote the entry-wise ReLU (e.g. $f \circ \sigma \circ g$).

### A.2 AUTOMATA, SEMIGROUPS, AND GROUPS

Recall that a semiautomaton $\mathcal{A} = (Q, \Sigma, \delta)$ has a state space $Q$, an input alphabet $\Sigma$, and a transition function $\delta : Q \times \Sigma \to Q$. For any natural number $T$ and a starting state $q_0 \in Q$, by repeated composition of the transition function $\delta$, one can use $\mathcal{A}$ to define a map from a sequence of inputs $(\sigma_1, \ldots, \sigma_T) \in \Sigma^T$ to a sequence of states $(q_1, \ldots, q_T) \in Q^T$ via:

$$q_t := \delta(q_{t-1}, \sigma_t), \forall t \in [T].$$

Here and below, it is helpful to use a matrix-vector notation to express the computation of semi-automata. For a given semiautomaton we can always identify the state space $Q$ with index set $\{1, \ldots, |Q|\}$ and use a one-hot encoding of states into $\{0, 1\}^{|Q|}$. For each input symbol $\sigma \in \Sigma$, we associate a transition matrix $\delta(\cdot, \sigma) \in \{0, 1\}^{|Q| \times |Q|}$ with entries $[\delta(\cdot, \sigma)]_{q', q} = \mathbf{1}\{\delta(q, \sigma) = q'\}$. This implies that for all $q, \sigma$, we have $e_{\delta(q, \sigma)} = \delta(\cdot, \sigma)e_q$, so that the computation of the semiautomaton amounts to repeated matrix-vector multiplication.

While semiautomata are remarkably expressive, we discuss a few simple examples throughout this background section to elucidate the key concepts.

**Example 1** (Parity). *Let $Q = \Sigma = \{0, 1\}$ and let $\delta(q, 0) = q$ and $\delta(q, 1) = 1 - q$. Then, starting with $q_0 = 0$, the state at time $t$, $q_t$, is 1 if the binary sequence $(\sigma_1, \ldots, \sigma_t)$ has an odd number of 1s.*

**Example 2** (Flip-flop). *Let $Q = \{1, 2\}, \Sigma = \{\perp, 1, 2\}$ and let $\delta$ be given by*

$$\delta(\cdot, \perp) = I_{2 \times 2}, \quad \delta(\cdot, 1) = \begin{pmatrix} 1 & 1 \\ 0 & 0 \end{pmatrix}, \quad \delta(\cdot, 2) = \begin{pmatrix} 0 & 0 \\ 1 & 1 \end{pmatrix}$$

*As the name suggests, this semiautomaton implements a simple memory operation where the state at time $t$ is the value of the most recent non-$\perp$ input symbol.*

**Example 3** (1D gridworld). *Let $S$ be a natural number, $Q = \{0, 1, \ldots, S\}$ and $\Sigma = \{L, \perp, R\}$. Then the transition matrices are given by:*

$$\delta(\cdot, \perp) = I_{S+1 \times S+1}, \quad \delta(\cdot, L) = \begin{pmatrix} 1 & \ddots & & \\ 0 & & I_{S \times S} & \\ \vdots & & & \ddots \\ 0 & \ldots & \ldots & 0 \end{pmatrix} \quad \delta(\cdot, R) = \begin{pmatrix} 0 & \ldots & \ldots & 0 \\ \ddots & & & \vdots \\ & I_{S \times S} & & 0 \\ & & \ddots & 1 \end{pmatrix}.$$

*This semiautomaton describes the movement of an agent along a line segment where actions $-1$ and $+1$ correspond to decrementing and incrementing the state respectively, except that the decrement input has no effect at state $0$ and the increment input has no effect at state $S$.*

Note that we have chosen a convention which differs slightly from the main paper (i.e. Figure 1): we enumerate the indices starting from $0$ rather than $1$. This is because the proofs are stated more naturally when the boundaries of the gridworld are identified with the indices $0$ and $S$.

For a semiautomaton $\mathcal{A} = (Q, \Sigma, \delta)$ each input symbol $\sigma \in \Sigma$ defines a function $\delta(\cdot, \sigma) : Q \to Q$. These functions can be composed in the standard way, and we use $\delta(\cdot, \sigma_{1:t})$ to denote the $t$-fold function composition. Note that $\delta(q_0, \sigma_{1:t})$ is precisely the value of the state at time $t$ on input $\sigma_{1:t}$. Thus, the set of all functions that can be obtained by composition of the transition operator, formally

$$\mathcal{T}(\mathcal{A}) := \{\delta(\cdot, \sigma_{1:t}) : t \in \mathbb{N}, \sigma_{1:t} \in \Sigma^t\},$$

plays a central role in describing the computation of the semiautomaton. This object is a *transformation semigroup*. We now turn to describing the necessary algebraic background.

Recall that a *group* $(\mathcal{G}, \cdot)$ is a set $\mathcal{G}$ equipped with a binary operation $\cdot : \mathcal{G} \times \mathcal{G} \to \mathcal{G}$ such that

- *(identity)* There exists an identity element $e \in \mathcal{G}$ such that $e \cdot g = g \cdot e = g$ for all $g \in \mathcal{G}$.

- *(invertibility)* Every element $g \in \mathcal{G}$ has an inverse $g^{-1} \in \mathcal{G}$ such that $g \cdot g^{-1} = g^{-1} \cdot g = e$

- *(associativity)* The binary operation is associative: $(g_1 \cdot g_2) \cdot g_3 = g_1 \cdot (g_2 \cdot g_3)$.

A *monoid* is less structured than a group; there must be an identity element and the binary operation must be associative, but invertibility is relaxed. A *semigroup* is even less structured: the only requirement is that the binary operation is associative.

It is common to let $\mathcal{G}$ be a subset of functions from $Q \to Q$ where $Q$ is some ground set and let the binary operation be function composition. In this case, the structure is called a *permutation group* or *transformation monoid/semigroup* depending on which subset of the above properties hold. For transformation groups, since every element has an inverse under function composition, it is immediate that every element is some permutation over the ground set.

In fact, taking $\mathcal{G}$ to be a subset of functions as above is without loss of generality: by Cayley's theorem every group is isomorphic (equivalent after renaming elements) to a transformation group on some ground set, and we can take the ground set to have the same number of elements as the original group (for finite groups). Analogously, all semigroups are isomorphic to a transformation semigroup, but the ground set may need one additional element (for the identity); this is Cayley's theorem for semigroups. It is also clear that every transformation semigroup can be realized by some semiautomaton by trivially having the input symbols correspond to the functions in $\mathcal{G}$.[12] Therefore we have lost no structure when passing from finite semiautomata to finite semigroups.

Before discussing the compositional structure of semigroups, we give one more canonical example.

---

[12]More succinctly, inputs can correspond to a generating set of the group, but this is not relevant for our results.

**Example 4** (Cyclic group). *Let $S$ be a natural number let $Q = \{0, 1, \ldots, S-1\}$ and let $\Sigma = \{1\}$ have only one element. The dynamics are given by $\delta(q, 1) = (q + 1) \mod S$. Clearly this semiautomaton implements counting modulo $S$. The underlying group is the* cyclic group, *denoted $C_S$, which is isomorphic to the integers mod $S$ with addition as the binary operation. Note that in this case, the operation is commutative, which makes the group* abelian.

Let us now turn to the compositional structure of groups and semigroups. Since it is without loss of generality to consider transformation (semi)groups, we always take the binary operation to be function composition. A *subgroup $H$ of a group $G$* is a subset of the elements of $G$ that is also a group, denoted as $H \leq G$. In particular it must be closed under the binary operation. $N$ is a *normal subgroup*, denoted $N \triangleleft G$, if in addition to being a subgroup, it satisfies that $\{gn : n \in N\} = \{ng : n \in N\}$. (These sets are known as the left and right cosets of $N$ in $G$ and denoted $gN$ and $Ng$ respectively.) [13] Normal subgroups can also arise as the kernel of a mapping from $G$ to a subgroup $H$ of $G$. Let $\phi : G \to H$ be a mapping that preserves the group operation (i.e., a group homomorphism) and let $\ker(\phi) := \{g : \phi(g) = \mathrm{id}\}$. Then $\ker(\phi)$ is a normal subgroup of $G$. We will see below that normal subgroups provide a weak form of commutativity, that allows us to construct more complex groups out of simpler ones.

**Direct products.** The most natural way to compose larger groups from smaller ones is via the *direct product*. Given two groups $G$ and $H$, we can form a new group with elements $\{(g, h) : g \in G, h \in H\}$ with a binary operation that is applied component-wise $(g, h) \cdot (g', h') = (g \cdot g', h \cdot h')$ (here, $\cdot$ is overloaded to be the group operation for all three groups). This direct product group is denoted $G \times H$. In the context of permutation groups, say $G$ is a permutation group over ground set $Q_G$ and $H$ is over ground set $Q_H$. Then $G \times H$ has ground set $Q_G \times Q_H$ and every function in $G \times H$ factorizes component-wise, i.e., every element in $G \times H$ is identified with a permutation $(q_G, q_H) \mapsto (g(q_G), h(q_H))$ where $g \in G, h \in H$.

Observe that $G \times H$ contains normal subgroups which are isomorphic to both $G$ and $H$. To see this, take $N = \{(e_G, h) : h \in H\}$ where $e_G$ is the identity element in $G$. Then since $ge_G = e_G g$ and since $H$ is closed under its group operation, we have $(g, h)N = N(g, h)$ for all $(g, h) \in G \times H$. A symmetric argument shows that $G$ is also a normal subgroup of the direct product.

Note that we can analogously define direct products in the absence of the group axioms, and thus for monoids and semigroups. This gives a natural construction of the semigroup corresponding to moving around both axes of a 2-dimensional rectangular gridworld, as a concatenation of two non-interacting 1-dimensional gridworlds:

**Example 5** (2D gridworld). *If $G_S$ is the transformation semigroup of the 1-d grid world with $S + 1$ states, then $G_S \times G_S$ corresponds to a 2-dimensional gridworld. A semiautomaton that yields this transformation semigroup has state space $Q = \{(i, j) : i, j \in \{0, \ldots, S\}\}$ and 5 actions: increment or decrement $i$ or $j$, subject to boundary effects, or do nothing.*

The definition of direct product extends straightforwardly to more than two terms $G_1 \times G_2 \times \ldots \times G_n$; we identify the items with tuples $(g_1, g_2, \ldots, g_n)$.

**Semidirect products.** However, it is possible to compose larger groups so that one of the subgroups is *not* a normal subgroup. This operation is called a *semidirect product*, with the group law $(g, h) \cdot (g', h') = (g \cdot \phi_h(g'), h \cdot h')$ for some $\phi_h$ to be defined later. Observe that in the direct product $G \times H$, we have constructed the elements from ordered pairs ($g \in G, h \in H$), lifting $G$ and $H$ into a shared *product space* (i.e., the Cartesian product of the underlying sets of $G$ and $H$), defining the group operation as simply applying those of $G$ and $H$ separately.

In fact, there are other ways, to define the group operation in the product space, but a difficulty arises: we need to find other nontrivial multiplication rules on pairs $(g, h)$, and we cannot take for granted that an arbitrary binary operation satisfies the group axioms. We would like to define other operations $(g, h) \cdot (g', h')$ which output an element of $g$ and an element of $h$. An attempt would be to pick two arbitrary injective homomorphisms $\phi_G, \phi_H$ which embed $G$ and $H$ into a "shared space," so that elements of $G$ and $H$ can be multiplied together:

$$(g, h) \cdot (g', h') := \phi_G(g) \cdot \phi_H(h) \cdot \phi_G(g') \cdot \phi_H(h').$$

---

[13]An equivalent definition of a normal group is a subgroup $N$ such that $g^{-1}ng \in N, \forall g \in \mathcal{G}, n \in N$.

However, we need to ensure that this group operation is closed. Since all elements of the group are of the form $(g, h)$ where $g \in G$ and $h \in H$, we must find a pair $(\tilde{g}, \tilde{h})$ that yields the right hand side of the above display when embedded into the shared space via $\phi_G, \phi_H$. For this, the most natural choice is $\tilde{g} = g \cdot g'$ and $\tilde{h} = h \cdot h'$, and thus we must check that:

$$\phi_G(g \cdot g') \cdot \phi_H(h \cdot h') = \phi_G(g) \cdot \phi_G(g') \cdot \phi_H(h) \cdot \phi_H(h')$$
$$= \phi_G(g) \cdot \phi_H(h) \cdot \phi_G(g') \cdot \phi_H(h') = (g, h) \cdot (g', h')$$

However, the middle equality may not hold, because $\phi_G(g')$ and $\phi_H(h)$ are not guaranteed to commute. (Observe that for the special case of $g \mapsto (g, e_H), h \mapsto (e_G, h)$, these two elements always commute, giving rise to the direct product.)

Eliding $\phi_G, \phi_H$ and simply using $g, h$ as elements of the shared space, a sufficient condition for this to hold is that $hg'h^{-1} \in G$, since then, for some $\tilde{g} \in G$,

$$(g, h) \cdot (g, h') = ghg'(h^{-1}h)h' = g(hg'h^{-1})hh' = g\tilde{g}hh',$$

which is of the form $\phi_G(\cdot) \cdot \phi_H(\cdot)$ since both $G$ and $H$ are themselves closed. This condition is precisely that $G$ is a normal subgroup.

There is a degree of freedom here: for each pair $h$ and $g'$, we can choose which element of $G$ is given by $hg'h^{-1}$. When we make this choice we must ensure all of the group axioms are preserved, e.g., when $h = e_H$ we should always have $e_H g' e_H = g'$. Suppose we make this choice and define $\phi_h : g \mapsto hgh^{-1} \in G$ (this $\phi$ is a homomorphism from $H \to \text{Aut}(G)$, where $\text{Aut}(\cdot)$ denotes the *automorphism group*, the group of bijections on $G$ that preserve the group axioms, under composition). Then, these ordered pairs do indeed form a group, but the group operation is

$$(g, h) \cdot (g', h') = g\phi_h(g')hh'$$

This object is the semidirect product, and it is denoted $G \rtimes H$. Note that the choice of mapping $\phi$ is unspecified in the notation, and, in general, different choices of $\phi$ will yield different structures for the semidirect product.

Finally, when $G = N \rtimes H$, both $N$ and $H$ are subgroups of $G$, but $N$ is also a normal subgroup. To see this, we need to check that $hN = Nh$ for any $h \in H$. This is equivalent to $hnh^{-1} \in N$ for each $h, n$, but we defined the group operation to be $hnh^{-1} = \phi_h(n) \in N$, specifically so this would hold. On the other hand, $H$ may not be a normal subgroup, and in this sense the semidirect product is a generalization of the direct product (for which both subgroups are normal). However, when the mapping $\phi$ is trivial, that is $\phi_h(n) = n$ then both $N$ and $H$ are normal subgroups, and one can verify that in this case the semidirect product and direct product coincide.

**Example 6** (Dihedral group). *Consider a semiautomaton with $Q = \{0, \ldots, S - 1\} \times \{-1, +1\}$ and input alphabet $\Sigma = \{\text{advance}, \text{reverse}\}$. The transitions are given by:*

$$\delta((s, b), \text{advance}) = (s + b \bmod S, b)$$
$$\delta((s, b), \text{reverse}) = (s, -b)$$

*The transformation semigroup for this semiautomaton is $C_S \rtimes C_2$ where $C_S$ is the cyclic group on $S$ elements (cf. Example 4). $C_2$ has two elements, the identity $e$ and one element $h$ such that $hh = e$. $C_S$ has $S$ elements where each element $g$ is a function that adds some number $k \in \{0, \ldots, S - 1\}$ to the input modulo $S$. The inverse $g^{-1}$ is naturally to subtract $k$ to the input, modulo $S$. The homomorphism $\phi$ in the semidirect product is such that $\phi_e(g) = g$ and $\phi_h(g) = g^{-1}$.*

**Wreath products.** We define one more type of product between groups $N$ and $H$: the *wreath product* $N \wr H := (N \times \ldots \times N) \rtimes H$. This is a group containing $|N|^{|H|} \cdot |H|$ elements (rather than $|N| \cdot |H|$, like the direct and semidirect products). Intuitively, it is defined by creating one copy of $N$ per element in $H$ via the direct product, then letting $H$ specify a way to *exchange* these copies. Formally, $N \wr H$ is the unique group generated by

$$(g_1, \ldots, g_{|H|}, h) \quad \forall g_i \in N, h \in H,$$

where

$$(g_1, \ldots, g_{|H|}, e_H) \cdot (g'_1, \ldots, g'_{|H|}, e_H) := (g_1 \cdot g'_1, \ldots, g_{|H|} \cdot g'_{|H|}, e_H) \quad \forall g_i, g'_i \in N,$$

and

$$(g_1, \ldots, g_{|H|}, e_H) \cdot (e_N, \ldots, e_N, h) := (g_{\pi_h(1)}, \ldots, g_{\pi_h(|H|)}, e_H) \quad \forall g_i \in N, h \in H, \quad \text{(A.1)}$$

where we have enumerated the elements of $H$ in arbitrary order, such that each $\pi_h : [H] \to [H]$ is the permutation defined by right multiplication $h' \mapsto h'h$ (by convention).

To write this explicitly as a semidirect product $N \wr H := (N \times \ldots \times N) \rtimes H$, the homomorphism into the direct product's automorphism group $\phi : H \to \mathrm{Aut}(N \times \ldots \times N)$ is given by A.1: for each $h \in H$, $\phi$ is the automorphism defined by permuting the indices between the terms in the direct product, according to the permutation induced by right multiplication by $h$.

**Example 7** (Rubik's Cube). *A naive way to construct the Rubik's Cube is to assign labels* $\{1, \ldots, 54\}$ *to the stickers on the cube, and define the Rubik's Cube group $G$ via the sticker configurations reachable by the 6 face turns (which each specify a permutation $\delta_L, \delta_R, \delta_U, \delta_D, \delta_B, \delta_F :$ $[54] \to [54]$ of the stickers). This establishes $G$ as a subgroup of $S_{54}$. First, notice that the 6 central stickers never move (so this is really improvable to $S_{48}$). Next, notice that the $24 = 8 \times 3$ vertex stickers never switch places with the $24 = 12 \times 2$ edge stickers. The vertex stickers form a subset of the wreath product $C_3 \wr S_8$, while the edge stickers form a subset of the wreath product $C_2 \wr S_{12}$. In all, this realizes $G$ as a subgroup of a direct product of wreath products:*

$$G \le (C_3 \wr S_8) \times (C_2 \wr S_{12}).$$

*Among other consequences towards solving the Rubik's Cube, this gives an improved upper bound on the size of $G$ (which turns out to still be off by a factor of 12, because of nontrivial invariants preserved by the face rotations, a.k.a. unreachable configurations).*

**Quotients, simple groups, and maximal subgroups.** When $N$ is a normal subgroup of $G$, the quotient group $G/N$ is defined as $\{gN : g \in G\}$ with binary operation $(gN)(g'N) = (gg')N$. The fact that $N$ is a normal subgroup implies that this is a well defined group. We can also check that if $G = N \rtimes H$ then the quotient group $G/N$ is isomorphic to $H$, which matches the intuition for multiplication and division.

A $G$ group is *simple* if it has no non-trivial normal subgroups. Intuitively, a simple group cannot be factorized into components; this generalizes the fact that a prime number admits no non-trivial factorization. When $G$ is not simple then it has a non-trivial normal subgroup, say $N$. We call $N$ proper if $N \ne G$. We call a proper subgroup $N$ *maximal* if there is no other proper normal subgroup $N' \lhd G$ such that $N \lhd N'$. Equivalently, $N$ is a maximal proper normal subgroup if and only if $G/N$ is simple. This is akin to extracting a prime factor from a number, since the quotient group $G/N$ cannot be further factorized. We will revisit this idea of factorization when defining *composition series* and *solvable groups* in Section C.3.2.

**Group extensions.** Finally, we provide some additional terminology related to these different notions of products, which provide a cleaner unifying language in which to state our constructions. Let $N, H$ be arbitrary groups. Which groups $G$ contain a normal subgroup isomorphic $N$, such that the quotient $G/N$ is isomorphic to $H$? Such a group $G$ is said to be an *extension* of $N$ over $H$. The direct product $G = N \times H$ is known as the *trivial extension*. A semidirect product $G = N \rtimes H$ is known as a *split extension*. However, not all extensions are split extensions; the smallest example is the *quaternion group* $Q_8$, the group of unit quaternions $\{\pm 1, \pm i, \pm j, \pm k\}$ under multiplication ($i^2 = j^2 = k^2 = ijk = -1$), which cannot be realized as a semidirect product of smaller groups. In general, it is very hard to derive interesting properties of a group extension based on the properties of $N$ and $H$. Fortunately, there *is* a characterization of general extensions. The Krasner-Kaloujnine *universal embedding theorem* (Krasner & Kaloujnine, 1951) states that all extensions $G$ can be found as subgroups of the wreath product $N \wr H$. The proof of Theorem 2 essentially shows how to *implement* the different kinds of group extensions, given constructions which implement the substructures $N, H$. In the worst case, we will have to implement a wreath product.

## A.3 SHALLOW CIRCUIT COMPLEXITY CLASSES

We provide an extremely abridged selection of relevant concepts in circuit complexity. For a systematic introduction, refer to (Arora & Barak, 2009). In particular, we discuss each circuit complexity class and inclusion below:

$$\mathsf{NC}^0 \subset \mathsf{AC}^0 \subset \mathsf{ACC}^0 \subseteq \mathsf{TC}^0 \subseteq \mathsf{NC}^1.$$

- $\text{NC}^0$ is the class of constant-depth, constant-fan-in, polynomial-sized AND/OR/NOT circuits. If a constant-depth Transformer only uses the constant-degree sparse selection constructions in (Edelman et al., 2022), it can be viewed as representing functions in this class. However, the representational power of these circuits is extremely limited: they cannot express any function which depend on a number of inputs growing with $T$.

- $\text{AC}^0$ is the class of constant-depth, unbounded-fan-in, polynomial-sized AND/OR circuits, allowing NOT gates only at the inputs. A classic result is that the parity of $T$ bits is not in $\text{AC}^0$ (Furst et al., 1984); Hahn (2020) concludes the same for bounded-norm (and thus bounded-Lipschitz-constant) constant-depth Transformers.

- $\text{ACC}^0$ extends $\text{AC}^0$ with an additional type of unbounded-fan-in gate known as $\text{MOD}_p$ for any prime number $p$, which checks if the sum of the input bits is a multiple of $p$. Theorem 2 comes from the fact that the semigroup word problem (which is essentially identical to semiautomaton simulation) is in this class; see (Barrington & Thérien, 1988).

- $\text{TC}^0$ extends $\text{AC}^0$ with an additional type of unbounded-fan-in gate called MAJ, which computes the majority of an odd number of input bits (a *threshold* gate). It is straightforward to simulate modular counters using a polynomial number of parallel thresholds (i.e. $\text{ACC}^0 \subseteq \text{TC}^0$). Whether this inclusion is strict (*can you simulate a threshold in constant depth with modular counters?*) is a salient open problem in circuit complexity. Threshold circuits are a very natural model for objects of interest in machine learning like decision trees and neural networks (Merrill et al., 2021).

- $\text{NC}^1$ is the class of $O(\log T)$-depth, constant-fan-in, polynomial-sized AND/OR/NOT circuits. It is an extremely popular and natural complexity class capturing *efficiently parallelizable* algorithms. It is unknown whether any of the inclusions in the "larger" classes $\text{TC}^0 \subseteq \text{NC}^1 \subseteq \text{L} \subseteq \text{P}$ are strict.

## A.4 THE TRANSFORMER ARCHITECTURE

In this section, we define the Transformer function class used in our theoretical results, and discuss remaining discrepancies with the true architecture.

An $L$-layer *Transformer* is a sequence-to-sequence network $f_{\text{tf}} : \mathbb{R}^{T \times d} \times \Theta_{\text{tf}} \to \mathbb{R}^{T \times d}$, consisting of alternating self-attention blocks and feedforward blocks

$$f_{\text{tf}} := f_{\text{mlp}}^{(L)} \circ f_{\text{attn}}^{(L)} \circ f_{\text{mlp}}^{(L-1)} \circ ... \circ f_{\text{attn}}^{(1)}.$$

The parameter space $\Theta_{\text{tf}}$ is the Cartesian product of those of the individual blocks (without recurrent weight sharing across layers, by default). We define these two types of blocks below.

**Attention.** A single-headed ($H = 1$) *self-attention block* is a sequence-to-sequence network $f_{\text{attn}} : \mathbb{R}^{T \times d} \times \Theta_{\text{attn}} \to \mathbb{R}^{T \times d}$, parameterized by $\theta_{\text{attn}} = (W_Q, W_K, W_V, W_C)$. With an inner embedding dimension $k$, the shapes of these matrices are as follows: $W_Q, W_K, W_V, W_C^\top \in \mathbb{R}^{d \times k}$. Each head, indexed by $t \in [T]$, computes pairwise *query-key alignment scores* $\langle W_Q^\top x_t, W_K^\top x_{t'} \rangle$ for each position $t' \in [T]$, normalizes them with a $T$-dimensional causally-masked softmax (forcing weights for positions $t > t'$ to be 0), and uses these *attention weights* $\alpha \in \mathbb{R}^T$ to mix value embeddings: $\sum_{t' \in [T]} \alpha_{t'} W_V^\top x_{t'}$. This mixture is mapped back into $\mathbb{R}^d$ by multiplying by $W_C$, to form the $t$-th row of the output matrix. In a single equation:

$$f_{\text{attn}}(X; W_Q, W_K, W_V, W_C) := \text{CausalAttn}(X W_Q W_K^\top X^\top) X W_V W_C,$$

where $\text{CausalAttn} : \mathbb{R}^{T \times T} \to \mathbb{R}^{T \times T}$ applies a row-wise causally-masked $T$-dimensional softmax function. The standard softmax function $\text{softmax}(z) : \mathbb{R}^T \to \mathbb{R}^T$ is defined by

$$[\text{softmax}(z)]_t := \frac{e^{z_t}}{\sum_{t' \in [T]} e^{z_{t'}}};$$

the causally-masked softmax at row $t$ is defined to be $\text{softmax}(z_{1:t})$ on the first $t$ coordinates, and 0 on the rest. To implement the causal masking operation, it is customary to set the entries above the diagonal of the attention score matrix $X W_Q W_K^\top X^\top$ to $-\infty$, then obtaining $\text{CausalAttn}(X W_Q W_K^\top X^\top)$ via a row-wise softmax (letting $e^{-\infty}$ evaluate to 0).

In general, for any positive integer $H$, a *multi-headed self-attention block* consists of a sum of $H$ copies of the above construction, each with its own parameters.

This component is often called *soft attention*: the softmax performs continuous selection, taking a convex combination of its inputs. In contrast, *hard attention* refers to attention heads which perform truly sparse selection (putting weight $1$ on the position with the highest score, and $0$ on all others).

**Feedforward MLP.** An $L'$-layer *position-wise feedforward MLP block* is a sequence-to-sequence network $f_{\text{mlp}} : \mathbb{R}^{T \times d} \times \Theta_{\text{mlp}} \to \mathbb{R}^{T \times d}$, parameterized by $\theta_{\text{mlp}} = (W_1, b_1, \ldots, W_{L'}, b_{L'})$. For a choice of activation function $\sigma : \mathbb{R} \to \mathbb{R}$ (which is always ReLU in our theoretical constructions, for simplicity), $f_{\text{mlp}}$ applies the same nonlinear map $(x \mapsto W_{L'}x + b_{L'}) \circ \sigma \circ \ldots \circ \sigma \circ (x \mapsto W_1 x + b_1)$ to each row $t$ of the input matrix $X \in \mathbb{R}^{T \times d}$ (with the same parameters per position $t$); here, $\sigma$ is applied pointwise.

Finally, an extra term $P \in \mathbb{R}^{T \times d}$ is added to the first layer's input, the matrix of *position encodings*.

**Residual connections.** It is typical to add *residual connections* which bypass each block. That is, letting id denote the identity function in $\mathbb{R}^{T \times d}$, the network (with position encodings) becomes

$$f_{\text{tf}} := (\text{id} + f_{\text{mlp}}^{(L)}) \circ (\text{id} + f_{\text{attn}}^{(L)}) \circ (\text{id} + f_{\text{mlp}}^{(L-1)}) \circ \ldots \circ (\text{id} + f_{\text{attn}}^{(1)}) + (\text{id} + P).$$

At the level of granularity of the results in this paper (up to negligible changes in the width and weight norms), this changes very little from the viewpoint of representation. A residual connection can be implemented (or negated) by appending two ReLU activations to a non-residual network:

$$x = (x)_+ - (-x)_+.$$

Similarly, a residual connection can be implemented with one attention head (with internal embedding dimension $k = d$), as long as it is able to select its own position (which will be true in all of our constructions).

In some of our constructions, we choose to use residual connections (sometimes restricted to certain dimensions); it will be very natural to view the embedding space $\mathbb{R}^d$ as a "workspace", where residual connections ensure that downstream layers can access the input (and position) embeddings, as well as outputs of all earlier layers. We will specify whether to use residual connections in each construction, to make the proofs as clear as possible. When we do so, we do not add the extra weights explicitly to $f_{\text{attn}}$ and $f_{\text{mlp}}$.

**Layer normalization.** For simplicity of presentation, we omit the normalization layers which are usually present after each attention and MLP block. It would be straightforward (but an unnecessary complication) to modify the function approximation gadgets in our constructions to operate with unit-norm embeddings.

**Padding tokens.** Finally, it will greatly simplify the constructions to add padding tokens: to simulate a semiautomaton at length $T$, we will choose to prepend $\tau$ tokens, with explicitly chosen embeddings, which do not depend on the input $\sigma_{1:T}$. Theorem 1 uses $\tau = \Theta(T)$ padding, and Theorem 2 uses $\tau = 1$. In both cases, padding is not strictly necessary (the same functionality could be implemented by the MLPs without substantially changing our results), but we find that it leads to the most intuitive and concise constructions.

**Complexity measures.** We define the following quantities associated with a Transformer network, and briefly outline their connection to familiar concepts in circuit complexity:

- The dimensions according to the definition of a sequence-to-sequence network: *sequence length* $T$ and *embedding dimension* $d$. Up to a factor of bit precision, this corresponds to the number of inputs in a classical Boolean circuit. We will exclusively define architectures where $d$ is independent of $T$.

- Its *depth* $L$, the number of repeated $f_{\text{mlp}} \circ f_{\text{attn}}$ blocks. When each of these modules contains a constant number of sequential computations, this coincides with the usual notion of circuit depth, up to a constant factor. This is true in practice and our theoretical treatment (the attention and MLP have a constant number of layers).

- The other shape parameters from the definition of the architecture: *number of heads* (per layer and position) $H$[14], and *internal embedding dimension* $k$. When $f_{\mathrm{mlp}}$ is an $L'$-layer MLP, it has *MLP intermediate widths* $d_1, \ldots, d_{L'-1}$. We will exclusively think of $L'$ as a small constant, so that the number of sequential matrix multiplications in the entire network is within a constant factor of $L$.

- Its *attention width* $w_{\mathrm{attn}}$ is defined to be the maximum of $\{d, Hk\}$, and its *MLP width* $w_{\mathrm{mlp}}$ is defined as the maximum of $\{d_1, \ldots, d_{L'-1}\}$. Taking $w = \max(w_{\mathrm{attn}}, w_{\mathrm{mlp}})$ as a coarse upper bound we will use to summarize the number of per-position trainable embeddings in our constructions. To map this to the usual notion of *circuit size*, note that the computations are repeated position-wise. Thus, Transformers induce a computational graph with $O(T \cdot L \cdot w)$ gates and $O(T \cdot L \cdot w^2)$ wires. The position-wise parameter-sharing induces a special notion of circuit uniformity.

- A bound on its $\infty$-weight norms: the largest absolute value of any trainable parameter. These can be converted into norm-based generalization bounds via the results in Edelman et al. (2022). Note that the results in this paper go beyond the *sparse variable creation* constructions of bounded-norm attention heads; in general, the norms scale with $T$. The attention heads express meaningful non-sparse functions. Aside from the positive experimental results, we do not directly investigate generalization in this paper.

- The bit precision (length of finite-precision truncation of real numbers in a computational graph implementing $f_{\mathrm{tf}}$), which lets us implement approximate real-valued computations as Boolean (or discrete arithmetic) circuits. With infinite-precision real numbers, there are pathological constructions for RNNs (Siegelmann & Sontag, 1992) and Transformers (Merrill et al., 2021) which give single parameters of neural networks infinite representational power. Throughout this work, our circuits will work with $O(\log T)$ bit precision, which can represent real numbers (as integers $\lfloor x \cdot 2^c \rceil$ in their binary representation, for some choice of $c = \Theta(\log T)$) with magnitude up to $O(\mathrm{poly}(T))$, with $O(1/\mathrm{poly}(T))$ approximation error. Since this is far from the focus of our results, we will elide details for the remainder of this paper, returning to these considerations only to make Theorem 4 more concrete. All of our constructions are robust up to this noise level: this is because the internal weight norms and activations are bounded by a quantity at most polynomial in $T$, and the function approximation construction in Lemmas 1 and 2 can tolerate $1/\mathrm{poly}(T)$ perturbations using $\mathrm{poly}(T)$ weight norms.

### A.5 Additional discussion of related work

**Relevant applications.** We first provide references for the "reasoning-like" applications of neural networks mentioned in the main paper.

- Program synthesis: (Chen et al., 2021b; Schuster et al., 2021; Li et al., 2022).
- Mathematical reasoning: (Lample & Charton, 2019; Polu & Sutskever, 2020; Drori et al., 2022).
- Neural dynamics models for decision-making: recurrent (Hafner et al., 2019; Ye et al., 2021; Micheli et al., 2022) and non-recurrent (Chen et al., 2021a; Janner et al., 2021).

**Synthetic combinatorial experiments (and relations).** We provide an expanded discussion of empirical analyses of neural networks trained on synthetic combinatorial tasks.

- *Pointer Value Retrieval:* Zhang et al. (2021a) propose a benchmark of tasks based on pointer value retrieval (PVR) to study the generalization ability (in-distribution as well as distribution shift) of different neural network architectures. Their key idea behind the task is "indirection through a pointer rule", that is, a specific position of the input acts as a pointer to the relevant position (window) of the input which contains the answer. Using our results, we can implement a certain sub-class of PVR tasks: (1) we use the first attention layer to identify the pointer, and (2) we use the second attention layer to select the window between the pointer value and the width. (2) is doable with $O(1)$ attention heads if we are computing a function that is based on the sum (for example, $\mod n$). Otherwise it would require the window size number of attention heads similar to our grid-world construction.

---

[14]There will be a notational collision between $h, H$ denoting attention heads, and $h \in H$ denoting an element in a group. We keep the overloaded notation for clarity, and this will certainly be unambiguous.

- *LEGO*: Zhang et al. (2022) propose a task based on solving a simple chain-of-reasoning problem based on group-based equality constraints. They study the ability of transformers to generalize the entire chain of reasoning given only part of the chain while training. A direct comparison to our setting is not clear since this task is not modelled as a sequence-to-sequence task, however it serves as another example of the emergence of "shortcut" solutions: transformers solve certain variables without resolving the chain of reasoning.

- *Dyck*: Several works (Hahn, 2020; Ebrahimi et al., 2020; Newman et al., 2020; Yao et al., 2021) have studied the ability of Transformers to represent Dyck languages, both for generation and closing bracket prediction. The most closely related to our work is Yao et al. (2021), which constructs a clever depth-2 as well as a depth-$D$ solution for bounded-depth $D$ Dyck languages. Bounded-depth Dyck can be captured by our semiautomata formalism and our main construction would recover the depth-$2^D$ solution by default. Their depth-2 construction bears semblance to the constructions we use in Theorem 3: they implement a counter in the first layer similar to our $\mod n$ construction, and implement a proximity-based depth matching in the second layer. Our grid-world construction generalizes their construction to a significantly more complex problem. We view our work as a generalization of their results to a wider class of semiautomata.

- *Parity*: Another commonly studied synthetic setup is the task of learning parities. Edelman et al. (2022); Barak et al. (2022) perform a theoretical and empirical study of the ability of Transformers (and other architectures) to learn sparse parities where the support size $k \ll T$. Bhattamishra et al. (2020); Schwarzschild et al. (2021) study the task of computing prefix sum in the binary basis (which is essentially parity of the prefix sum) for Transformers and recurrent models, repsectively. Anil et al. (2022); Wei et al. (2022) study essentially the same problem however they model the task as a natural language task and use pretrained Transformers.

- *Modular addition:* In the pursuit of understanding grokking, Nanda & Lieberum (2022) focus on the task of adding two 5 digit numbers modulo a large prime (113 in their setting). They take the viewpoint of mechanistic interpretability and attempt to reverse engineer what the Transformer is learning on this task in the low sample regime. They claim that the trained model learns sinusoidal encodings that we also use in our theoretical constructions. Note that our setting of modular counters performs a $T$-way summation, while their setting involves only a 2-way summation (with carryover). Inspired by their work, we do some preliminary investigation into interpreting the trained Transformer on the grid world (see Figure 7).

**Formal languages and neural networks.** Dyck languages are particularly interesting for their completeness property: the Chomsky-Schützenberger representation theorem (Chomsky & Schützenberger, 1959) states that all context-free languages can be (homomorphically) represented by the intersection of a Dyck language and a regular language. For more on this topic, see the discussion in Yao et al. (2021). In the context of regular languages (which in general induce finite-state automata), our findings imply that $O(\log T)$-depth networks can simulate all context-free languages (Theorem 1), and $O(1)$-depth networks can represent some of them. The obstructing regular languages are the ones whose associated syntactic monoids are non-solvable. We further note that the gridworld semigroups are *aperiodic* and thus simulable by star-free regular expressions (Schützenberger, 1965) and $AC^0$ circuits (Chandra et al., 1983; Barrington & Thérien, 1988). We did not see a way for this to generically entail $O(1)$-depth shortcuts with self-attention. For the relation between the Chomsky hierarchy and various neural networks *in practice*, Delétang et al. (2022) provide an extensive empirical study for memory-augmented RNNs and Transformers on tasks spanning all 4 levels of the hierarchy, and conclude the Transformers lack the ability to even recognize regular languages. Their results do not contradict with ours, since they measure performance on "inductive inference", which is similar to our length generalization setup where we also see the failure of Transformer.

**Different axes of generalization: length, size, and algorithmic.** There has been much recent interest in quantifying out of distribution generalization of trained models under distribution shifts that maintain some notion of "logical" invariance. Wei et al. (2022); Anil et al. (2022) empirically investigate the ability of pre-trained Transformers to generalize to longer sequence length for parity-like problems modelled as language tasks. Xu et al. (2020) study size generalization in graph neural networks where they train on small graphs and evaluate on larger sized graphs with similar structural properties. Schwarzschild et al. (2021); Bansal et al. (2022) focus on length and algorithmic generalization for recurrent models where they train on simple/easy instances of the underlying

problem and evaluate on harder/complex instances using the power of recurrence to simulate extra computational steps, inspired by the ideas of Neural Turing Machines (Graves et al., 2014) and Adaptive Computation Time (Graves, 2016). We view our results as complementing those of Yao et al. (2021); Anil et al. (2022) for a richer class of problems. Our use of scratchpad is inspired by Nye et al. (2021); Wei et al. (2022); Anil et al. (2022).

**Recurrent Transformers.**    Our work is not the first to notice that Transformer architectures make brittle predictions out-of-distribution. Indeed, even the seminal paper introducing the architecture (Vaswani et al., 2017) notes that length generalization is promoted by a subtle hyperparameter choice (namely, the positional encoding scheme). Furthermore, there have been several attempts to reconcile this gap by modifying Transformers to behave more like RNNs; (Dehghani et al., 2019; Nye et al., 2021; Wei et al., 2022; Anil et al., 2022; Hutchins et al., 2022). Kasai et al. (2021) consider training a non-recurrent Transformer, and finetuning it into an RNN. All of these works have some element of natural language experiments: either the task is end-to-end language modeling, or the synthetic reasoning task is framed as a natural language problem, for a pretrain-finetune pipeline. We view our work as strengthening the foundations of these lines of inquiry. Theoretically, we provide structural guarantees for how shallow non-recurrent models can (perhaps deceptively) fit recurrent dynamics over long sequences. Empirically, we perform a *pure* (no confounds arising from the influence of a natural langauge corpus) analogue of the experiments seeking to help neural networks follow long chains of reasoning.

**Recurrent vs. non-recurrent sequence transduction.**    As mentioned briefly towards the end of Section 5, the setting of indirectly-supervised semiautomata matches that of autoregressive generative modeling (a.k.a. next-token prediction), if the continuations of the sequence depend on the state of a latent semiautomaton. This is the case in (for example) generating Dyck languages (Yao et al., 2021), where the possible continuations are {all possible open brackets, if the stack $q_t$ is not full} $\cup$ {close bracket which pairs with the top of the stack $q_t$}. We note that when an autoregressive model is used for sequence generation via a token-by-token inference procedure, this amounts to a special case of scratchpad inference (with a naive 1-step training procedure): the constant-depth network is used as a single iteration of a recurrent network, whose state is the completed prefix of the current generated sequence. Non-autoregressive natural language generation and transduction are an exciting area of research (Gu et al., 2017); for a recent survey, see Xiao et al. (2022). Our results are relevant to this line of work, suggesting that there may not be an expressivity barrier to expressing deep recurrent linguistic primitives, but there may be issues with out-of-distribution robustness.

**Algebraic structures in deep learning.**    Another area where tools from abstract algebra are used to reason about neural networks is *geometric deep learning*, a research program which seeks to understand how to specify inductive biases stemming from algebraic invariances. For a recent survey, see Bronstein et al. (2021). In contrast, this work studies the ability of a fixed architecture to learn a wide variety of algebraic operations, in the absence of special priors (but a large amount of data). There are certainly possible connections (e.g. *"how do you bias an architecture to perform operations in a known group, when there is limited data?"*) to explore in future work.

**Theoretical role of depth.**    Our theoretical results can be interpreted as a *depth separation* result: contingent on $\mathsf{TC}^0 \neq \mathsf{NC}^1$, it takes strictly more layers to simulate non-solvable semiautomata, compared to their solvable counterparts. In a similar spirit, there have been several works establishing depth separation for feed-forward neural networks (mostly using ReLU activations) (Telgarsky, 2016; Eldan & Shamir, 2016; Daniely, 2017; Lee et al., 2017; Safran et al., 2019). These results are usually constructive in nature, that is, they show the existence of functions that can be represented by depth $L$ but would require exponential-width for depth $L - 1$ (or $\sqrt{L}$, depending on the result).

## B   EXPERIMENTS

### B.1   SECTION 4: SGD FINDS THE SHORTCUTS, UNDER IDEAL SUPERVISION

This section contains a full description and discussion of the in-distribution simulation experiments from Section 4.

### B.1.1 Shallow Transformers simulate small groups and semigroups

The main experiments in this paper investigate whether gradient-based training of Transformers finds low-depth solutions to the problem of simulating semiautomata. In these experiments, we consider a wide variety of semiautomata $\mathcal{A}$, corresponding to various groups and semigroups, and construct a distribution $\mathcal{D}_{\mathcal{A}}$ over input sequences $(\sigma_1, \ldots \sigma_T)$ and their corresponding state sequences $(q_1, \ldots, q_T) = \mathcal{A}_{T,q_0}(\sigma_{1:T})$. In each setting, the $\sigma_t$ are chosen uniformly at random from the set of valid tokens in $\Sigma$. [15] Given this distribution $\mathcal{D}_{\mathcal{A}}$, and a sequence-to-sequence neural network (with a token embedding and a linear classification head) which maps $\Sigma^T$ to token predictions $Y \in \mathbb{R}^{T \times |Q|}$ (such that $Y_{t,q} := \widehat{\mathbf{Pr}}_\theta(q_t = q | \sigma_{1:t})$), we establish the task of minimizing the cross-entropy loss

$$L(\theta) := \frac{1}{T} \sum_{t=1}^{T} \log(1/Y_{t,q_t}).$$

This defines a supervised learning problem over sequences.

Note that without intermediate states in the input these problems exhibit *long-range dependencies*: for example, in the parity semiautomaton (and for any semiautomaton whose transformation semigroup is a group), every $q_t$ depends on *every* preceding input $\{\sigma_{t'} : t' < t\}$. Indeed, this is why previous studies have used group operations as a benchmark for reasoning (Anil et al., 2022; Zhang et al., 2022).

**Settings.** We proceed to enumerate the semiautomata considered in these simulation experiments.

- Cyclic groups $C_2, C_3, \ldots, C_8$. For each cyclic group $C_n$ (realized as $Q := \{0, 1, \ldots, n-1\}$ under mod-$n$ addition), we choose the generator set $\Sigma$ to be the full set of group elements $\{0, \ldots, n-1\}$. An alternative could be to let $\Sigma$ be a minimal[16] set $\{0, 1\}$, which we do not use in the experiments.

- Direct products of cyclic groups $C_2 \times C_2, C_2 \times C_2 \times C_2$, realized as concatenated copies of the component semiautomata. Note that $C_6$ (which is isomorphic to $C_2 \times C_3$), included in the above set, is another example.

- Dihedral groups $D_6, D_8$. Our realization of $D_{2n}$ chooses $Q = \{0, 1, \ldots, n-1\} \times \{0, 1\}$ and $\Sigma = \{(1, 0), (0, 1)\}$. Since these groups are non-abelian, it is already not so straightforward (compared to parity) to see why constant-depth shortcuts should exist.

- Permutation groups $A_4, S_4, A_5, S_5$. We choose $Q$ to be the set of $n!$ permutations for $S_n$ (*symmetric group*), and $Q$ to be the set of $\frac{n!}{2}$ even permutations for $A_n$ (*alternating group* on $n$ elements). The generator set for $S_n$ consists of the minimal generators, a transposition and an $n$-cycle, as well as 6 other permutations. [17] For $A_n$, we choose the 3-cycles of the form $(12i)$ for $i \in \{3, 4, \cdots, n\}$. Note that $A_4, S_4$ are solvable (leading to constant-depth shortcuts), while $A_5, S_5$ are not. Also, note that to learn a constant-depth shortcut for $A_4$, a model needs to discover the wondrous fact that $A_4$ has a nontrivial normal subgroup, that of its double transpositions.

- The quaternion group $Q_8$. This is the smallest example of a non-abelian solvable group which is not realizable as a semidirect product of smaller groups, thus requiring the full wreath product construction (Lemma 10) in our theory.

- The Dyck language $\mathrm{Dyck}_{n,k}$ (correctly nested brackets of $k$ types, with depth at most $n$). We take $n = 4$, $k = 2$ in the experiments. To realize $\mathrm{Dyck}_{n,k}$ as a semiautomaton simulation problem, the state $Q$ is the state of the stack which implements Dyck language recognition (there are thus $\sum_{i=0}^{n} k^i$ distinct states); [18] $\Sigma$ is the set of $2k$ opening and closing brackets. The

---

[15]Take for instance the Dyck language, if the current stack is empty, then $\sigma_t$ is chosen uniformly from the choices of open parentheses but not the closing parentheses. This is in accordance with Yao et al. (2021).

[16]In the sense that it induces a non-trivial learning problem on this group. If we only pick the generator $\{1\}$, the output sequence is deterministic, and there is no learning problem.

[17]These other permutations are chosen following the ordering given by the sympy.combinatorics package. They are not necessary for covering the state space (since the minimal set of 2 permutations already suffice to cover $Q$), but can help speed up the mixing of the states.

[18]In the experiments we use $(k+1)^n$ classes (i.e. each of the $n$ positions can take $(k+1)$ possible values), $\sum_{i=0}^{n} k^i$ of which are reachable.

distribution in inputs is slightly different, since there is a notion of "illegal" inputs: if the stack is empty, then the set of feasible inputs contain all the opening brackets; if the stack is full (i.e. reaching depth $n$), then the only feasible input is the closing bracket for the opening bracket at the top of the stack.

- Gridworld semiautomata $\mathrm{Grid}_4, \mathrm{Grid}_9$, where $Q = \{0, 1, \cdots, n-1\}$ (for $n = 4$ or 9) and $\Sigma = \{\pm 1\}$.[19] For this special case, we have a constant-depth solution as stated in Theorem 3.

**Training.** We focus on the *online learning* setting for all experiments in this paper: at training iteration $i$, draw a fresh minibatch of samples from $\mathcal{D}_\mathcal{A}$, compute the network's loss and gradients on this minibatch, and update the model's weights using a standard first-order optimizer (we use AdamW (Loshchilov & Hutter, 2017)). This is to mitigate the orthogonal challenge of overfitting; note that the purpose of these experiments is to determine *whether* standard gradient-based training finds shortcut solutions in these combinatorial settings (in a reasonable amount of time), not *how efficiently*. We do not investigate how to improve sample efficiency in this paper. The results in the paper are based on sinusoidal positional encodings (Vaswani et al., 2017) unless otherwise specified.

**Sequence length.** We report our main results with sequence length $T = 100$, which is large enough to rule out memorization: for this choice of $T$, the inputs come from a uniform distribution over $|\Sigma|^{100} > 10^{30}$ sequences, rendering it overwhelmingly unlikely for a sample to appear twice between training and evaluation. We observed positive results in most of the settings for larger $T$, but training became prohibitively unstable and computationally expensive; mitigating this is an interesting direction for future empirically-focused studies.

**Depth.** We seek to investigate the sufficient depth for learning to simulate each semiautomaton. Thus, for each problem setting, we vary the number of layers $L$ in the Transformer between 1 and 16. Note that we do not attempt in this work to distinguish between depths $O(\log T)$ and $O(1)$, nor do we attempt to tackle the problem of exhaustively enumerating and characterizing the shortcut solutions for any particular semiautomaton.

**Results.** For each task and number of layers, we report the highest (Figure 5) and median (Figure 6) accuracies over 20 runs. The accuracy is calculated at token level (i.e. $\frac{1}{T}\sum_{t\in[T]}\mathbb{1}[\hat{q}_t = q_t]$), as opposed to the sequence-level accuracy (i.e. $\mathbb{1}[\hat{q}_{1:T} = q_{1:T}]$) as reported in Bhattamishra et al. (2020). We evaluate in-distribution accuracy on independent (unseen) samples of $\mathcal{D}_\mathcal{A}$, which contain 2048 sequences of length $T = 100$.[20] As shown in Figure 5, Transformers, trained with standard gradient-based methods, are able to find solutions which generalize well (in-distribution) on all of the tasks. Performance tends to improve as the number of layers increases (there is a small amount of non-monotonicity in some settings due to training instability); the sufficient depth to achieve high accuracy varies depending on the problem setting, as discussed below.

**Trends in sufficient depth.** The minimum number of layers required to achieve 99%+ performance reflects our beliefs on the difficulty of the task: a high-level trend is that the semigroups which don't contain groups (which only require memory lookups) are the easiest to learn, and among the groups, the larger non-abelian groups require more layers to learn, with the non-solvable group $S_5$ requiring the largest depth.[21] Between the non-abelian groups, the difficulty of learning $Q_8$ compared to $D_8$ (which has the same cardinality) agrees with our theoretical characterizations of the respective constant-depth shortcuts for these groups: $D_8$ can be written as a semidirect product of smaller groups, while $Q_8$ cannot, so our theoretical construction of a constant-depth shortcut must embed $Q_8$ in a larger structure (i.e. the wreath product).

**Improving training stability.** Throughout these experiments, we observe the following forms of training instability: high variance in training curves (based on initialization and random seeds for

---

[19] $-1$ for $L$, 1 for $R$. We omit the no-op $\perp$ in the experiment which does not change the difficulty of the task.

[20] This size is sufficient for evaluating the model performance: for example, for $C_2$ (i.e. parity), evaluating a model on 10 evaluation sets of this size gives a standard deviation of $0.031\%$ in the accuracy.

[21] However, we stress that these experiments do *not* control for the fact that larger groups have richer supervision (for example, $A_5$ has more informative labels than $A_4$), possibly accounting for the counterintuitive result that the latter requires more layers, despite being a subgroup of the former.

| | 1 | 2 | 3 | 4 | 5 | 6 | 7 | 8 | 9 | 10 | 11 | 12 | 13 | 14 | 15 | 16 |
|---|---|---|---|---|---|---|---|---|---|---|---|---|---|---|---|---|
| Dyck | 99.3 | 100 | 100 | 100 | 100 | 100 | 100 | 100 | 100 | 100 | 100 | 100 | 100 | 100 | 100 | 100 |
| $Grid_4$ | 99.9 | 100 | 100 | 100 | 100 | 100 | 100 | 100 | 100 | 100 | 100 | 100 | 100 | 100 | 100 | 100 |
| $Grid_9$ | 92.2 | 100 | 100 | 100 | 100 | 100 | 100 | 100 | 100 | 100 | 100 | 100 | 100 | 100 | 100 | 100 |
| $C_2$ | 77.6 | 99.8 | 99.9 | 100 | 100 | 99.5 | 100 | 99.7 | 100 | 100 | 100 | 100 | 100 | 100 | 100 | 100 |
| $C_3$ | 54.6 | 94.6 | 96.7 | 99.4 | 100 | 100 | 99.8 | 100 | 99.9 | 100 | 100 | 100 | 100 | 100 | 99.8 | 100 |
| $C_4$ | 95.1 | 92.3 | 84.2 | 99.9 | 99.7 | 99.9 | 100 | 100 | 100 | 100 | 100 | 100 | 100 | 100 | 100 | 100 |
| $C_5$ | 89.0 | 99.1 | 99.9 | 100 | 100 | 100 | 100 | 100 | 100 | 100 | 100 | 100 | 100 | 100 | 100 | 100 |
| $C_6$ | 59.8 | 98.7 | 75.5 | 99.9 | 99.8 | 99.9 | 99.9 | 100 | 100 | 100 | 99.8 | 99.9 | 100 | 99.8 | 99.9 | 99.9 |
| $C_7$ | 90.9 | 95.0 | 99.9 | 99.9 | 100 | 99.9 | 100 | 100 | 100 | 100 | 100 | 99.8 | 100 | 100 | 100 | 100 |
| $C_8$ | 79.6 | 96.2 | 99.8 | 99.8 | 99.9 | 100 | 99.9 | 99.9 | 100 | 99.4 | 99.9 | 99.9 | 99.9 | 100 | 99.9 | 99.9 |
| $C_2^2$ | 90.5 | 98.8 | 99.9 | 100 | 100 | 99.9 | 100 | 100 | 99.9 | 99.9 | 100 | 100 | 100 | 100 | 100 | 100 |
| $C_2^3$ | 65.0 | 77.9 | 99.9 | 97.9 | 100 | 99.8 | 98.2 | 99.9 | 100 | 100 | 91.9 | 95.9 | 91.7 | 90.6 | 87.5 | 80.6 |
| $D_6$ | 25.4 | 27.2 | 47.4 | 75.2 | 100 | 100 | 100 | 100 | 100 | 100 | 100 | 100 | 100 | 100 | 100 | 100 |
| $D_8$ | 45.6 | 98.0 | 100 | 100 | 100 | 100 | 100 | 100 | 100 | 100 | 100 | 100 | 100 | 100 | 100 | 100 |
| $Q_8$ | 31.6 | 49.2 | 59.6 | 60.4 | 73.5 | 99.3 | 100 | 100 | 100 | 100 | 100 | 100 | 100 | 100 | 100 | 100 |
| $A_4$ | 25.0 | 35.4 | 49.1 | 59.3 | 62.6 | 82.3 | 90.9 | 98.0 | 98.0 | 99.1 | 99.8 | 100 | 99.7 | 100 | 100 | 100 |
| $A_5$ | 12.5 | 23.1 | 32.5 | 46.7 | 71.2 | 98.8 | 100 | 100 | 100 | 100 | 100 | 100 | 100 | 100 | 100 | 100 |
| $S_4$ | 11.3 | 17.6 | 22.0 | 27.1 | 37.7 | 44.8 | 50.8 | 72.5 | 91.3 | 97.1 | 97.9 | 98.7 | 99.9 | 100 | 99.8 | 99.9 |
| $S_5$ | 7.9 | 11.8 | 14.6 | 19.7 | 26.0 | 28.4 | 32.8 | 51.8 | 86.3 | 94.8 | 90.2 | 97.2 | 99.3 | 99.1 | 99.9 | 99.9 |

Figure 5: A complete version of Figure 3, for various tasks (rows) and numbers of network layers (columns). Reported performance is the *maximum* test accuracy over 20 runs.

| | 1 | 2 | 3 | 4 | 5 | 6 | 7 | 8 | 9 | 10 | 11 | 12 | 13 | 14 | 15 | 16 |
|---|---|---|---|---|---|---|---|---|---|---|---|---|---|---|---|---|
| Dyck | 98.8 | 100 | 100 | 100 | 100 | 100 | 100 | 100 | 100 | 100 | 100 | 100 | 100 | 100 | 100 | 100 |
| $Grid_4$ | 99.8 | 100 | 100 | 100 | 100 | 100 | 100 | 100 | 100 | 100 | 100 | 100 | 100 | 100 | 100 | 100 |
| $Grid_9$ | 91.4 | 100 | 100 | 100 | 100 | 100 | 100 | 100 | 100 | 100 | 100 | 100 | 100 | 100 | 100 | 100 |
| $C_2$ | 56.4 | 83.0 | 79.9 | 80.9 | 89.1 | 85.2 | 84.8 | 84.9 | 88.8 | 94.5 | 98.3 | 86.4 | 90.4 | 88.7 | 94.6 | 99.3 |
| $C_3$ | 40.3 | 69.1 | 78.2 | 85.0 | 84.0 | 84.9 | 87.9 | 96.2 | 99.4 | 89.2 | 82.5 | 99.3 | 87.4 | 98.0 | 89.6 | 92.0 |
| $C_4$ | 56.8 | 63.8 | 56.2 | 64.2 | 69.5 | 71.5 | 75.9 | 73.7 | 85.8 | 68.0 | 77.1 | 84.1 | 64.9 | 71.1 | 64.3 | 99.3 |
| $C_5$ | 75.6 | 62.7 | 99.0 | 99.5 | 99.8 | 99.9 | 99.8 | 99.5 | 99.8 | 99.8 | 99.7 | 99.8 | 99.8 | 99.9 | 99.9 | 99.7 |
| $C_6$ | 45.8 | 49.0 | 53.0 | 59.6 | 75.5 | 77.0 | 95.6 | 91.2 | 83.4 | 59.6 | 98.4 | 72.9 | 89.7 | 94.5 | 99.8 | 87.5 |
| $C_7$ | 51.0 | 76.2 | 99.7 | 99.7 | 99.6 | 99.6 | 99.4 | 99.7 | 99.7 | 99.6 | 99.6 | 99.6 | 99.7 | 99.6 | 99.8 | 99.7 |
| $C_8$ | 60.5 | 58.8 | 99.0 | 98.5 | 99.6 | 99.7 | 99.4 | 99.5 | 99.6 | 98.5 | 99.5 | 99.8 | 99.8 | 99.6 | 99.3 | 99.7 |
| $C_2^2$ | 62.6 | 73.1 | 78.4 | 73.4 | 74.9 | 79.8 | 84.1 | 82.4 | 77.0 | 70.6 | 69.0 | 71.9 | 70.6 | 76.9 | 68.3 | 59.3 |
| $C_2^3$ | 50.0 | 61.4 | 60.6 | 60.7 | 72.4 | 63.2 | 63.8 | 66.4 | 69.8 | 59.0 | 63.4 | 54.6 | 59.5 | 53.0 | 44.7 | 48.4 |
| $D_6$ | 24.8 | 26.8 | 40.8 | 57.2 | 81.3 | 91.6 | 100 | 99.6 | 100 | 100 | 93.0 | 96.2 | 100 | 97.7 | 99.6 | 99.3 |
| $D_8$ | 38.1 | 63.6 | 99.7 | 100 | 100 | 100 | 100 | 100 | 100 | 100 | 100 | 100 | 100 | 100 | 100 | 100 |
| $Q_8$ | 29.0 | 45.8 | 38.5 | 42.7 | 57.4 | 79.5 | 84.7 | 89.2 | 95.9 | 98.1 | 98.8 | 97.8 | 99.8 | 98.3 | 98.8 | 99.4 |
| $A_4$ | 19.7 | 30.4 | 41.0 | 45.4 | 44.7 | 52.8 | 60.0 | 68.3 | 72.8 | 74.1 | 91.4 | 82.6 | 88.2 | 97.9 | 99.0 | 98.5 |
| $A_5$ | 10.5 | 18.7 | 26.6 | 30.5 | 40.6 | 63.9 | 77.2 | 99.4 | 99.3 | 100 | 100 | 100 | 100 | 100 | 99.9 | 100 |
| $S_4$ | 10.7 | 15.1 | 18.8 | 22.9 | 25.0 | 31.1 | 36.6 | 43.6 | 56.2 | 71.0 | 73.1 | 88.1 | 91.0 | 97.6 | 95.6 | 97.8 |
| $S_5$ | 7.1 | 11.0 | 13.1 | 16.5 | 20.9 | 24.3 | 29.4 | 37.6 | 40.1 | 59.0 | 60.4 | 91.3 | 91.2 | 94.6 | 98.0 | 99.1 |

Figure 6: The *median* accuracy for various tasks (rows) and numbers of network layers (columns). Reported performance is the *median* test accuracy over 20 runs.

the gradient-based optimization algorithm), and negative progress (i.e. non-monotonic loss curves), even for training runs which eventually converge successfully. This is evident in Figure 3(b),(c) and in the significant difference between the maximum accuracies in Figure 5 and the median in Figure 6.

To stabilize training, we experiment with dropout and exponential moving average (EMA)[22]. The effectiveness of dropout varies across datasets; for example, we find using a dropout of 0.1 (the best among $\{0, 0.1, 0.2, 0.3\}$) to be helpful for Dihedral and Quaternion, while such dropout hurts the training of Dyck and Gridworld. We find EMA to be generally useful, and fix the decay parameter $\gamma = 0.9$ in the experiments since the performance of the EMA model does not seem to be sensitive to the choice of $\gamma \in \{0.85, 0.9, 0.95\}$. Further, increasing the patience of the learning rate scheduler can be helpful.

### B.1.2 VISUALIZING AND INTERPRETING ATTENTION HEADS

Although we defer a fine-grained mechanistic interpretability study (*"which group/semigroup factorizations did these shallow Transformers discover, if any?"*) to future work, we provide some preliminary visualizations of attention heatmaps which strongly corroborate their theoretical counterparts. In particular, consider the gridworld setup in Theorem 3. The theoretical construction consists of two steps: the first attention layer calculates the prefix sum of the actions (i.e. the sum of $\{\tilde{\sigma}_i\}_{i \in [T]} \in \{0, \pm 1\}^T$), and the second attention layer identifies the last time the process is at a boundary state (i.e. 0 or $S$) where the process can be "reset" (i.e. the model can ignore the history before the boundary state and only needs to calculate the sum of subsequent actions).

We have seen in Figure 3 that the network indeed learns to 1) compute the prefix sum, as evidenced by the uniform attention in the first layer, and 2) detect boundary states, as highlighted by large attention scores in the last layer. Figure 7 provides more examples of attention patterns, which are taken from the last layer of a 4-layer GPT-2 model on two randomly selected $\mathrm{Grid}_9$ sequences. We highlight the locations where the process is at a boundary state (white strips for state 0 or gray strips for state $S = 8$), which align well with the highly activated positions of the attention heads, showing that the model learns to locate the closest boundary states. Moreover, when processing tokens appearing later in the sequence than these highly activated positions, no attention weight is put on tokens *before* these positions. This suggests that these highly activated locations reset the state so that the model does not need to look further back past them.

### B.2 SECTION 5: FAILURES OF SHORTCUTS IN MORE CHALLENGING SETTINGS

Our theoretical and main empirical findings have shown that not only do shallow non-recurrent networks subsume deeper finite-state recurrent models in theory, these shallow solutions can also be found empirically via standard gradient-based training. However, experiments in Section 4 and Appendix B are in an idealized setting, with full state supervision during training and in-distribution evaluation at test time. This section studies more challenging settings where these assumptions are relaxed. We consider training under indirect (Section B.2.1) or incomplete (Section B.2.2) state supervision, and evaluation on sequences that is out-of-distribution (Section B.2.3) or of longer lengths (Section B.2.4).

### B.2.1 CHALLENGES FROM INDIRECT SUPERVISION

One type of limited supervision is that the observations may not provide full information of the underlying state. To model this, we consider the case where instead of observing the state $q$ directly, we get a function of the state, denoted $\varphi(q)$, where $\varphi : Q \to \tilde{Q}$ is non-injective (i.e. $|\tilde{Q}| < |Q|$). In each of the experiments involving partially-observable semiautomata, we specify the underlying semiautomaton, as well as the observation function $\varphi$.

- *Dyck language with stack top observations:* For $\mathrm{Dyck}_{n,k}$, the state $Q$ is the state of the stack which takes $\sum_{i=0}^{n} k^i$ values. We take $\varphi$ to be the function that takes in a stack and returns the element at the top of the stack, which is either one of the $k$ open brackets if the stack if non-

---

[22]We use the EMA implementation from https://github.com/fadel/pytorch_ema.

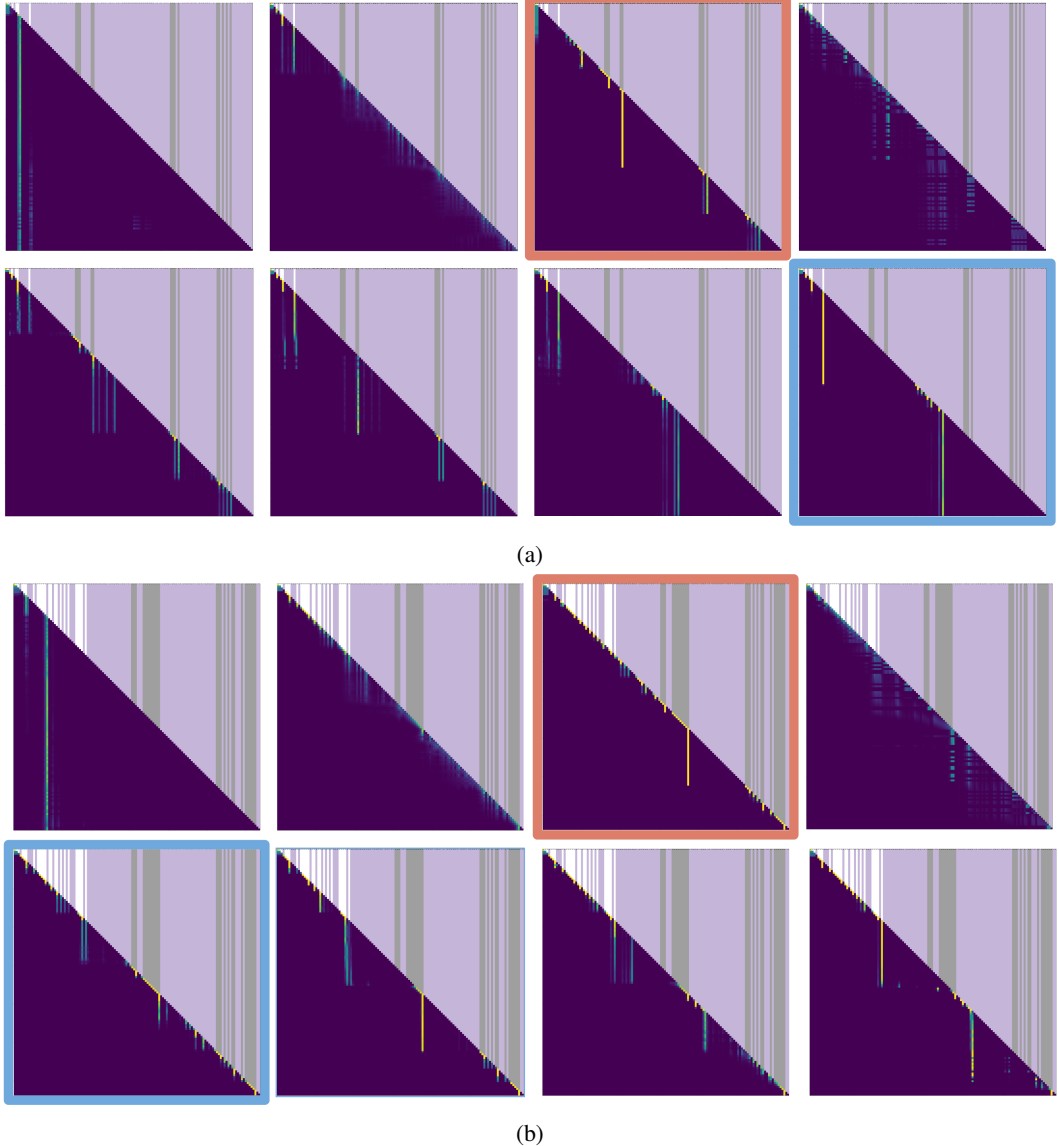

(a)

(b)

Figure 7: Visualization of the entire set of 8 attention heads on two randomly selected length-128 sequences for models trained on $\text{Grid}_9$. The lower triangles visualize the attention patterns, while the upper triangles are ignored by the attention head because of the causal mask. We use the upper triangles to visualize the positions of state 0 and state $S = 8$ in the output: *white strips* mark the position of state 0 and *gray strips* mark state $S = 8$. Example heads that clearly detect state 0 and $S$ are highlighted with blue and red frames, respectively. Note that in many cases, the white/gray strips align with the locations of high attention scores (the bright yellow patterns). This suggests that the model indeed learns to identify the boundary states and that the construction in Theorem 3 agrees with solutions found in practice.

| Task | Dyck$_{4,8}$ | Grid$_9$ | $S_5$ | $C_4$ | $D_8$ | $(\texttt{abab})^\star$, (1) | $(\texttt{abab})^\star$, (2) |
|---|---|---|---|---|---|---|---|
| **Observation** | stack top | $\mathbb{1}_{\text{boundary}}$ | $\pi_{1:t}(1)$ | $\mathbb{1}_{0 \bmod 4}$ | location | accept | accept |
| **Accuracy** | 100.0 | 100.0 | 99.6 | 99.9 | 100.0 | 100.0 | 100.0 |

Figure 8: Accuracies with indirect supervision, extending results in Figure 4(a). The numbers are the maximum over 25 runs. As a reference, LSTM gets 100% on all tasks.

empty, or a special token $\perp$ indicating an empty stack, i.e. $\tilde{Q} := \{1, 2, \cdots, k, \perp\}$. We consider $k = 8$ (as opposed to $k = 2$ in Section 4) to make the prediction task more challenging.

- *Gridworld with boundary observations:* We consider the case where the underlying semiautomaton is Grid$_9$ with $Q = \{0, 1, \cdots, 8\}$. The observation function $\varphi : Q \to \{0, 1\}$ outputs whether the current state is of two boundary states, i.e. at state 0 or state $S = 8$.

- *Permutations with single-element observations:* We take the permutation group $S_5$ with $Q$ is the set of 5! operations. The observation function $\varphi : Q \to \{1, 2, 3, 4, 5\}$ returns the first value of the permutation. For example, $\varphi((2, 1, 4, 3, 5)) = 2$. We use a set of 5 generators for the experiments.

- *Cyclic group with "0 mod 4" observations:* We take $C_4$ as the underlying group with $Q = \{0, 1, 2, 3\}$. The observation function computes whether the current state is state 0, i.e. $\varphi(q) = \mathbb{1}[q = 0]$.

- *Dihedral group with rotation component only:* Recall that $D_{2n} = C_n \rtimes C_2$. We take $n = 4$ with $Q = \{0, 1, 2, 3\} \times \{0, 1\}$, and let the observation function $\psi$ output only the the first component (i.e. $\tilde{Q} = \{0, 1, 2, 3\}$).

- *$(\texttt{abab})^*$:* We consider one semiautomaton which is not featured in Section 4: the one which recognizes the regular expression $(\texttt{abab})^*$, which is also studied in Bhattamishra et al. (2020). The underlying semiautomaton has 5 states: 4 states are in a cyclic fashion when seeing repeated patterns of $abab$, and a fifth absorbing "failure" state is entered if any other pattern is seen. For example, the input sequence $ababababaaabab$ corresponds to states 012301244444, where the $5^{th}$ "a" leads to the absorbing state. The observation function $\varphi$ computes whether the current state is the "accepting" state (i.e. state 3), with $\tilde{Q} = \{0, 1\}$. For example, the output of $\varphi$ for the input sequence $abababab$ is 000100010, and the output for the input sequence $ababbabab$ is 000111111, i.e. the sequence enters the absorbing state at position 5 and never recovers.

  We consider two distributions on the input sequences: (1) the input is always a sequence of the form $abababa\cdots$ (i.e. the process is never in the absorbing state), which is the setup in Bhattamishra et al. (2020); and (2) the input is of the form $abababa\cdots$ with probability 0.5, and is some randomly drawn string of $a, b$ otherwise. Note that case (1) can be solved purely based on the positional encoding, since the label is 1 when the position is a multiple of 4 and 0 otherwise, while case (2) is more difficult since the model needs to take into account the input tokens.

**Results.** We train GPT-2-like models on sequences of length 40. We use 16 layers for $S_5$ and 8 layers for other tasks, with embedding dimension $d = 512$ and $H = 8$ attention heads. As shown in Figure 8, the model is able to achieve near-perfect in-distribution accuracies for all tasks. An interesting side finding is that the choice of positional encoding turns out to be important for both cases of $(abab)^*$: learning is challenging for linear encoding (i.e. $p_i \propto i$) but is easy when using sinusoidal positional encoding, which is likely because the sinusoidal encoding naturally matches the periodicity in $(abab)^*$. In all other experiments, we use sinusoidal positional encodings unless otherwise noted.

### B.2.2 CHALLENGES FROM INCOMPLETE SUPERVISION

Another challenge of limited supervision is that the observation sequence may be incomplete, that is, we may not be able to get supervision on the states at every time step. We consider the task of learning length 100 sequences, where the state at each position is revealed with some probability $p_{\text{reveal}} \in (0, 1]$.

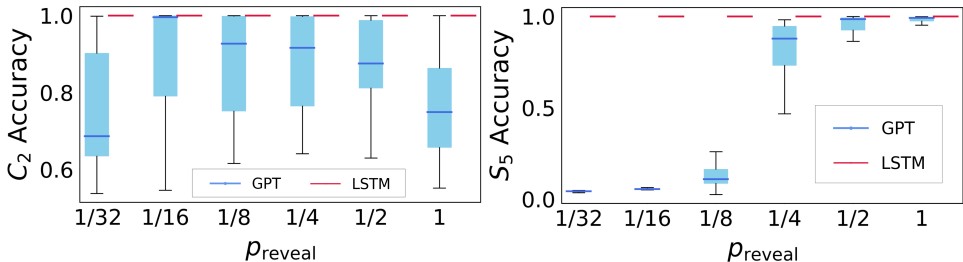

Figure 9: Learning from incomplete state sequences, extending results from Figure 4(b): accuracy vs. position-wise probability of a hidden token (i.e. $p_{\text{reveal}}$), for GPT and LSTM. While LSTM is able to maintain a perfect accuracy across different values of $p_{\text{reveal}}$, GPT's performance may degrade as labels get sparser. The mean and standard deviation are taken over 25 runs.

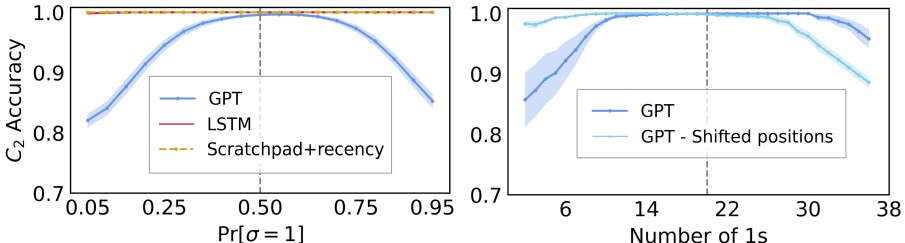

Figure 10: OOD generalization performance on $C_2$: (*Left*) Accuracy on sequences of the same length as training, with varying $Pr[\sigma = 1] = 0.5$. GPT fails at OOD generalization, whereas recurrent solutions implemented by LSTM and Scratchpad with recency bias is robust to different distributions. (*Right*) Accuracy on sequences of the same length as training, with a varying number of 1s in each sequence. GPT has worse performance on counts less frequently seen during training. The lines show the mean accuracy (with shadows showing standard error) over 25 replicates.

**Results.** Figure 9 shows the accuracy against $p_{\text{reveal}}$, for $S_5$ and $C_2$ (i.e. parity). Transformer training pipeline is worse than LSTM at tolerating incomplete supervision: while Transformer is able to maintain the performance across $p_{\text{reveal}}$ for $C_2$, the performance degrades significantly at lower $p_{\text{reveal}}$ for $S_5$. We leave improving the robustness to sparse supervision to future work.

### B.2.3  OUT-OF-DISTRIBUTION GENERALIZATION

The previous subsections show positive results on learning shallow non-recurrent shortcuts with limited supervision during training, either in the form of indirect observations or incomplete observation sequences. In this section, we study challenges at test time, and evaluate Transformers on their *out-of-distribution* generalization performance. For this and the next subsection, the models are trained in the standard way with full state supervision. The training sequences are of length 40, where each position has an equal probability of being 0 or 1, i.e. $\mathbf{Pr}[\sigma = 1] = 0.5$. At test time, the sequences of the same length as training, but the Bernoulli parameter $\mathbf{Pr}[\sigma = 1]$ varies in the range $\{0.05, 0.1, 0.15, \ldots, 0.9, 0.95\}$.

**Negative results for vanilla Transformers.** Figure 10 (*left*) shows the accuracy as $\mathbf{Pr}[\sigma = 1]$ varies. The performance of the Transformer degrades sharply as the test distribution changes away from training, failing at out-of-distribution generalization. Given the theoretical construction of modular counters (Lemma 6), our hypothesis is that Transformer may be learning a shortcut solution that computes the parity by counting the number of 1s, and that counts less frequently seen during training will cause the model to fail. The experimental results agree with the hypothesis: as $\mathbf{Pr}[\sigma = 1]$ deviates from 0.5, it is less likely for the value of the count (which concentrates around $T \times \mathbf{Pr}[\sigma = 1]$) to be seen during training, hence the performance degrades. In contrast, an LSTM recurrent network maintains perfect accuracy when evaluated on all values of $\mathbf{Pr}[\sigma = 1]$.

We further test this hypothesis by checking how the accuracy changes as we vary the count (i.e. the number of 1s) in the input sequence. As shown in Figure 10 *(right)*, Transformer's performance

degrades as the count moves away from the expected number during training, agreeing with the hypothesis. It might appear strange that GPT fails at a lower count more than a higher count. However, this may be because the shortcut learns a correlation between the count and the position: during training, a lower count is more likely to appear early in an input sequence, as opposed to the testing scenario where a lower count is equally likely to appear at a later part of an sequence. This is further supported by the observation that training the model with randomly shifted positions significantly improves the performance at lower counts.

**Guiding the Transformer to learn the recurrent solution.** We investigate one established mitigation for the out-of-distribution brittleness of non-recurrent Transformers: *scratchpad* training and inference. Given a sequence of inputs $(\sigma_1, \ldots, \sigma_T)$ and states $(q_1, \ldots, q_T)$, in the standard (non-recurrent) sequence-to-sequence learning pipeline, the network receives $\sigma_{1:T}$ as input, and outputs the sequence of predictions for $q_t$. In scratchpad training (Nye et al., 2021; Wei et al., 2022), we instead feed the network an interleaved sequence of inputs and states $(\sigma_1, q_1, \sigma_2, q_2, \sigma_3, q_3, \ldots, q_{T-1}, \sigma_T)$ (with an appropriately expanded token vocabulary), and define the network's state predictions to be those at the appropriately aligned positions: $(\hat{q}_1, \perp, \hat{q}_2, \perp, \ldots, \perp, \hat{q}_T)$ (where $\perp$ denotes a position where the prediction is ignored by the loss function). During inference, we iteratively fill in the state predictions. This removes the need for the network to learn long-range dependencies in a single non-recurrent pass, by splitting it into $T$ sequential state prediction problems which can depend on previous predicted state $\hat{q}_{t-1}$; one can think of this as a way to guide a shallow Transformer to learn the recurrent solution (i.e. explicit depth-$\Theta(T)$ iteration of the state transition function), rather than a shortcut.

We note that introducing the scratchpad itself is not sufficient to remove the parallel solution, since the model can simply ignore the scratchpad positions and find the same parallel shortcut as before. The good news is that we can couple scratchpad with an explicit *recency bias* in the attention mechanism (Press et al., 2022) which biases the model towards putting more attention weights on closer input. Intuitively, if the model is only allowed to put attention on the current input token and the current scratchpad (which is simply the current state), then the model is forced to be recurrent; recency bias can be considered as a soft relaxation of the same idea. Combining scratchpad and recency bias, we are able to train a Transformer to learn the recurrent solution, which is resilient to distribution shift; see Figure 10 *(left)*. Notice that this mitigation completely foregoes the computational advantage of a shallow shortcut; we leave it to future work to obtain shortcuts which are resilient to distribution shift. Towards this, the constructions used in the proof of Theorem 1 may be helpful. Finally as a side note, even though the state transitions are Markov, the dependency in the input sequence can still be long range, so we do not expect recency bias to help without scratchpad, since in this case the output can depend uniformly on each input positions (e.g. consider parity).

### B.2.4 LENGTH GENERALIZATION

**Settings for length generalization.** Our final setup is length generalization, where the model is evaluated on sequences of lengths unseen during training. Promoting this difficult desideratum of *length generalization* is an intricate problem in its own right; see Yao et al. (2021); Anil et al. (2022) for more experiments similar to ours and more discussions on length generalization in A.5. In the following, we check the length generalization performance on $\text{Dyck}_{4,2}$ and $C_2$ (with $Pr[\sigma = 1] = 0.5$), where the model is trained on sequences of length 40 and tested on sequences of length $\{8, 16, 24, \cdots, 120, 128\}$.

**Results.** Figure 11 shows the performance on sequences of various lengths. In contrast to LSTM's perfect performance on all scenarios, Transformer's accuracy drops sharply as we move to lengths unseen during training. This is not purely due to unseen values of the positional encoding: randomly shifting the positions during training can cover all the positions seen during testing, which helps improve the length generalization performance but cannot make it perfect; we see similar results for removing positional encodings altogether. However, similar to the OOD setup in the previous subsection, we empirically show that the above flaws are circumventable. Using a combination of *scratchpad* (a.k.a. "chain-of-thought") (Nye et al., 2021; Wei et al., 2022) and recency bias (Press et al., 2022), we demonstrate that Transformers can be guided towards learning recurrent (depth-$T$) solutions, which generalize out-of-distribution and to longer sequence lengths (Figure 11,

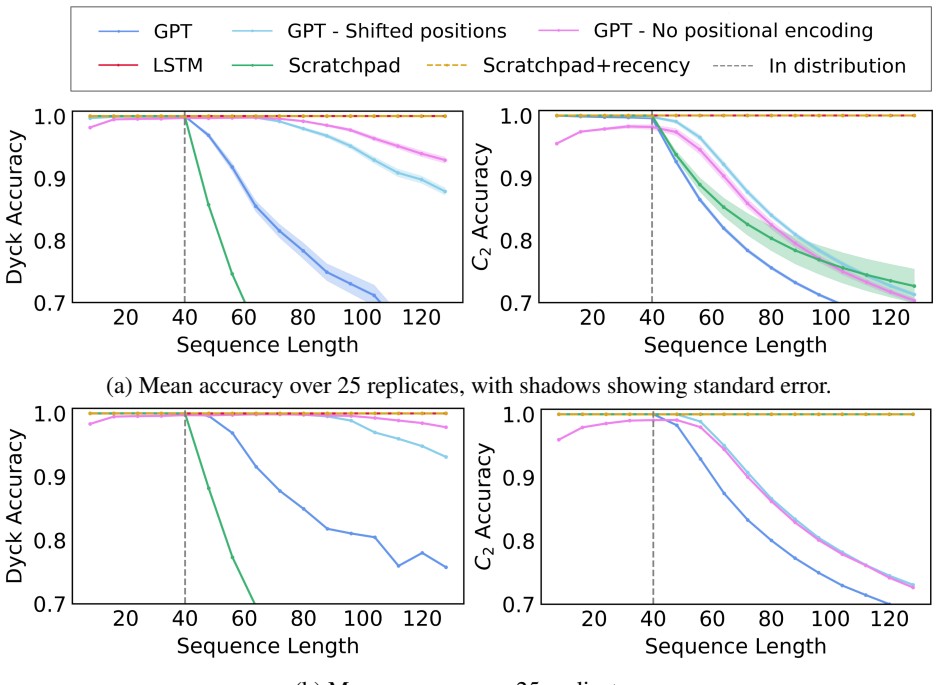

(a) Mean accuracy over 25 replicates, with shadows showing standard error.

(b) Max accuracy over 25 replicates.

Figure 11: Length generalization on Dyck and $C_2$: Transformer fails to generalize, but adding Scratchpad (Nye et al., 2021) and recency bias (Press et al., 2022) serves as a remedy. For "GPT–Shifted positions", the positions in a sequence are shifted by a random number. For "GPT–No positional encoding", no position encodings are provided but the causal mask is still present.

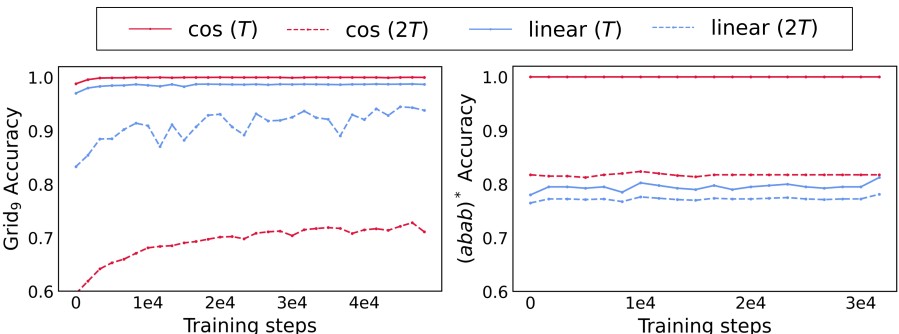

Figure 12: Choice of the positional encoding: while having similar or even superior in-distribution performance (on sequences of length $T = 40$), sinusoidal positional encoding may suffer a larger generalization gap than linear positional encoding when testing on length $2T$. The lines show the mean accuracy over 25 replicates.

yellow curves). The results also confirm that the inclusion of recency bias is necessary: without it, scratchpad training shows no improvement on length generalization.

**Impact of positional encoding.** Figure 11 also shows some interesting findings related to positional encoding, which is believed to be a key component for Transformers and a topic with active research (Ke et al., 2020; Chu et al., 2021). While this work does not aim to improve positional encoding, some of our results may be of interest for future research.

*Sinusoidal vs linear encoding:* We find that the conventional sinusoidal encoding (which is the default for results in this paper) seems to generalize worse to unseen length than linear encoding (where $p_i \propto i$), despite having comparable or better in-distribution performance. Figure 12 shows

examples on $\text{Grid}_9$ and partially observed $(abab)^*$, where we compare the accuracy on freshly drawn samples of the same length as the training sequences (i.e. in-distribution), or of twice the training length. For $\text{Grid}_9$, both positional encodings achieve comparable accuracy, however the sinusoidal encoding performs significantly worse when tested on sequences of doubled length. For partially observed $(abab)^*$ where the label is whether the current string is a multiple of $abab$, the sinusoidal encoding has a clear advantage over the linear encoding on in-distribution performance. However, when tested on sequences of lengths twice as those during training, the performance gap between the two positional encodings shrinks significantly.

*Training with shifted positions:* In general, unseen positions appear to be a major contributor to Transformer's failure of length generalization. This is evidenced by the comparison between Transformer trained with absolute positional encoding, and Transformers trained with random shifts added to the positional encoding: for each batch, we sample a random positive integer in [0,400] and add it to the position indices before calculating the positional encoding; this random integer is the same for each batch and varies across batches. Figure 11 shows that adding such random shifts gives a significant boost to Transformer's length generalization performance, for both Dyck and $C_2$. This suggests that a main challenge to length generalization is the distribution shifts due to positions unseen during training, and finding better positional encoding could be a potential remedy for poor length generalization.

As a side note, we also find that *removing positional encoding altogether* helps improve generalization for both parity and Dyck. For the former, removing positional encodings makes sense since parity is a symmetric function where the ordering of the arguments does not matter, [23] though the positive result for Dyck is less clearly understood. Note that removing positional encoding does not mean having no position information, since the use of the causal mask implicitly encodes the position, which is also noted in Bhattamishra et al. (2020) and concurrent work by Haviv et al. (2022). Understanding this phenomenon is tangential to the current work and is left to future work.

## B.3 ADDITIONAL DETAILS

**Hyperparameters.** For GPT-2 models, we fix the embedding dimension and MLP width to 512 and the number of heads to 8 in all experiments in Section 4, and vary the number of layers from 1 to 16. For LSTM, we fix the embedding dimension to 64, the hidden dimension to 128, and the number of layers to 1. We use the AdamW optimizer (Loshchilov & Hutter, 2017), with learning rate in {3e-5, 1e-4, 3e-4} for GPT-2 or {1e-3, 3e-3} for LSTM, weight decay 1e-4 for GPT-2 or 1e-9 for LSTM, and batch size 16 for GPT-2 or 64 for LSTM. As detailed in Section B.1.1, the models are trained in an online fashion with freshly drawn samples in each batch. The number of freshly drawn samples ranges from 600k to 5000k for different datasets, which is much fewer than the number of possible strings of length 100.

**Implementation details.** Our experiments are implemented with PyTorch (Paszke et al., 2019). The Transformers architectures are taken from the HuggingFace Transformers library (Wolf et al., 2019), using the GPT-2 configuration as a base. The LSTM architecture is the default one provided by the PyTorch library.

**Computational resources.** The experiments were performed on an internal cluster with NVIDIA Tesla P40, P100, V100, and A100 GPUs. For the experiments in Section 4, each training run took up to 10 hours on a single GPU, for a total of $\approx 10^4$ GPU hours. The remaining experiments in Section 5 amount to less than 1% of this expenditure.

---

[23]Empirically, we are able to achieve non-trivial accuracy (even when evaluated at the sequence level) without positional encoding, whereas Bhattamishra et al. (2020) reports 0 accuracy. The discrepancy may be due to different model size: Bhattamishra et al. considers Transformers with up to 4 layers, 4 heads and dimension up to 32, whereas for the parity experiments we consider Transformers with 8 layers, 8 heads, and dimension 512.

## C    PROOFS

### C.1    USEFUL DEFINITIONS AND LEMMAS

**Formal definitions of simulation.**    We first recall the notions of simulation introduced in Section 2:

- A *function* can simulate an automaton for particular choices of $T, q_0$. For a semiautomaton $\mathcal{A} = (Q, \Sigma, \delta)$, a function $f : \Sigma^T \to Q^T$ simulates $\mathcal{A}_{T,q_0}$ if $f(\sigma_{1:T}) = \mathcal{A}_{T,q_0}(\sigma_{1:T})$ for all input sequences $\sigma_{1:T}$. Here, the right-hand side denotes the sequence of states $q_{1:T}$ induced by the input sequence $\sigma_{1:T}$ under the transitions $\delta$ starting from state $q_0$.

- A *function class* can simulate multiple functions associated with a semiautomaton. For a semiautomaton $\mathcal{A} = (Q, \Sigma, \delta)$ and a positive integer $T$, a function class $\mathcal{F}$ (a set of functions $f : \Sigma^T \to Q^T$) *simulates* $\mathcal{A}$ at length $T$ if, for every $q_0 \in Q$, there is function $f_{q_0} \in \mathcal{F}$ which simulates $\mathcal{A}_{T,q_0}$.

Our proofs rely on composing "gadgets" which simulate various substructures of the transformation semigroup $\mathcal{T}(\mathcal{A})$. Thus, it will be useful to establish a third notion of simulation, which works for functions in the embedding space $\mathbb{R}^d$ rather than the symbol spaces $Q, \Sigma$. For clarity, we give this notion a different name (*continuous simulation*):

- For a semiautomaton $\mathcal{A} = (Q, \Sigma, \delta)$, a function $f : \mathbb{R}^d \to \mathbb{R}^d$ *continuously simulates* $\mathcal{A}_{T,q_0}$ if there exist functions $E : \Sigma \to \mathbb{R}^d, W : \mathrm{im} f \to Q$ such that $W \circ f \circ E$ simulates $\mathcal{A}_{T,q_0}$.

When $W$ is a linear threshold function $z \mapsto \arg\max_q [Wz]_q$, this corresponds to a standard classification head. However, our constructions may leverage other encodings of discrete objects.

**Function approximation.**    We provide some simple function approximation results below.

**Lemma 1** (1D discrete function interpolation with an MLP). *Let $\mathcal{X}$ be a finite subset of $\mathbb{R}$, such that $|x| \leq B_x$ for all $x \in \mathcal{X}$, and $|x - x'| \geq \Delta$ for all $x \neq x' \in \mathcal{X}$. Let $f : \mathcal{X} \to \mathbb{R}^d$ be such that $\|f(x)\|_\infty \leq B_y$ for all $x \in \mathcal{X}$. Then, there is a 2-layer ReLU network for which*

$$f_{\mathrm{mlp}}(x + \xi; \theta_{\mathrm{mlp}}) = f(x) \qquad \forall x \in \mathcal{X}, \quad |\xi| \leq \Delta/4.$$

*The inner dimension is $d' = 4|\mathcal{X}|$, and the weights satisfy*

$$\|W_1\|_\infty \leq \frac{4}{\Delta}, \quad \|b_1\|_\infty \leq \frac{4B_x}{\Delta} + 2, \quad \|W_2\|_\infty \leq B_y, \quad b_2 = 0.$$

*Proof.* For each $x_0 \in \mathcal{X}$, we construct an indicator $\psi_{x_0}(x)$ for $x_0$, out of 4 ReLU units. Letting $\Delta' := \Delta/4$, the construction is

$$\psi_{x_0}(x) := \left( \frac{x - (x_0 - 2\Delta')}{\Delta'} \right)_+ - \left( \frac{x - (x_0 - \Delta')}{\Delta'} \right)_+$$
$$- \left( \frac{x - (x_0 + \Delta')}{\Delta'} \right)_+ + \left( \frac{x - (x_0 + 2\Delta')}{\Delta'} \right)_+.$$

The second layer simply sums these indicators, weighted by each $f(x_0)$.    □

**Lemma 2** (General discrete function interpolation with an MLP). *Let $\mathcal{X}$ be a finite subset of $\mathbb{R}^{d_{\mathrm{in}}}$, such that $\|x\|_\infty \leq B_x$ for all $x \in \mathcal{X}$, and $\|x - x'\|_\infty \geq \Delta$ for all $x \neq x' \in \mathcal{X}$. Let $f : \mathcal{X} \to \mathbb{R}^{d_{\mathrm{out}}}$ be such that $\|f(x)\|_\infty \leq B_y$ for all $x \in \mathcal{X}$. Then, there is a 3-layer ReLU network for which*

$$f_{\mathrm{mlp}}(x + \xi; \theta_{\mathrm{mlp}}) = f(x) \qquad \forall x \in \mathcal{X}, \quad |\xi| \leq \Delta/4.$$

*Letting $\mathcal{X}_i$ denote the set of unique values in coordinate $i$, the inner MLP dimensions are as follows:*

$$d_1 = 4 \sum_{i \in [d_{\mathrm{in}}]} |\mathcal{X}_i|, \quad d_2 = |\mathcal{X}|.$$

*The weights satisfy*

$$\|W_1\|_\infty \leq \frac{4}{\Delta}, \quad \|b_1\|_\infty \leq \frac{4B_x}{\Delta} + 2, \quad \|W_2\|_\infty \leq 1, \quad \|b_2\|_\infty \leq d_{\mathrm{in}}, \quad \|W_3\|_\infty \leq B_y, \quad b_3 = 0.$$

*Proof.* The first layer uses the same construction as that in Lemma 1, creating indicators for each $x \in \mathcal{X}_i$ for each $i$. For each $x \in X$, the second layer has an activation which sums the indicators from each $x_i$, with bias $-d_{\text{in}}$ (thus creating indicators for each $x$). The third layer outputs $f(x)$ for each indicator. □

When we apply Lemmas 1 and 2 in recursive constructions, and $B_x/\Delta \geq 1$, we will opt to use the bound $\|b_1\|_\infty \leq 6 B_x/\Delta$, to reduce the clutter of propagating the 2 term without resorting to asymptotic notation.

We also introduce a simpler version of Lemma 1 for the special case of the threshold function $f(x) := \mathbb{1}[x > 0]$:

**Lemma 3** (Threshold with an MLP). *Let $\mathcal{X}$ be a subset of $\mathbb{R}$, and $|x| \geq \Delta$ for all $x \in \mathcal{X}$. Then, there is a 2-layer ReLU network for which*

$$f_{\text{mlp}}(x + \xi; \theta_{\text{mlp}}) = \mathbb{1}[x > 0] \qquad \forall x \in \mathcal{X}, \quad |\xi| \leq \Delta/4.$$

*The inner dimension is $d' = 2$, and the weights satisfy*

$$\|W_1\|_\infty \leq \frac{1}{\Delta}, \quad \|b_1\|_\infty \leq 1/2, \quad \|W_2\|_\infty \leq 1, \quad b_2 = 0.$$

*Proof.* We construct the threshold using 2 ReLU units. The construction is

$$\psi(x) := \left(\frac{x + \Delta}{2\Delta}\right)_+ - \left(\frac{x - \Delta}{2\Delta}\right)_+.$$

□

**Selection via soft attention.** We record some useful lemmas pertaining to approximating hard coordinate selection with soft attention. The following is a simplified version of Lemma B.7 from (Edelman et al., 2022) (which generalizes this to multi-index selection):

**Lemma 4** (Softmax approximates hard max). *Let $z \in \mathbb{R}^T$. Let $\mathsf{softmax}(z) : \mathbb{R}^T \to \mathbb{R}^T$ denote the $T$-dimensional softmax function:*

$$[\mathsf{softmax}(z)]_t := \frac{e^{z_t}}{\sum_{t' \in [T]} e^{z_{t'}}}.$$

*Let $t^* := \arg\max_t z_t$. Suppose that for all $t' \neq t^*$, $z_{t'} \leq z_{t^*} - \gamma$. Then,*

$$\|\mathsf{softmax}(z) - e_{t^*}\|_1 \leq 2T \cdot e^{-\gamma}.$$

*Proof.* Without loss of generality, $\max z = \gamma$ (since the softmax function is invariant under shifting all inputs by the same value), so that all other coordinates are non-positive. Also, assume $T \geq 2$ (the $T = 1$ case is trivial). We have

$$[\mathrm{softmax}(z)]_{t^*} = \frac{e^\gamma}{e^\gamma + \sum_{t \neq t^*} e^t} \geq \frac{e^\gamma}{e^\gamma + T - 1} = 1 - \frac{T-1}{e^\gamma + T - 1} \geq 1 - \frac{T-1}{e^\gamma},$$

and for $t' \neq t$,

$$[\mathrm{softmax}(z)]_{t'} = \frac{e^{t'}}{e^\gamma + \sum_{t \neq t^*} e^t} \leq \frac{1}{e^\gamma}.$$

Thus, the 1-norm of the difference is bounded by

$$\frac{T-1}{e^\gamma} + (T-1) \cdot \frac{1}{e^\gamma} < \frac{2T}{e^\gamma},$$

as claimed. □

**Positional embeddings.** We note the following elementary fact about 2-dimensional circular embeddings.

**Proposition 5** (Circular embeddings). *Consider $p_1, \ldots, p_T$, the $T$ equally-spaced points on the 2-dimensional circle:*

$$[p_t]_1 := \cos\left(\frac{2\pi t}{T}\right), \quad [p_t]_2 := \sin\left(\frac{2\pi t}{T}\right).$$

*Then, for any $t \neq t'$,*

$$|\langle p_t, p_{t'} \rangle| \leq 1 - \frac{2\pi^2}{T^2} < 1 - \frac{19.7}{T^2}.$$

C.2 PROOF OF THEOREM 1: LOGARITHMIC-DEPTH SHORTCUTS VIA PARALLEL PREFIX SUM

In this section, we give the full statement and proof of the universal existence of logarithmic-depth shortcuts.

**Theorem 1** (Simulation is parallelizable). *Let $\mathcal{A} = (Q, \Sigma, \delta)$ be a semiautomaton, $q_0 \in Q$, and $T \geq 1$. Then, there is a depth-$\lceil \log_2 T \rceil$ Transformer which continuously simulates $\mathcal{A}_{T,q_0}$, with embedding dimension $2|Q| + 2$, MLP width $|Q|^2 + |Q|$, and $\infty$-weight norms at most $\max\{4|Q| + 2, 10T\sqrt{\log|Q| + \log T}\}$. It has $H = 2$ heads with embedding dimension $|Q|$ implying $2|Q| + 2$ attention width, and a 3-layer MLP.*

*Proof.* The basic idea is that all prefix compositions $\delta(\cdot, \sigma_t) \circ \ldots \circ \delta(\cdot, \sigma_1)$ can be evaluated in logarithmic depth using a binary tree whose leaves are the per-input transition functions $\delta(\cdot, \sigma) : Q \to Q$. The attention heads select the pairs of functions that need to be composed, while the feedforward networks implement function composition. The network will manipulate functions in terms of their *transition maps*: for example, the encoding of $f := (1 \mapsto 1, 2 \mapsto 1, 3 \mapsto 2)$ is

$$\sum_{q \in \{1,2,3\}} f(q) \cdot e_q = [1\ 1\ 2].$$

**Small nuances.** We will produce a construction for the case where $T$ is a power of 2; general $T$ can be handled via padding. To simplify the construction, we also introduce $T$ padding positions $-(T-1), \ldots, 0$ at the beginning; while this greatly simplifies the positional selection construction, this padding construction could be replaced with a slightly more complicated MLP. Also, in this construction, we do not need to use residual connections; the parallel prefix sum algorithm we use can be executed "in place", saving a logarithmic factor in the width. We do assume access to the 2 positional embeddings at each layer; in the absence of residual connections, the identity function restricted to these 2 dimensions can be implemented by the MLP and attention heads.

Let $L = \log_2 T$ be the depth of the binary tree. We choose $d := 2|Q| + 2$. Instead of indexing the dimensions by $[d]$, we give them names:

- *Left function encoding* dimensions $(q, \mathsf{L})$ for each $q \in Q$.

- *Right function encoding* dimensions $(q, \mathsf{R})$ for each $q \in Q$.

- *Positional encoding* dimensions $\mathsf{P}_1, \mathsf{P}_2$.

Without loss of generality, let $Q = [|Q|] = \{1, \ldots, Q\}$ (selecting an arbitrary enumeration of the state space). Also, assume $|Q| \geq 2$ (if not, add a dummy state). We choose $E(\sigma_t) := \sum_{q \in Q} \delta(q, \sigma_t) \cdot e_{(q,\mathsf{R})}$, mapping each input symbol to the "transition map" of its transitions. At the padding positions $-(T-1), \ldots, 0$, we will encode the "go to $q_0$" function: $\sum_{q \in Q} q_0 \cdot e_{(q,\mathsf{R})}$.

**Function composition gadget.** We first introduce the construction for function composition with a 3-layer ReLU MLP, which will be used by all layers. It gives an exponential improvement over the generic universal function approximation gadget from Lemma 2.

**Lemma 5.** *There exists a 3-layer ReLU MLP $\phi_{\mathrm{mlp}} : \mathbb{R}^d \to \mathbb{R}^d$, with fixed parameters $W_1, b_1, W_2, b_2, W_3$ whose dimensions and weights only depend on $Q$, such that for all $f, g : Q \to Q$, $\phi_{\mathrm{mlp}}$ outputs the transition map of $f \circ g$ given the concatenated transition maps of $f$ and $g$. That is, for all $|Q|^{2|Q|}$ choices of $f, g$:*

$$\phi\left(\sum_{q \in Q} g(q) \cdot e_{(q,\mathsf{L})} + \sum_{q \in Q} f(q) \cdot e_{(q,\mathsf{R})}\right) = \sum_{q \in Q}(f \circ g)(q) \cdot e_{(q,\mathsf{R})}.$$

*The intermediate dimensions are $d_1 = |Q|^2 + |Q|$ and $d_2 = |Q|^2$, and weight norms are bounded by $4|Q| + 2$.*

*Proof.* The first layer uses Lemma 1 to create $|Q|^2$ indicators: one to recognize each value along the $e_{(q,\mathsf{L})}$ direction. Let us index these by $q, q' \in Q$. Then, this gives us $W_1 \in \mathbb{R}^{d \times 4|Q|^2}, b_1 \in \mathbb{R}^{4|Q|^2}, W_2' \in \mathbb{R}^{|Q|^2}$ such that

$$[((z \to W_2'z) \circ \sigma \circ (z \mapsto W_1 z + b_1))(z)]_{q,q'} = \mathbf{1}[e_{(q,\mathsf{L})}^\top z = q'].$$

We also add $Q$ more weights which let the inputs pass through along the $e_{(q,\mathsf{R})}$ directions (add $Q$ more rows $e_{(q,\mathsf{R})}^\top$ to $W_1, W_2'$, calling these indices $\bullet q$ for all $q \in Q$; set biases to 0), for a total of $4|Q|^2 + |Q|$ hidden units and $|Q|^2 + |Q|$ output dimensions of $W_2'$.

The second layer implements multiplication between the indicators and function values. The outputs are again indexed by $q, q' \in Q$. We define $W_2'' \in \mathbb{R}^{|Q|^2 \times (|Q|^2 + |Q|)}$ and $b_2'' \in \mathbb{R}^{|Q|^2}$ to be such that

$$[W_2'']_{(q,q'),(\bar{q},\bar{q}')} := |Q| \cdot \mathbf{1}[(q,q') = (\bar{q},\bar{q}')], \quad [W_2'']_{(q,q'),\bullet\bar{q}} := \mathbf{1}[q' = \bar{q}], \quad [b_2'']_{(q,q')} = -|Q|,$$
$$\forall q, q', \bar{q}, \bar{q}' \in Q.$$

Overall, so far we have

$$[(\sigma \circ (z \mapsto W_2'' W_2' z + b_2'') \circ \sigma \circ (z \mapsto W_1 z + b_1))(z)]_{q,q'} = g(q') \cdot \mathbf{1}[e_{(q,\mathsf{L})}^\top z = q'].$$

The third layer $W_3 \in \mathbb{R}^{d \times |Q|^2}$ simply converts these activations back into an transition map:

$$W_3 = \sum_{q' \in Q} e_{(q,\mathsf{R})} e_{q,q'}^\top.$$

Finally, we note the weight norms:

$$\|W_1\|_\infty \leq 4|Q|, \quad \|b_1\|_\infty \leq 4|Q| + 2, \quad \|W_2'' W_2'\|_\infty \leq 4|Q|, \quad \|b_2''\|_\infty = |Q|, \quad \|W_3\|_\infty = 1.$$

$\square$

**Recursive parallel scan.** The rest of the construction uses a standard parallel algorithm for computing all prefix function compositions: *at layer $l \in [L]$, compose the function at position $t$ with the function at position $t - 2^{l-1}$.* This is a standard algorithm for computing all prefix compositions of associative binary operations with a logarithmic-depth circuit (Hillis & Steele Jr., 1986). We choose the position embeddings to enable implementing these "look-backs" with rotation matrices. For each $t \in \{-T + 1, \ldots, 0, 1, \ldots, T\}$, we use the circle embeddings

$$P_{t,\mathsf{P}_1} := \cos\left(\frac{\pi t}{T}\right), \quad P_{t,\mathsf{P}_2} := \sin\left(\frac{\pi t}{T}\right).$$

In detail, for each $1 \leq l \leq L$:

- Let $\theta := -\frac{\pi 2^{l-1}}{T}, \gamma := 100T^2(\log|Q| + \log T)$.

- Let $H := 2, k := |Q|$. Recall that $|Q| \geq 2$. We will index the heads by superscripts [L], [R].

- Select $W_Q^{[\mathsf{L}]} = W_Q^{[\mathsf{R}]} = W_K^{[\mathsf{R}]} := \sqrt{\gamma} \cdot (e_{\mathsf{P}_1} e_1^\top + e_{\mathsf{P}_2} e_2^\top)$.

- Select $W_K^{[\mathsf{L}]} := \sqrt{\gamma} \cdot (e_{\mathsf{P}_1} e_1^\top + e_{\mathsf{P}_2} e_2^\top) \rho_\theta$, where $\rho_\theta$ is the rotation matrix

$$\begin{bmatrix} \cos(\theta) & \sin(\theta) \\ -\sin(\theta) & \cos(\theta) \end{bmatrix}$$

  in the $e_1, e_2$ basis.

- Select $W_V^{[\mathsf{L}]} = W_V^{[\mathsf{R}]} := \sum_{q \in Q} e_{(q,\mathsf{L})} e_q^\top.$

- Select $W_C^{[\mathsf{L}]} = \sum_{q \in Q} e_q e_{(q,\mathsf{L})}^\top, W_C^{[\mathsf{R}]} = \sum_{q \in Q} e_q e_{(q,\mathsf{R})}^\top.$

At layer $l$, let $\alpha^{[\mathsf{L}]}, \alpha^{[\mathsf{R}]} \in \mathbb{R}^{2T}$ denote the attention mixture weights of the two heads. With this choice of $\gamma$, Lemma 4 and Proposition 5, for each $t \in [T]$, we are guaranteed that $\left\| \alpha^{[\mathsf{R}]} - e_t \right\|_1$ and $\left\| \alpha^{[\mathsf{L}]} - e_{t-2^{l-1}} \right\|_1$ are both at most $\frac{0.1}{|Q| \cdot T}$. Thus, by Hölder's inequality (noting that this mixture is over $T$ vectors of $\infty$-norm at most $|Q|$), this attention layer's output is 0.1-close in the $\infty$-norm to the concatenated transition maps of the functions at positions $t$ and $t - 2^{l-1}$, allowing us to invoke the perturbation-robust function approximation guarantee of Lemma 1 with $\Delta = 1$. For the MLP, we use the function composition gadget.

Thus, at the final layer, the $(q, \mathsf{R})$ dimensions at position $t$ contains the transition map of the prefix composition

$$\delta(\cdot, \sigma_t) \circ \ldots \circ \delta(\cdot, \sigma_1) \circ (q \mapsto q_0).$$

It suffices to choose $W$ to be $z \mapsto e_{(q,\mathsf{R})}^\top z$ for an arbitrary $q$ to read out the sequence of states as scalar outputs in $[|Q|]$. To output a one-hot encoding, an additional MLP (appended to the end of the final layer) would be required. $\qquad\square$

### C.3 PROOF OF THEOREM 2: CONSTANT-DEPTH SHORTCUTS VIA KROHN-RHODES DECOMPOSITION

We begin with the full statement of the theorem:

**Theorem 2** (Transformer Krohn-Rhodes). *Let $\mathcal{A} = (Q, \Sigma, \delta)$ be a solvable semiautomaton (see Definition 6), $q_0 \in Q$, and $T \geq 1$. Then, there is a depth-$O(|Q|^2 \log |Q|)$ Transformer which continuously simulates $\mathcal{A}_{T,q_0}$, with embedding dimension $O(2^{|Q|} |\mathcal{T}(\mathcal{A})|)$, MLP width $|Q|^{O(2^{|Q|})} + O(2^{|Q|} |Q| |\mathcal{T}(\mathcal{A})| T)$, attention width $O(|Q| 2^{|Q|} |\mathcal{T}(\mathcal{A})|)$ heads, and weight norms are bounded by $6|Q| T \log T + 6 \max\{|Q|, |\Sigma|\}$.*

We will begin by presenting self-contained constructions for the two atoms in the Krohn-Rhodes decomposition: a modular counter and a memory unit. In Appendix C.3.2, we will introduce necessary background from Krohn-Rhodes theory (including the definition of a solvable semiautomaton). In Appendix C.3.3 and C.3.4, we will complete the proof of Theorem 2.

#### C.3.1 BASE CASES: MODULAR COUNTING AND MEMORY

**Base case 1: modular addition.** We will start with a construction of a tiny network which lets us simulate any semiautomaton whose transformation semigroup is a cyclic group. Later on, we will use copies of this unit to handle all solvable groups. The construction simply uses attention to perform a flat prefix sum, and an MLP to compute the modular sum.

**Definition 2** (Modular counter semiautomaton). *For any positive integer $n$, define the mod-$n$ modular counter semiautomaton $\mathcal{A} = (Q, \Sigma, \delta)$:*

$$Q := \{0, \ldots, n-1\},$$

$$\Sigma := \{0, \ldots, n-1\},$$

$$\delta(q, \sigma) := (q + \sigma) \bmod n, \quad \forall q \in Q, \sigma \in \Sigma.$$

**Lemma 6** (Simulating a modular counter). *Let $\mathcal{A} = (Q, \Sigma, \delta)$ be the mod-$n$ modular counter semiautomaton. Let $q_0 \in Q$, and $T \geq 1$. Then, there is a depth-1 Transformer which continuously simulates $\mathcal{A}_{T,q_0}$, with embedding dimension 3, width $4nT$, and $\infty$-weight norms at most $4nT + 2$. It has $H = 1$ head with embedding dimension $k = 1$, and a 2-layer ReLU MLP.*

*Proof.* The intuition is simply that the lower triangular matrix causal mask can implement simulation in this cyclic group by performing unweighted prefix sums. The only subtlety is that selecting $W_Q = W_K = 0$ does not quite give us prefix sums: the attention mixture weights at position $t$ are $\frac{1}{t}\sum_{t'\in[t]} e_{t'}$, while we would like the normalizing factor to be uniform across positions ($1/T$ rather than $1/t$). It is possible to undo this normalization using the MLP; however, a particulaly simple solution is to use an additional padding input $\perp$ and 1-dimensional position embeddings to "absorb" a fraction of the attention proportional to $1 - t/T$.

We proceed to formalize this construction, beginning with the input embedding and attention block:

- Select $d := 3, k := 1, H := 1$. Intuitively, the 3 dimensions implement {input/output, padding, position} "channels".

- Select input symbol embeddings $E(\sigma) := \sigma \cdot e_1 \in \mathbb{R}^d$ for each $\sigma \in \Sigma$.

- Include an extra position $\perp$, with embedding $E(\perp) := e_2$ and position encoding $P_{\perp,:} := 0$. Think of this as padding at position 0; it is not masked out by the causal attention mask at any position $t \geq 1$.

- For $t \in [T]$, select $P_{t,:} := \gamma_t e_3$, where $\gamma_t := \log(2T - t)$ is such that $\frac{1}{e^{\gamma_t + t}} = \frac{1}{2T}$.

- Select $W_Q := e_3, W_K := e_2, W_V := e_1, W_C^\top := e_1$.

- We do not need residual connections.

In the output of this attention module, for any input sequence $\sigma_{1:T}$, the 1$^{\text{st}}$ channel of the output at position $t$ is then

$$s := \frac{1}{2T}\sum_{t\in[T]} \sigma_t.$$

where $z_t \in \{0, \ldots, n-1\}$ is such that $\delta(\cdot, \sigma_t) = g^{z_t}$. The MLP simply needs to memorize the function

$$s \cdot e_1 \mapsto (Ts \bmod n) \cdot e_1.$$

We invoke Lemma 1, with $\Delta = \frac{1}{2T}, B_x = \frac{n-1}{2} < \frac{n}{2}, B_y = n$. The number of possible values of $S$ (the cardinality of $\mathcal{X}$ in Lemma 1) is at most $nT$. $\square$

**Base case 2: memory lookups.** It turns out that to simulate *semigroups* instead of groups, the only additional ingredient is a *memory unit*, a semiautomaton for which there are "read" and "write" operations. The minimal example of this is a *flip-flop* (Example 2), a semiautomaton which can sequentially remember and retrieve a single bit $\in \{0, 1\}$, and whose transformation semigroup is the *flip-flop monoid*. It will be convenient to generalize this object to $Q$ states:

**Definition 3** (Memory semiautomaton). For a given state set $Q$, define the *memory* semiautomaton $\mathcal{A} = (Q, \Sigma, \delta)$:

$$\Sigma = Q \cup \{\perp\},$$
$$\delta(q, \sigma) := \sigma \quad \forall q \in Q, \sigma \in \Sigma, \sigma \neq \perp,$$
$$\delta(q, \perp) := q \quad \forall q \in Q.$$

**Lemma 7** (Simulating a memory semiautomaton). *Let $\mathcal{A} = (Q, \Sigma, \delta)$ be the memory semiautomaton. Let $q_0 \in Q$, and $T \geq 1$. Then, there is a depth-1 Transformer which continuously simulates $\mathcal{A}_{T,q_0}$, with embedding dimension 4, width $4|Q|$, and $\infty$-weight norms at most $2T\log(|Q|T)$. It has $H = 1$ head with embedding dimension $k = 2$, and a 2-layer ReLU MLP.*

*Proof.* We start in state $q_0 \in Q$. Our goal is to identify the closest non-$\perp$ token and output the corresponding state. The attention construction is:

- Select $d = 4, k = 2, H = 1$.

- Select input symbol encodings

$$E(\sigma) := (\mathbb{1}[\sigma = \bot]q_0 + \mathbb{1}[\sigma \neq \bot]\sigma_i)e_1 + \mathbb{1}[\sigma = \bot]e_2 + e_4 \in \mathbb{R}^d,$$

  where the first coordinate denotes the action that sets the state[24], the second coordinate denotes whether the input is the no-op action $\bot$, and the fourth coordinate is padding.

- We use positional encoding $P_{t,:} := (t/T) \cdot e_3$.

- $W_Q := [-2e_4 \quad e_4] \in \mathbb{R}^{4 \times 2}$, $W_K := [ce_2 \quad ce_3] \in \mathbb{R}^{4 \times 2}$ for $c = O(T \log(|Q|T))$ as explained below, $W_V := [e_1 \quad 0] \in \mathbb{R}^{4 \times 2}$, and $W_C^\top := [e_1 \quad 0] \in \mathbb{R}^{4 \times 2}$.

The unnormalized attention score computed for position $i$ attending to $j$ is $c(j/T - \mathbb{1}[\sigma_j = \bot])$. Note that the max attention value is achieved at the closest reset action: the unnormalized scored is non-negative if and only if $\sigma_j \neq \bot$, and $j/T$ increases with $j$ ensuring that the closest position is chosen.

Denote this max position as $j_{\max}$. In the setting of hard attention, the output for the $i_{th}$ token after the attention module is $E(\sigma_{j_{\max}})^\top e_1$. In particular, this value is $q_0$ if and only if $\sigma_j = \bot, \forall j \leq i$, i.e. the semiautomaton never leaves the starting state. Otherwise, the value is the value of the nearest non-$\bot$ state (including the current state).

By Lemma 4 (with $\gamma = c/T$), we can approximate hard-attention by soft-attention weights $\alpha \in \mathbb{R}^T$, that is, $\|\alpha - e_{j_{\max}}\|_1 \leq 2T \cdot e^{-c/T}$. This implies, that the output of the attention layer, $\left| \sum_{j \leq i} \alpha_j E(\sigma_j)^\top e_1 - E(\sigma_{j_{\max}})^\top e_1 \right| \leq 2T|Q| \cdot e^{-c/T}$. Then, the MLP can simply round the first coordinate, and we can invoke Lemma 1 with $\Delta = 8T|Q|\exp(-c/T) = 1/2$ (for $c = T \log(16|Q|T)$), $B_x = |Q|$, $B_y = |Q|$ to get weight norm bound $(4 + \log(|Q|T))T \leq 2\log(|Q|T)T$ and width $4|Q|$. $\qquad \square$

### C.3.2 PRIME DECOMPOSITIONS OF GROUPS AND SEMIGROUPS

The key idea behind the proof of Theorem 2 is that all semigroups (and thus, all transformation semigroups of semiautomata) admit a "prime factorization" into elementary components, which turn out to be *simple groups* and copies of the *flip-flop monoid*, which have both been discussed in Appendix A.2. This is somewhat counterintuitive: the only constraint on the algebraic structure of a semigroup is associativity (and indeed, there are many more semigroups than groups), but all of these structures can be built using these two types of "atoms". These components, as well as the *cascade product* under which this notion of "factorization" is defined, are naturally and efficiently implementable by constant-depth self-attention networks.

**The special case of groups.** We begin by discussing the analogous decomposition for *groups*, which generalizes the fact that integers have unique prime factorizations. Let $G$ be a finite group, and let

$$G = H_n \triangleright H_{n-1} \triangleright \cdots \triangleright H_1 \triangleright H_0 = 1$$

be a *composition series*: each $H_i$ is a maximal proper normal subgroup of $H_{i+1}$; 1 denotes the trivial group with 1 element. Then the quotient group $H_{i+1}/H_i$ is called a *composition factor*. The Jordan-Hölder theorem tells us that one can think about the set of composition factors as an invariant of $G$.

**Theorem 6** (Jordan-Hölder). *Any two composition series of $G$ are equivalent: they have the same length $n$, and the sequences of compositions factors $H_{i+1}/H_i$ are equivalent under permutation and isomorphism.*

When each $H_{i+1}/H_i$ is abelian, $G$ is called a *solvable* group. It turns out that each $H_{i+1}/H_i$ is a simple group, so the composition factors of solvable groups can only be cyclic groups of prime order (because every finitely generated abelian group is a direct product of cyclic groups, and, of these, only those of prime order are simple). The smallest non-solvable group is $A_5$, realizable as the

---

[24]Technically $\sigma = \bot$ does not reset the state. We will see that when $q_0$ is selected, it must be that the semiautomaton is always in state $q_0$.

group of even permutations of $5$ elements. As a part of Theorem 2, we will use the composition series to iteratively build neural networks which simulates solvable group operations, requiring intricate constructions to do this with depth independent of the sequence length $T$.

**Adding memory to handle semigroups.** Now, we move on to semigroups. When not all of the input symbols to a semiautomaton induce permutations, we no longer have the group axiom of invertibility (also, if there is no explicit identity symbol, we are not guaranteed to have the monoid axiom of an identity element either). Intuitively, this would seem to induce a much larger family of algebraic structures; an analogy, which is formalizable by representation theory, is that we are now considering a collection of general matrices under multiplication, instead of only invertible ones. The non-invertible transitions *collapse the rank* of the transformations, reducing the set of reachable transformations whenever they are included in an input sequence.

A landmark result of Krohn & Rhodes (1965) tames the seemingly vast and unorderly universe of general finite semigroups. It extends the Jordan-Hölder theorem to the case of semigroups, for a more sophisticated notion of decomposition. Since that work, many variations have arisen, in terms of its precise statement, construction of the decomposition, and proof of correctness. Out of these, an important development is the *holonomy decomposition* method (Zeiger, 1967; Eilenberg, 1974), which forms the basis of our results. We extract the definitions and theorems from Maler & Pnueli (1994), whose exposition emphasizes explicitly tracking the construction of the semiautomaton. We also refer to Maler (2010); Egri-Nagy & Nehaniv (2015); Zimmermann (2020) as recent expositions, containing historical context.

**Definition 4** (Cascade product; cf. (Maler, 2010), Definition 11). Let $n$ be a positive integer. For each $i \in [n]$, let $\mathcal{A}^{(i)} = (Q^{(i)}, \Sigma^{(i)}, \delta^{(i)})$ be a semiautomaton. For $i \in \{2, \ldots, n\}$, let $\phi^{(i)} : Q^{(1)} \times \cdots \times Q^{(i-1)} \times \Sigma \to \Sigma^{(i)}$ denote a *dependency function*. This object $(\{\mathcal{A}^{(i)}\}; \{\phi^{(i)}\})$ is called a *transformation cascade*, and defines a *cascade product semiautomaton* $\mathcal{A} = (Q^{(1)} \times \cdots \times Q^{(n)}, \Sigma^{(1)}, \delta)$ by "feedforward simulation" under the dependency function. We define $\delta$ by the $i$-th component of its output (which we call $\delta^{(\leq i)} : Q^{(1)} \times \cdots \times Q^{(n)} \times \Sigma^{(1)} : Q^{(i)}$):

$$\delta^{(\leq i)}((q^{(1)}, \ldots, q^{(n)}), \sigma) := \delta^{(i)}(q^{(i)}, \sigma^{(i)}),$$

where

$$\sigma^{(i)} := \begin{cases} \sigma & \text{if } i = 1 \\ \phi^{(i)}(q^{(1)}, \ldots, q^{(i-1)}, \sigma) & \text{otherwise} \end{cases}.$$

The corresponding transformation semigroup $\mathcal{T}(\mathcal{A})$ is known as a *cascade product semigroup*.

Intuitively, the cascade specifies a way to compose semiautomata hierarchically: the first layer $i = 1$ maps input sequences to its state sequence, and each internal layer receives an input which depends on the states of all of the preceding layers. Algebraically, the cascade product semigroup is a sub-semigroup of the larger *wreath product* of semigroups (the straightforward analogue of the wreath product of groups, discussed in Section A.2). Although this is useful from an algebraic point of view, we will not use this perspective; the cascade product is a smaller substructure of the wreath product which is sufficient for semiautomaton simulation.

Finally, it will be convenient to define *permutation-reset semiautomata*, which are a useful intermediate step in the Krohn-Rhodes decomposition. To obtain our final result, we will further break these semiautomata down into flip-flops and simple groups.

**Definition 5** (Permutation-reset semiautomaton; cf. (Maler & Pnueli, 1994), Definition 12). A semiautomaton $\mathcal{A} = (Q, \Sigma, \delta)$ is a *permutation-reset* semiautomaton if, for each $\sigma \in \Sigma$, the transition function $\delta(\cdot, \sigma) : Q \to Q$ is either a bijection (i.e. a permutation over the states of $Q$) or constant (i.e. maps every state to some $q(\sigma)$). Associated with each permutation-reset semiautomaton is its *permutation group*, generated by only the bijections.

Now we can state the Krohn-Rhodes theorem, which decomposes *every* finite semiautomaton into a transformation cascade.

**Theorem 7** (Krohn-Rhodes). *Let $\mathcal{A} = (Q, \Sigma, \delta)$ be a semiautomaton. Then, there exists a transformation cascade $\{\mathcal{A}^{(1)}, \ldots, \mathcal{A}^{(n)}; \phi^{(2)}, \ldots, \phi^{(n)}\}$, defining a cascade product semiautomaton $\mathcal{A}'$, such that:*

   (i) *The input symbol space of $\mathcal{A}^{(1)}$ (and thus, that of $\mathcal{A}'$) is $\Sigma$, the same as that of $\mathcal{A}$.*

  (ii) *Letting $Q^{(i)}$ denote the state space of $\mathcal{A}^{(i)}$, there exists a function $\mathcal{W} : Q^{(1)} \times \cdots \times Q^{(n)} \to Q$ such that $\mathcal{W} \circ \mathcal{A}'_{T,q_0}$ simulates $\mathcal{A}_{T,q_0}$ for all $T \geq 1, q_0 \in Q$. For each $i \in [n]$, the transformation semigroup $\mathcal{T}(\mathcal{A}^{(i)})$ is a permutation-reset semiautomaton with at most $|Q|$ states, whose permutation group is a (possibly trivial) subgroup of $\mathcal{T}(\mathcal{A})$ (Maler & Pnueli (1994), Theorem 4).*

  (iii) *The number of semiautomata in the cascade is $n \leq 2^{|Q|}$. Furthermore, the cascade has at most $|Q|$ levels: the indices can be partitioned into at most $L \leq |Q|$ contiguous subsets $N^{(1)} = \{1, \ldots, n_1\}, N^{(2)}, \{n_1 + 1, \ldots, n_1 + n_2\}, \ldots, N^{(L)} = \{n - n_L + 1, \ldots, n\}$ such that $\phi^{(i)}$ only depends on input indices from previous partitions (Maler & Pnueli (1994), Claim 11 & Corollary 12).*

With this decomposition, we are now able to define a *solvable* semiautomaton.

**Definition 6** (Solvable semiautomaton)**.** Let $\mathcal{A} = (Q, \Sigma, \delta)$ be a semiautomaton. We call $\mathcal{A}$ *solvable* if all the permutation groups associated with all of the permutation-reset automata from Theorem 7 are solvable groups.

The remainder of this section will build our construction from the bottom up:

- Appendix C.3.3 will build up from the base case of cyclic groups (Lemma 6), using increasingly sophisticated notions of group products, culminating in a recursive construction which simulates all stages of the Jordan-Hölder composition series. The crucial step is a construction for simulating the semidirect product of groups, given networks which simulate the individual components; this allows us to handle the solvable non-abelian groups.

- Appendix C.3.4 will build networks which simulate permutation-reset semiautomata. A new base case arises: the *memory unit* (Lemma 7), a semiautomaton whose transformation semigroup is a generalization of the flip-flop monoid. Combining the constructions for solvable groups and memory units, we obtain simulators for solvable permutation-reset semiautomata. Finally, the cascade product guaranteed by Krohn-Rhodes (Theorem 7) glues all of these pieces together, giving us the final result.

### C.3.3    SIMULATING SOLVABLE GROUPS

We begin by handling groups. Now, we are ready to specify the recursive constructions which "glue" these components together to form solvable groups. We will proceed in a "bottom-up" order:

   (i) Define a canonical semiautomaton $\mathcal{A}^G$ corresponding to each group $G$ (Definition 7), such that if a network can simulate $\mathcal{A}^G$, it can simulate any other semiautomaton whose transformation semigroup $\mathcal{T}(\mathcal{A}^G)$ is $G$. This lets us talk about simulating *groups*, rather than particular semiautomata. We will show how to turn simulators for groups $N$ and $H$ into simulators for extensions of $N$ by $H$, for increasingly sophisticated extensions, until all cases have been captured.

  (ii) Show how to build the *trivial extension*: given networks which simulate the groups $N$ and $H$, simulate the direct product $G \cong N \times H$, by simply running the individual simulators in parallel (Lemma 8). Combined with Lemma 6, this immediately allows us to simulate arbitrary abelian groups with depth 1, since every abelian group is isomorphic to a direct product of cyclic groups.

  (iii) Show how to build a *split extension*: given networks which simulate a normal subgroup $N$ and quotient $H$, construct a network which simulates any semidirect product $G \cong N \rtimes H$ (Lemma 9). This is the first place where we will require a sequential cascade of layers. It will allow us to handle certain families of non-abelian groups (including $S_3, D_{2n}, A_4, S_4$)

  (iv) Show how to build *arbitrary* extensions (any $G$ which contains $N$ as a normal subgroup, and for which the quotient group $G/N$ is isomorphic to $H$), using the wreath product (Lemma 10), which contains all of the group extensions. The wreath product is itself the semidirect product between a $|H|$-way direct product and $H$, so this can be done in a constant number of layers, by the above. This finally lets us implement any step of a

composition series. In particular, using cyclic groups as a simulable base case, this shows that we can simulate all solvable groups.

**Step (i).** It will be convenient to associate with each group $G$ a canonical "complete" semiautomaton for the class of all semiautomata $\mathcal{A}$ for which $\mathcal{T}(\mathcal{A}) = G$. It is simply the one whose input symbol space $\Sigma$ is every transformation reachable by some sequence of inputs (i.e. every element of $\mathcal{T}(\mathcal{A})$). (For a semigroup, we would also want to adjoin the identity element if it is missing, however, we will only find it useful to define this for groups.)

**Definition 7** (Canonical group semiautomaton; simulating a group). Let $G$ be a finite group. Then, we define the *canonical group semiautomaton* for $G$ as the semiautomaton $(Q, \Sigma, \delta)$ defined by:

- $Q := G$, the set of elements of $G$. Note that if (for example) $G = S_n$, we are setting the state space to be the set of $n!$ permutations, not the ground set $[n]$.

- $\Sigma := G$. That is, we include *all* functions in the input symbol space.

- $\delta(g, h) := h \cdot g$, for all $\forall g \in Q, h \in \Sigma$. (In algebraic terms, we are embedding the $G$ into its *left regular representation*, a.k.a. *left multiplication action*.) Thus, if we take $q_0$ to be the identity element, the sequence of states $q_1, q_2, \ldots, q_T$ corresponds to $q_t = \sigma_t \sigma_{t-1} \ldots \sigma_1$.

- When we simulate the canonical group semiautomaton, we will always choose $q_0$ to be the identity element $e_G$.

A sequence-to-sequence network is said to continuously *simulate* $G$ at length $T$ if it continuously simulates the canonical group semiautomaton of $G$ at length $T$.

**Notation for composable implementations.** Let us furthermore formalize an *implementation* of group simulation. For any finite group $G$, $T \geq 1$, $q_0 \in G$, let $\mathcal{A}^G = (Q, \Sigma, \delta)$ be the canonical semiautomaton for $G$. Then, we summarize a family of concrete implementations of networks which continuously simulate of $\mathcal{A}_{T, e_G}$. We write $\mathsf{sim} : (G, T) \mapsto (E : G \to \mathbb{R}^d, f_{\mathrm{tf}} : \mathbb{R}^{T \times d} \to \mathbb{R}^{T \times d}, W : \mathbb{R}^d \to G)$, where the shape parameters of the output can depend on $G, T$.

To reduce notational clutter, we will access the shape attributes of an implementation via "object-oriented" notation, defining

$$\mathsf{sim}(G, T).\{\mathsf{depth}, \mathsf{dim}, \mathsf{heads}, \mathsf{headDim}, \mathsf{mlpWidth}, \mathsf{normBound}\}$$

to respectively denote the complexity-parameterizing quantities

$$\{L, d, H, k, \max_j\{d_j'\}, B\},$$

defined in Appendix A.4. Also, we will let $\mathsf{sim}(G, T).\{E, \theta, W\}$ respectively denote the encoding layer $E$, network parameters $\theta_{\mathrm{nn}}$, and decoding layer $W$.

**Canonical encodings of group elements.** We also enforce that throughout our constructions of networks which simulate groups, we will maintain that all networks and their submodules manipulate encodings via *integer vectors* in a consecutive range $\{0, \ldots, n - 1\}$. Furthermore, the identity element will always map to the zero vector. We will keep track of the dimensionality of these vectors $\mathsf{sim}(G, T).\mathsf{repDim} \leq d$, and their maximum entries $\mathsf{sim}(G, T).\mathsf{repSize} - 1$. All encoders $E$ and decoders $W$ will map all group elements to and from this kind of representation, and we will choose $W = E^{-1}$. In all, the networks will keep a repDim-dimensional "workspace" of integer vectors, with entries bounded by $\mathsf{repSize} - 1$. When combining groups via the various products constructions, we will combine the components' individual workspaces to create a larger workspace for the product group's elements.

We make some additional remarks on implementations:

- Note that the canonical semiautomaton "forgets" about the semiautomaton abstraction, and never assumes that $G$ is a permutation group on the original state space $Q$ of the semiautomaton we would like to simulate. Indeed, when $N \lhd G$ are permutation groups on $Q$, there is no natural

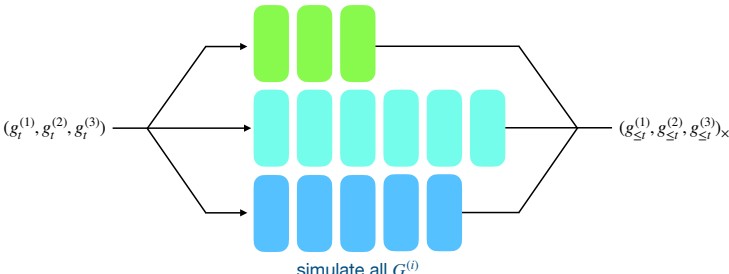

Figure 13: Recursive construction for simulating a direct product of groups $G^{(1)} \times \cdots \times G^{(n)}$. Any number of groups can be simulated in parallel without increasing the depth.

permutation group on $Q$ associated with the quotient $H \cong G/N$; it turns out that will consider simulators for $N$ and $H$.[25]

- To return to solving the simulation problem for some semiautomaton $\mathcal{A} = (Q, \Sigma, \delta)$ whose transformation semigroup is isomorphic to $G$ (at length $T$ and initial state $q_0$), let $\mu : G \to S_Q$ denote this isomorphism. We use $\mathcal{A}^G_{T, e_G}(\sigma_{1:T})$ as the network, with an encoding layer $E \circ \mu^{-1}$, and decoding layer $(\pi \mapsto \pi(q_0)) \circ \mu \circ W$, which can be memorized by an MLP of width $O(|G|)$ via Lemma 2.

- The modular counter semiautomaton, for which we constructed a simulator in Lemma 6, is the canonical group semiautomaton for the corresponding cyclic group $C_n$. Calling this construction $\mathsf{sim}_{C_n}$, we can easily verify that it satisfies the canonical simulator's conditions, and:
  - $\mathsf{sim}_{C_n}.\mathsf{depth} = 1$.
  - $\mathsf{sim}_{C_n}.\mathsf{dim} = 3$.
  - $\mathsf{sim}_{C_n}.\mathsf{heads} = 1$.
  - $\mathsf{sim}_{C_n}.\mathsf{headDim} = 1$.
  - $\mathsf{sim}_{C_n}.\mathsf{mlpWidth} = 4|G| \cdot T$.
  - $\mathsf{sim}_{C_n}.\mathsf{normBound} \leq 4|G| \cdot T + 2 \leq 6|G| \cdot T$.
  - $\mathsf{sim}_{C_n}.\mathsf{repDim} = 1$.
  - $\mathsf{sim}_{C_n}.\mathsf{repSize} = |G|$.

**Step (ii).** As a precursor to the more sophisticated products, we formalize the obvious fact that two non-interacting parallel semiautomata can be simulated without increasing the depth. First, we define the direct product semiautomaton:

**Definition 8** (Direct product of semiautomata). Let $\mathcal{A} = (Q, \Sigma, \delta), \mathcal{A}' = (Q', \Sigma', \delta')$ be two semiautomata. Then, $\mathcal{A} \times \mathcal{A}' = (Q \times Q', \Sigma \cup \{e\} \times \Sigma' \cup \{e\}, \delta \times \delta')$ denotes the natural direct product semiautomaton. Its states are ordered pairs $(q \in Q, q' \in Q')$. Its input symbols are defined similarly, adjoining identity inputs (so that $\delta(q, e) = q, \delta'(q', e) = q'$). The transitions $\delta \times \delta'$ are defined such that

$$(\delta \times \delta')((q, q'), (\sigma, \sigma')) := (\delta(q, \sigma), \delta'(q', \sigma')).$$

Note that under this definition, we have $\mathcal{T}(\mathcal{A} \times \mathcal{A}') = \mathcal{T}(\mathcal{A}) \times \mathcal{T}(\mathcal{A}')$. In particular, for two groups $G, H$, we have $G \times H = \mathcal{T}(\mathcal{A}^G) \times \mathcal{T}(\mathcal{A}^H) = \mathcal{T}(\mathcal{A}^{G \times H}) = G \times H$.

**Lemma 8** (Direct product via parallel simulation). *Let $G^{(1)}, \dots, G^{(n)}$ be a collection of finite groups, and let $T \geq 1$. Suppose each group admits a simulation $\mathsf{sim}_i := \mathsf{sim}(G^{(i)}, T)$. Then, there is a simulation of the direct product group $\mathsf{sim}_\times := \mathsf{sim}(G^{(1)} \times \dots \times G^{(n)}, T)$, whose sizes satisfy:*

---

[25]There is nothing in general preventing quotient groups from being extremely large groups which are not realizable as smaller permutation groups. For concrete examples, see (Kovács & Praeger, 1989). When we ultimately specialize to simulating the composition series of solvable groups, the largest groups we will handle will be the cyclic groups of prime order, so we will in the end be guaranteed that the groups we want to simulate are realizable with $\leq |Q|$ states, but not directly or canonically.

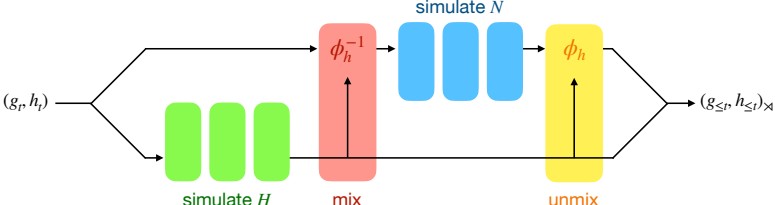

Figure 14: Recursive construction for simulating the semidirect product $N \rtimes H$. The quotient group $H$ is simulated first; these outputs are used to "re-map" the inputs into the simulator for $N$.

- $\circ$ $\mathsf{sim}_\times.\mathsf{depth} = \max_i\{\mathsf{sim}_i.\mathsf{depth}\}$.
- $\circ$ $\mathsf{sim}_\times.\mathsf{dim} = \sum_i\{\mathsf{sim}_i.\mathsf{dim}\}$.
- $\circ$ $\mathsf{sim}_\times.\mathsf{heads} = \sum_i\{\mathsf{sim}_i.\mathsf{heads}\}$.
- $\circ$ $\mathsf{sim}_\times.\mathsf{headDim} = \max_i\{\mathsf{sim}_i.\mathsf{headDim}\}$.
- $\circ$ $\mathsf{sim}_\times.\mathsf{mlpWidth} = \sum_i\{\mathsf{sim}_i.\mathsf{mlpWidth}\}$.
- $\circ$ $\mathsf{sim}_\times.\mathsf{normBound} \leq \max_i\{\mathsf{sim}_i.\mathsf{normBound}\}$.
- $\circ$ $\mathsf{sim}_\times.\mathsf{repDim} = \sum_i\{\mathsf{sim}_i.\mathsf{repDim}\}$.
- $\circ$ $\mathsf{sim}_\times.\mathsf{repSize} = \max_i\{\mathsf{sim}_i.\mathsf{repSize}\}$.

*Proof.* First, we pad all of the individual $\mathsf{sim}_i$ with layers implementing identity (add residual connections, and set attention $W_V$ and all MLP weight matrices to 0), so that all of them have depth $\max_i\{\mathsf{sim}_i.\mathsf{depth}\}$.

Then, the intuition is to construct the direct product semiautomaton by concatenating the "workspaces" of each $G^{(i)}$. In other words, we set the canonical encoding $\mathsf{sim}_\times.E$ of $(g^{(1)}, \ldots, g^{(n)})$ to be the concatenation of each $\mathsf{sim}_i$'s encodings.

The direct product simply lets each $\mathsf{sim}_i$ take inputs and outputs in its individual workspace. To enable this, we need enough parallel dimensions. We set an embedding space of dimension $\sum_i\{\mathsf{sim}_i.\mathsf{dim}\}$ (and similarly within the heads and MLPs), partitioning the coordinates such that in the product construction, each $\mathsf{sim}_i.E$ and $\mathsf{sim}_i.\theta$ only reads and writes to its own dimensions.

This clearly simulates the direct product group. Figure 13 provides a sketch of this construction. $\quad\square$

Note that the direct product construction already allows us to simulate all finite abelian groups in constant depth, since each such group is isomorphic to the direct product of a collection of abelian groups of prime power order.

**Step (iii).** Now, as a harder (and conceptually crucial) case, we show how to simulate a group which is a semidirect product of two groups we already know how to simulate. This encompasses the direct product as a special case, but can now handle some non-abelian groups which admit such decompositions (like the dihedral group $D_{2n}$). The catch is that we will have to simulate these groups using a *sequential* cascade of the individual simulators. This is the key lemma which lets us simulate non-abelian groups:

**Lemma 9** (Semidirect product via 4-stage cascade)**.** *Let $G$ be a finite group which is isomorphic to a semidirect product: $G \cong N \rtimes H$, where $N$ is a normal subgroup of $G$. Let $T \geq 1$. Suppose $N, H$ admit simulations $\mathsf{sim}_N := \mathsf{sim}(N, T), \mathsf{sim}_H := \mathsf{sim}(H, T)$. Then, there is a simulation of $G$, $\mathsf{sim}_\rtimes := \mathsf{sim}(G, T)$, whose sizes satisfy:*

- $\circ$ $\mathsf{sim}_\rtimes.\mathsf{depth} = \mathsf{sim}_N.\mathsf{depth} + \mathsf{sim}_H.\mathsf{depth} + 2$.
- $\circ$ $\mathsf{sim}_\rtimes.\mathsf{dim} = \mathsf{sim}_N.\mathsf{dim} + \mathsf{sim}_H.\mathsf{dim}$.
- $\circ$ $\mathsf{sim}_\rtimes.\mathsf{heads} = \max\{\mathsf{sim}_N.\mathsf{heads}, \mathsf{sim}_H.\mathsf{heads}\}$.
- $\circ$ $\mathsf{sim}_\rtimes.\mathsf{headDim} = \max\{\mathsf{sim}_N.\mathsf{headDim}, \mathsf{sim}_H.\mathsf{headDim}\}$.
- $\circ$ $\mathsf{sim}_\rtimes.\mathsf{mlpWidth} = \max\{\mathsf{sim}_{\{N,H\}}.\mathsf{mlpWidth}, 4|G|\}$.

- ○ $\mathsf{sim}_\rtimes.\mathsf{normBound} \leq \max\{\mathsf{sim}_{\{N,H\}}.\mathsf{normBound}, 6\,\mathsf{sim}_{\{N,H\}}.\mathsf{repSize}, \mathsf{sim}_N.\mathsf{repDim} + \mathsf{sim}_H.\mathsf{repDim}\}$.
- ○ $\mathsf{sim}_\rtimes.\mathsf{repDim} = \mathsf{sim}_N.\mathsf{repDim} + \mathsf{sim}_H.\mathsf{repDim}$.
- ○ $\mathsf{sim}_\rtimes.\mathsf{repSize} = \max\{\mathsf{sim}_N.\mathsf{repSize}, \mathsf{sim}_H.\mathsf{repSize}\}$.

*Proof.* The intuition is as follows, using the dihedral group $D_{2n} \cong C_n \rtimes C_2$ as an example:

- For simplicity, let us think of the "reversible car on a circular world" semiautomaton, whose transformation semigroup is $D_{2n}$. Its state consists of a direction $\in \{+1, -1\}$, and a position $\in \{0, 1, \ldots, n-1\}$. It has two types of inputs: "advance by $i$" (increment the position by $i$ in the current direction, modulo $n$), and "reverse" (flip the sign of the direction). Our simulation task is to track the car's state sequence, given a sequence of inputs (in constant depth, of course).

- It is intuitively clear that we can (and should) compute the sequence corresponding to "direction at time $t$", which is equivalent to simulating the parity semiautomaton.

- We will convert the "advance" moves via a "basis transformation": whenever the current direction is $-1$, an "advance by $i$" should be converted into $-i$. Then, we have reduced the problem to the prefix sum.

**Algorithm.** This intuition essentially shows us how to implement arbitrary semidirect products; we derive the basis change from $\phi$. Before implementing it with Transformer operations, we formalize this "basis transformation". Recall that by the definition of a semidirect product, the elements of $N \rtimes H$ can be written as pairs $(g \in N, h \in H)$, equipped with a homomorphism $\phi : h \to \mathrm{Aut}(N)$ which specifies a multiplication rule:

$$(g, h) \cdot (g', h') := (g\phi_h(g'), hh').$$

Let us write down the properties of $\phi$:

- $\phi$ is a homomorphism. That is, $\phi_{h \cdot h'} = \phi_h(\phi_{h'}(\cdot)) = \phi_h \circ \phi_{h'}$ as permutations on $N$.

- The output of that homomorphism, $\phi_h$, is *also* a homomorphism. That is, $\phi_h(gg') = \phi_h(g) \cdot \phi_h(g')$.

Let us roll out the definition of the semidirect product, given a sequence of inputs $(g_t, h_t)$:

$$(g_2, h_2) \cdot (g_1, h_1) = (g_2 \cdot \phi_{h_2}(g_1), h_2 h_1),$$

$$(g_3, h_3) \cdot (g_2, h_2) \cdot (g_1, h_1) = (g_3 \cdot \phi_{h_3}(g_2 \cdot \phi_{h_2}(g_1)), h_3 h_2 h_1),$$

$$(g_4, h_4) \cdots (g_1, h_1) = (g_4 \cdot \phi_{h_4}(g_3 \cdot \phi_{h_3}(g_2 \cdot \phi_{h_2}(g_1))), h_4 h_3 h_2 h_1).$$

In general, by induction, letting $(g_{\leq t}, h_{\leq t})$ denote $(g_t, h_t) \cdots (g_1, h_1)$, we have

$$g_{\leq t} = g_t \cdot \phi_{h_t}(g_{t-1}) \cdot \phi_{h_t h_{t-1}}(g_{t-2}) \cdots \phi_{h_t \ldots h_3}(g_2) \cdot \phi_{h_t \ldots h_2}(g_1).$$

Applying $\phi_{h_{\leq t}}^{-1}$ on both sides, we notice that

$$\phi_{h_{\leq t}}^{-1}(g_{\leq t}) = \phi_{h_{\leq t}}^{-1}(g_t) \cdot \phi_{h_{\leq t-1}}^{-1}(g_{t-1}) \cdots \phi_{h_{\leq 2}}^{-1}(g_2) \cdot \phi_{h_{\leq 1}}^{-1}(g_1).$$

Thus, it suffices to compute each $h_{\leq t} = h_t h_{t-1} \ldots h_1$, map each $g_t \mapsto \phi_{h_{\leq t}}^{-1}(g_t)$, compute the prefix products in these "coordinates", then invert the mapping to get back $g_{\leq t}$.

**Implementation.** Like before, we partition the embedding dimension in our construction $\mathsf{sim}_\rtimes$ into blocks, one for each component simulator. Let us index the dimensions by the $d_N := \mathsf{sim}_N.\mathsf{dim}$ indices in the "$N$ channel" and analogously for the $d_H$-dimensional "$H$ channel". We choose the canonical encoding $E$ to map elements to their individual channels:

$$E(g, h) = \mathsf{sim}_N.E(g) \textit{ (in the N channel)} + \mathsf{sim}_H.E(h) \textit{ (in the H channel)}.$$

We proceed to specify the construction layer-by-layer. Let $L_{\{N,H\}}$ denote $\mathsf{sim}_{\{N,H\}}.\mathsf{depth}$.

**Layers** 1 **through** $L_H$**: quotient group simulation.** As suggested by the intuitive sketch, we begin with $L_H$ Transformer layers, which are just a copy of $\mathsf{sim}_H.\theta$, reading and writing in the $H$ channel, with a parallel residual layer in the $N$ channel. So far, after these $L_H$ layers, the output at each position $t$ is an integer vector, whose $H$ channel contains $h_{\leq t}$, and whose $N$ channel contains $\mathsf{sim}_N.E(g_t)$.

**Layer** $L_H + 1$**: basis change.** Now, let us add one more "mixing" Transformer layer, whose attention block is identity[26]; we only need a 3-layer MLP block, which represents the function

$$(g \in N, h \in H) \mapsto \phi_h^{-1}(g).$$

To do this, we invoke Lemma 2 (choosing the output to be in the same representation as that used by $\mathsf{sim}_N.E$, in the $N$ channel), with

$$\Delta = 1, d_{\mathrm{in}} = \mathsf{sim}_N.\mathsf{repDim} + \mathsf{sim}_H.\mathsf{repDim},$$

$$B_x = \max\{\mathsf{sim}_N.\mathsf{repSize}, \mathsf{sim}_H.\mathsf{repSize}\}, B_y = \mathsf{sim}_N.\mathsf{repSize},$$

giving us a construction with

$$d_1 \leq 4(|N| + |H|), \quad d_2 \leq |N| \cdot |H|,$$

$$\|W_1\|_\infty \leq 4, \quad \|b_1\|_\infty \leq 6 \max\{\mathsf{sim}_N.\mathsf{repSize}, \mathsf{sim}_H.\mathsf{repSize}\},$$

$$\|W_2\|_\infty \leq 1, \quad \|b_2\|_\infty \leq \mathsf{sim}_N.\mathsf{repDim} + \mathsf{sim}_H.\mathsf{repDim}, \quad \|W_3\|_\infty \leq \mathsf{sim}_N.\mathsf{repSize}.$$

We also add residual connections in the $H$ channel. In summary, after this layer, the output at each position $t$ is an integer vector, whose $H$ channel contains $h_{\leq t}$, and whose $N$ channel contains $\phi_{h_{\leq t}}^{-1}(g_t)$.

**Layers** $L_H + 1$ **through** $L_H + L_N + 1$**: normal group simulation.** The next $L_N$ layers are a copy of $\mathsf{sim}_H.\theta$, with residual connections in the $H$ channel. After these layers, the output at each position $t$ is an integer vector, whose $H$ channel contains $h_{\leq t}$, and whose $N$ channel contains $\phi_{h_{\leq t}}^{-1}(g_{\leq t})$.

**Layer** $L_H + L_N + 2$**: undoing the basis change.** Now, we add one more Transformer layer, whose attention block is identity; we will again use a 3-layer MLP block to represent the inverse of our mapping function

$$(g \in N, h \in H) \mapsto \phi_h(g).$$

This uses Lemma 2, with exactly the same bounds.

At the end of this final "unmixing" layer, the output at each position $t$ is an integer vector, whose $H$ channel contains $h_{\leq t}$, and whose $N$ channel contains $g_{\leq t}$; thus, this is a valid simulation of the semidirect product.

This construction is sketched in Figure 14. $\qquad\qquad\qquad\qquad\qquad\qquad\qquad\qquad\qquad\square$

**Step (iv).** Note that $H \cong G/N$ does *not* imply that $G$ is a semidirect product of $N$ and $H$. Thus, although simulating semidirect products allows us to handle some families of non-abelian groups, this does not yet allow us to handle general solvable groups (i.e. general steps of a composition series, even with a cyclic quotient group). The smallest example is the non-abelian *quaternion group* $Q_8$, the group of unit quaternions under multiplication, which cannot be realized as a semidirect product of subgroups. Instead, we need to appeal to the Krasner–Kaloujnine *universal embedding theorem* (Krasner & Kaloujnine, 1951): a characterization of all of the groups $G$ which are extensions of $N$ by $H$, as subgroups of the wreath product $N \wr H$.

**Lemma 10** (Wreath product via direct and semidirect products). *Let $G$ be a finite group which is isomorphic to a wreath product: $G \cong N \wr H$. Let $T \geq 1$. Suppose $N, H$ admit simulations $\mathsf{sim}_N := \mathsf{sim}(N, T), \mathsf{sim}_H := \mathsf{sim}(H, T)$. Then, there is a simulation of $G$, $\mathsf{sim}_\wr := \mathsf{sim}(G, T)$. In the case where $\mathsf{sim}_\wr.\mathsf{repDim} = 1$, the sizes satisfy:*

---

[26]Even when an attention block simply implements identity, we choose to include it, rather than combining the preceding and subsequent MLPs into a single MLP. This is to ensure that if we compose a number of Transformer layers that depends on $|Q|$, the depth of each MLP is bounded by an absolute constant.

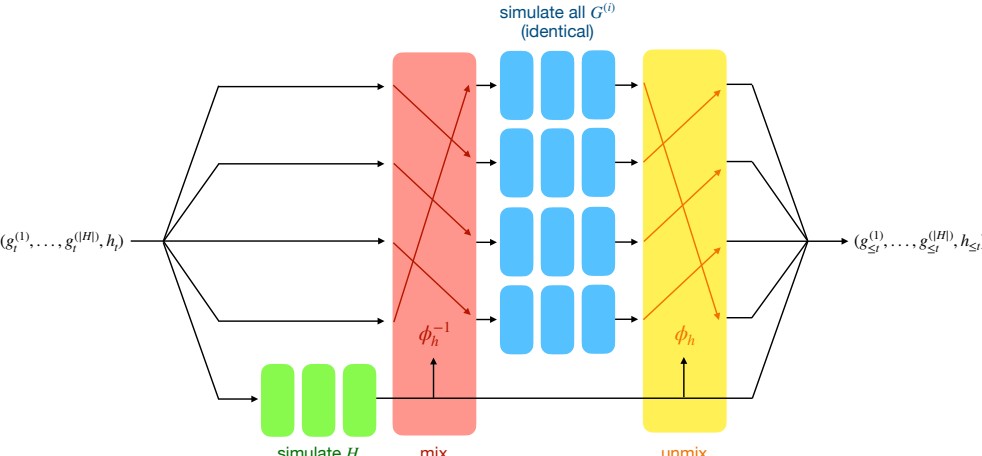

Figure 15: Recursive construction for simulating the wreath product $N \wr H$. One independent copy of $N$ is instantiated for each element of $H$, while the simulator for $H$ permutes them.

- $\circ$ $\mathsf{sim}_\wr.\mathsf{depth} = \mathsf{sim}_N.\mathsf{depth} + \mathsf{sim}_H.\mathsf{depth} + 2$.
- $\circ$ $\mathsf{sim}_\wr.\mathsf{dim} = |H| \cdot \mathsf{sim}_N.\mathsf{dim} + \mathsf{sim}_H.\mathsf{dim}$.
- $\circ$ $\mathsf{sim}_\wr.\mathsf{heads} = \max\{|H| \cdot \mathsf{sim}_N.\mathsf{heads}, \mathsf{sim}_H.\mathsf{heads}\}$.
- $\circ$ $\mathsf{sim}_\wr.\mathsf{headDim} = \max\{\mathsf{sim}_N.\mathsf{headDim}, \mathsf{sim}_H.\mathsf{headDim}\}$.
- $\circ$ $\mathsf{sim}_\wr.\mathsf{mlpWidth} = \max\{|H| \cdot \mathsf{sim}_N.\mathsf{mlpWidth}, \mathsf{sim}_H.\mathsf{mlpWidth}, 5|H|^2|N|\}$.
- $\circ$ $\mathsf{sim}_\wr.\mathsf{normBound} \leq \max\{\mathsf{sim}_{\{N,H\}}.\mathsf{normBound}, 6\,|H|\}$.
- $\circ$ $\mathsf{sim}_\wr.\mathsf{repDim} = |H| \cdot \mathsf{sim}_N.\mathsf{repDim} + 1$.
- $\circ$ $\mathsf{sim}_\wr.\mathsf{repSize} = \max\{\mathsf{sim}_N.\mathsf{repSize}, \mathsf{sim}_H.\mathsf{repSize}\}$.

*Proof.* Even though the wreath product's algebraic structure can be very complex, the construction just requires us to implement its relatively simple description. Applying Lemma 8, we have a network $\mathsf{sim}_\times$ which simulates $N \times \ldots \times N$. Then, we simply apply Lemma 9, using $\mathsf{sim}_H$ to "re-map" inputs to $\mathsf{sim}_\times$ for the normal subgroup. This construction is sketched in Figure 15.

**Concise implementation of reindexing.** We can make one interesting improvement over a generic application of Lemmas 8 and 9: the structure of the mixing function $\phi$, which specifies the semidirect product, is extremely regular. Very fortunately, the structure of $\phi$ allows us to avoid any dependence on the size of the wreath product group ($|N|^{|H|} \cdot |H|$) in the size measures of the implementation. A general automorphism on $N \times \cdots \times N$ is specified by its $|N|^{|H|}$ values. However, in this case, $\phi$ is just a permutation, specified by how each of the $|H|$ channels should switch places. Thus, much like the function composition gadget in Theorem 1, we can construct a simpler MLP than the generic one from Lemma 2.

Specifically, we would like to approximate the function $\phi : H \times (N \times \cdots \times N) \rightarrow (N \times \cdots \times N)$, which simply applies $\pi_h$ to the indices:

$$\phi_h(g^{(1)}, \ldots, g^{(|H|)}) := (g^{(\pi_h(1))}, \ldots, g^{(\pi_h(|H|))}).$$

In the component neural networks' representation space, we need the MLP to implement

$$\left(\mathsf{sim}_N.E(g^{(1)}), \ldots, \mathsf{sim}_N.E(g^{(|H|)}), \mathsf{sim}_H.E(h)\right) \mapsto \left(\mathsf{sim}_N.E(g^{(\pi_h(1))}), \ldots, \mathsf{sim}_N.E(g^{(\pi_h(|H|))})\right),$$

recalling that the elements of $g, h$ are represented by integer vectors with $\infty$-norm at most $\mathsf{sim}_{\{N,H\}}.\mathsf{repBound}$. Notice that when the representation of $|H|$ is a single integer, restricting to any particular coordinate in the representation of an element $g$, this is the same composition problem of function transition maps solved by Lemma 5 in the proof of Theorem 1, which uses its left inputs to permute its right inputs (modulo converting the representations from $\{0, \ldots, |H| - 1\}$ to

$\{1, \ldots, |H|\}$, which we can do by shifting the indicators at the input and final-layer output weights). Thus, $|N| \cdot \mathsf{sim}_N.\mathsf{dim}$ parallel copies of the 3-layer function composition MLP suffice, yielding

$$d_1 = 4|H|^2|N| + |H| \cdot |N| < 5|H|^2|N|, \quad d_2 = |H|^2|N|,$$

$$\|W_1\|_\infty \leq 4|H|, \quad \|b_1\|_\infty \leq 6|H|, \quad \|W_2''W_2'\|_\infty \leq 4|H|, \quad \|b_2''\|_\infty = |H|, \quad \|W_3\|_\infty = 1.$$

When the information about group elements in $H$ is encoded by multiple integers, it is straightforward to extend this construction, by replacing the one-dimensional indicator with the multidimensional indicator from Lemma 2. We will skip the details of this case, since our final results are only about solvable groups; when we want to simulate a general group extension, it will always come from the composition series, so that $H$ is always a cyclic group of prime order. $\qquad \square$

Thus, for general group extensions $G$, we can construct $\mathsf{sim}_\wr$, the wreath product simulator for $N \wr H$, and combine the individual simulators. Note that we can throw away the excess group elements from the simulator: only include in $\mathsf{sim}_\wr.E, \mathsf{sim}_\wr.W$ the group elements which correspond to the subgroup isomorphic to $G$. Then, no part of this construction needs to maintain a width or matrix entry scaling with $|N \wr H|$.

Putting all of this together, we state an intermediate theorem, which is our most general result for groups:

**Theorem 8** (Simulation of solvable groups). *Let $G$ be a solvable group which is isomorphic to a permutation group on $n$ elements. Let $T \geq 1$. Then, there is a Transformer network* $\mathsf{sim} := \mathsf{sim}(G, T)$ *which simulates $G$ at length $T$, for which we have the following size bounds:*

- ○ $\mathsf{sim}.\mathsf{depth} \leq 3\log_2 |G|.$
- ○ $\mathsf{sim}.\mathsf{dim} \leq 2|G|.$
- ○ $\mathsf{sim}.\mathsf{heads} \leq 2|G|.$
- ○ $\mathsf{sim}.\mathsf{headDim} = 1.$
- ○ $\mathsf{sim}.\mathsf{mlpWidth} \leq 20nT|G|.$
- ○ $\mathsf{sim}.\mathsf{normBound} \leq 6nT.$
- ○ $\mathsf{sim}.\mathsf{repDim} \leq 2|G|.$
- ○ $\mathsf{sim}.\mathsf{repSize} \leq n.$

*Proof.* Let

$$G = H_\ell \rhd H_{\ell-1} \rhd \cdots \rhd H_1 \rhd H_0 = 1$$

denote the composition series. Then, because $G$ is solvable, all of the quotient groups $K_i := H_{i+1}/H_i$ are abelian, thus cyclic groups of prime order. Since $G$ is assumed to be a subgroup of $S_n$, none of these primes can be greater than $n$. Thus, every quotient group $K_i$ in the chain satisfies $2 \leq |K_i| \leq n$. Also, note that the length of the composition series $\ell$ is at most $\log_2(|G|)$ (since each inclusion halves the size of the group).

We start with a simulation of $H_1$, which must be a cyclic group, and build the sequence of group extensions recursively until we obtain $G$. In the worst case (in the sense that the implementation size bounds from Lemma 10 are maximized), each step in the composition series must be manifested by a wreath products with $K := C_n$. Recall that we have:

- ○ $\mathsf{sim}_K.\mathsf{depth} = 1.$
- ○ $\mathsf{sim}_K.\mathsf{dim} = 3.$
- ○ $\mathsf{sim}_K.\mathsf{heads} = 1.$
- ○ $\mathsf{sim}_K.\mathsf{headDim} = 1.$
- ○ $\mathsf{sim}_K.\mathsf{mlpWidth} = 4nT.$
- ○ $\mathsf{sim}_K.\mathsf{normBound} \leq 6nT.$
- ○ $\mathsf{sim}_K.\mathsf{repDim} = 1.$
- ○ $\mathsf{sim}_K.\mathsf{repSize} \leq n.$

At each step $i = 0, \ldots, \ell - 1$, Lemma 10, with $H := K_i, N := H_i$, implies:

- $\circ$ $\mathsf{sim}_{H_{i+1}}$.depth $\leq \mathsf{sim}_{K_i}$.depth $+ 3$   (1 more layer to simulate the cyclic group $K_i$, and 2 from the wreath product's mixing operations).
- $\circ$ $\mathsf{sim}_{H_{i+1}}$.dim $\leq |K_i| \cdot \mathsf{sim}_{H_i}$.dim $+ 1$   (noting that all of the components can reuse the same $\perp$ and positional encoding dimensions).
- $\circ$ $\mathsf{sim}_{H_{i+1}}$.heads $\leq |K_i| \cdot \mathsf{sim}_{H_i}$.heads $+ 1$.
- $\circ$ $\mathsf{sim}_{H_{i+1}}$.headDim $\leq \max\{1, 1, \ldots, 1\} = 1$.
- $\circ$ $\mathsf{sim}_{H_{i+1}}$.mlpWidth $\leq \max\{|K_i| \cdot \mathsf{sim}_{H_i}.\mathsf{mlpWidth}, 4nT, 5|K_i|^2 \cdot |H_i|\}$.
- $\circ$ $\mathsf{sim}_{H_{i+1}}$.normBound $\leq \max\{6nT, 6\,|K_i|\}$.
- $\circ$ $\mathsf{sim}_{H_{i+1}}$.repDim $= |K_i| \cdot \mathsf{sim}_{H_i}$.repDim $+ 1$.
- $\circ$ $\mathsf{sim}_{H_{i+1}}$.repSize $\leq n$.

Iterating these recursive inequalities $\ell \leq \lfloor \log_2 T \rfloor$ times gives us the desired bounds. Note that we are using Lagrange's theorem ($\prod_i |K_i| = |G|$), as well as the fact that for positive integers $m_1, \ldots, m_\ell \geq 2$, we have a bound on the series of prefix products: $\sum_i \prod_{j \leq i} m_i \leq 2 \prod_{j \leq \ell} m_i$.

$\square$

### C.3.4 SIMULATING SEMIGROUPS

Now, using this construction and the results developed in the previous section for groups, we complete the construction for semigroups:

- We combine the memory gate construction (Lemma 7) and any network simulating a group to implement the corresponding permutation-reset semiautomaton (Definition 5), the elements of the cascade in Theorem 7.
- To finish, we implement the cascade product (Definition 4) of these permutation-reset semiautomata, guaranteed to exist by Theorem 7. This gives the full result.

First, we summarize the findings of Lemma 7, naming this neural network $\mathsf{sim}_M$ in our "object-oriented" notation. Note that since we are no longer simulating canonical group semiautomata past this point, repDim, repSize are no longer well-defined.

- $\circ$ $\mathsf{sim}_M$.depth $= 1$.
- $\circ$ $\mathsf{sim}_M$.dim $= 4$.
- $\circ$ $\mathsf{sim}_M$.heads $= 1$.
- $\circ$ $\mathsf{sim}_M$.headDim $= 2$.
- $\circ$ $\mathsf{sim}_M$.mlpWidth $= 4|Q|$.
- $\circ$ $\mathsf{sim}_M$.normBound $\leq 2T \log(|Q|\,T)$.

**Lemma 11** (Simulating a permutation-reset semiautomaton). *Let $\mathcal{A} = (Q, \Sigma, \delta)$ be a permutation-reset semiautomaton (see Definition 5), and let $G$ denote its permutation group. Let $T \geq 1, q_0 \in Q$. Let $\mathsf{sim}_G := \mathsf{sim}(G, T)$ be a Transformer network which continuously simulates $G$ at length $T$. Then, there is a Transformer network $\mathsf{sim}'_G$ which continuously simulates $\mathcal{A}_{T, q_0}$, with size bounds:*

- $\circ$ $\mathsf{sim}'_G$.depth $= \mathsf{sim}_G$.depth $+ \mathsf{sim}_M$.depth $+ 1 \leq 3 \log_2 |G| + 2$.
- $\circ$ $\mathsf{sim}'_G$.dim $= \mathsf{sim}_G$.dim $+ \mathsf{sim}_G$.repDim $+ \mathsf{sim}_M$.dim $\leq |G| + |Q| + 4$.
- $\circ$ $\mathsf{sim}'_G$.heads $= \mathsf{sim}_G$.heads $+ \mathsf{sim}_M$.heads $\leq 2|G| + 1$.
- $\circ$ $\mathsf{sim}'_G$.headDim $= \mathsf{sim}_G$.headDim $+ \mathsf{sim}_M$.headDim $+ \mathsf{sim}_G$.repDim $\leq |Q| + 3$.
- $\circ$ $\mathsf{sim}'_G$.mlpWidth $= \mathsf{sim}_G$.mlpWidth $+ \mathsf{sim}_M$.mlpWidth $+ |G|^2|Q| \leq 20nT|G| + 4|Q| + |G|^2|Q|$.
- $\circ$ $\mathsf{sim}'_G$.normBound $\leq \max\{\mathsf{sim}_G.\mathsf{normBound}, \mathsf{sim}_M.\mathsf{normBound}, 6|Q|\} \leq 6|Q|\,T \log T$.

*Proof.* Without loss of generality, we will let $Q := [|Q|]$.

We split the embedding space in our construction into two channels: the $\mathsf{sim}_G$.dim dimensions used by $G$, and a channel consisting of 4 additional dimensions, to be used by a copy of the memory

semiautomaton, whose symbol set is $Q$. Let us call these the $G$ and $M$ channels. For the reset symbols, let $E_M(\sigma)$ denote the 4-dimensional encoding of $\sigma$ from the memory semiautomaton.

Since we defined $G$ to be isomorphic to the permutation group associated with $\mathcal{A}$, there is a bijection $\Phi : G \to S_Q$ between group elements and permutations on $Q$. We choose the embedding $E$ as follows:

$$E(\sigma) := \begin{cases} \mathsf{sim}.E(\Phi^{-1}(\delta(\cdot, \sigma)))\,(G \text{ channel}) \,+\, E_M(\bot)\,(M \text{ channel})\,, & \text{bijections } \delta(\cdot, \sigma) \\ \mathsf{sim}.E(e_G)\,(G \text{ channel}) \,+\, E_M(q_\sigma)\,(M \text{ channel})\,, & \text{resets } \delta(\cdot, \sigma) = q_\sigma \end{cases}.$$

Let $L_G$ denote $\mathsf{sim}_G.\mathsf{depth}$.

**Layers 1 through $L_G$: group simulation.** The first $L_G$ layers are chosen to be a copy of $\mathsf{sim}_G.\theta$ in the $G$ channel, and only residual connections in the $M$ channel. At the end of this, given any inputs $\sigma_{1:T}$ which map via $\Phi^{-1}$ to $g_t$ (letting the group operation be identity when $\sigma_t$ is a reset symbol), the outputs in the $G$ channel will be $d_G := \mathsf{sim}_G.\mathsf{repDim}$-dimensional encodings of the prefix group products $g_{\leq t} = g_t g_{t-1} \cdots g_1$. Now, letting $r(t)$ denote the most recent reset ($\tau \leq t$ such that $\sigma_\tau$ is a reset token), we notice that the state we want can be derived from this sequence:

$$q_t = \Phi(g_t g_{t-1} \cdots g_{r(t)}) q_{\sigma_{r(t)}} = \Phi(g_{\leq t} \cdot g_{\leq r(t)}^{-1})(q_{\sigma_{r(t)}}). \tag{C.2}$$

Here, if there have been no resets up to time $t$, we define $r(t)$ to be 0. We treat $q_0$ like a reset symbol at the beginning of the sequence. Also, note that our canonical group semiautomaton simulator always uses $g_0 = e_G$ as its initial state.

**Layer $L_G + 1$: memory lookup and copy.** To implement the above, at layer $L_G + 1$, we put a copy of the memory semiautomaton in channel $M$, setting its initial state to $q_0$. We will modify this construction slightly, extending $W_V$ with the identity matrix on the $d_G$ group element encoding dimensions of channel $G$. Intuitively, when the memory unit "fetches" the last non-$\bot$ token, we would like it to copy the corresponding $g_{\leq t}$. Note that $\mathsf{sim}_G.\mathsf{repSize}$, the $\infty$-norm bound on the group element encodings, is at most $|Q|$ by Theorem 8, so we do not need to modify the $W_Q, W_K$ norms to increase the attention head's precision. The final modification is that we append to $W_C$ an identity matrix copying the $d_G$ embedding dimensions to a new $d_G$ dimensional channel, which we will call the $I$ (for "invert") channel (set to 0 in the embedding all preceding layers). Then, by Lemma 7, at the end of this layer, at each position $t$, the $M$ channel will contain $q_{\sigma_{r(t)}}$ in dimension 1, and $g_{\leq r(t)}$ in channel $I$. Finally, in channel $G$, we use only residual connections, preserving $g_{\leq t}$ in channel $G$.

**Layer $L_G + 2$: applying $\Phi(gh^{-1})$ pointwise.** This finally allows us to execute Equation C.2 at each position $t$. We use one more Transformer layer, with attention block implementing identity. The MLP memorizes the function $(g, h, q) \mapsto \Phi(gh^{-1})(q) \cdot e_1$ (the coordinate is selected arbitrarily), with the concatenated ($d_{\mathrm{inv}} := 2 \cdot \mathsf{sim}_G.\mathsf{repSize} + 1$)-dimensional encodings on the $(G, I, M)$ channels, whose activations have $\infty$-norms bounded by $|Q|$. We invoke Lemma 2, with parameters

$$\Delta = 1, d_{\mathrm{in}} = d_{\mathrm{inv}}, B_x = |Q|, B_y = |Q|,$$

giving us a construction with

$$d_1 \leq 4 d_{\mathrm{inv}}(|Q| + 1), \quad d_2 \leq |G|^2 |Q|,$$

$$\|W_1\|_\infty \leq 4, \quad \|b_1\|_\infty \leq 6|Q|, \quad \|W_2\|_\infty \leq 1, \quad \|b_2\|_\infty \leq d_{\mathrm{inv}}, \quad \|W_3\|_\infty \leq |Q|.$$

From this output, $W$ simply decodes the correct $q_t$ from dimension 1. $\qquad\square$

**Lemma 12** (Implementing the transformation cascade). *Let $\mathcal{A} = (Q, \Sigma, \delta)$ be a semiautomaton, and let $T \geq 1$. Let $\{\mathcal{A}^{(1)}, \ldots, \mathcal{A}^{(n)}; \phi^{(2)}, \ldots, \phi^{(n)}\}$ be the transformation cascade (Definition 4) which simulates $\mathcal{A}$, as guaranteed by Theorem 7. For each $i$, let $\mathsf{sim}_i$ be a Transformer network which continuously simulates the permutation-reset semiautomaton $\mathcal{A}^{(i)}$ at length $T$. Then, there is a Transformer network $\mathsf{sim}_\mathcal{A}$ which simulates $\mathcal{A}$ at length $T$. Its size bounds are:*

- $\mathsf{sim}_\mathcal{A}.\mathsf{depth} = |Q| \cdot (\max_i \{\mathsf{sim}_i.\mathsf{depth}\} + 1) - 1 \leq 3|Q|^2 \log |Q| + 7|Q|.$
- $\mathsf{sim}_\mathcal{A}.\mathsf{dim} = \sum_{i=1}^n \mathsf{sim}_i.\mathsf{dim} + 1 \leq 2^{|Q|}(|\mathcal{T}(\mathcal{A})| + |Q| + 4) + 1.$

- $\circ$ $\mathsf{sim}_{\mathcal{A}}.\mathsf{heads} = \sum_{i=1}^n \mathsf{sim}_i.\mathsf{heads} \leq 2^{|Q|+1}(|\mathcal{T}(\mathcal{A})| + 1)$.
- $\circ$ $\mathsf{sim}_{\mathcal{A}}.\mathsf{headDim} = \max_{i=1}^n \{\mathsf{sim}_i.\mathsf{headDim}\} \leq |Q| + 3$.
- $\circ$ $\mathsf{sim}_{\mathcal{A}}.\mathsf{mlpWidth} = \sum_{i=1}^n \mathsf{sim}_i.\mathsf{mlpWidth} + 2^{|Q|}|Q|^{2^{|Q|}}|\Sigma| \leq 2^{|Q|}(20|Q|\,|\mathcal{T}(\mathcal{A})|\,T + 4|Q| + |\mathcal{T}(\mathcal{A})|^2|Q| + |Q|^{2^{|Q|}}|\Sigma|)$.
- $\circ$ $\mathsf{sim}_{\mathcal{A}}.\mathsf{normBound} \leq \max_{i=1}^n \{\mathsf{sim}_i.\mathsf{normBound}\} \cup \{2^{|Q|}(|\mathcal{T}(\mathcal{A})| + 5|Q|)\} + 6\max\{|Q|, |\Sigma|\}$ $\leq \max\{6|Q|\,T\log T, 2^{|Q|}(|\mathcal{T}(\mathcal{A})| + 5|Q|)\} + 6\max\{|Q|, |\Sigma|\}$.

*Proof.* At this point, most of the work has been done for us.

We create a separate channel $i$ for each component permutation-reset semiautomaton $\mathcal{A}^{(i)}$. This requires a total of $\sum_{i=1}^n \mathsf{sim}_i.\mathsf{dim}$ embedding dimensions. In addition to these channels, we keep one dimension (with residual connections throughout the network) to represent the input $\sigma_t$. Let $e_\Sigma$ denotes the unit vector along this coordinate. Choosing an arbitrary enumeration to identify $\Sigma$ with $[\Sigma]$, we select the embeddings to be $E(\sigma) := \sigma \cdot e_\Sigma$.

The $L$ layers of $\mathsf{sim}_{\mathcal{A}}$ are divided into $|Q|$ *subnetworks* (which are just Transformer networks), which we will concatenate sequentially at the end. Let these subnetworks be indexed by $\widetilde{\ell} \in \{1, \dots, \widetilde{L}\}$. Each subnetwork starts with a parallel simulation (as in the direct product construction of Lemma 8, padding with layers implementing identity if their depths do not match), combining all of the $\mathsf{sim}_i.\theta$ in the $\ell$-th level of the Krohn-Rhodes decomposition, as defined by Theorem 7. Each $\mathsf{sim}_i.\theta$ is chosen to operate in its own channel $i$. We add residual connections on all of the input/output dimensions in each channel. Then, at the end of each subnetwork except the final one ($1 \leq \widetilde{\ell} \leq \widetilde{L}-1$), we append one more Transformer layer with identity attention block, whose MLP implements the "wiring" specified by $\phi^{(i)}$ from the next level of the decomposition.

Namely, we invoke Lemma 2 with $\Delta = 1, B_x = \max\{|Q|, |\Sigma|\}, B_y = |\Sigma|$, giving us for each pre-final-layer $i$ an MLP which represents the function

$$(\mathsf{sim}_1.W^{-1}(q^{(1)}), \dots, \mathsf{sim}_{i-1}.W^{-1}(q^{(i-1)}), E(\sigma)) \mapsto \mathsf{sim}_i.E(\phi(q^{(1)}, \dots, q^{(i-1)}, \sigma)),$$

where the inputs are stored in the respective $i' < i$ and $\Sigma$ channels, and the output is written to the $i$ channel. Here, the number of input dimensions is

$$d_{\mathrm{in}} = \sum_{i' < i} \mathsf{sim}_{i'}.\mathsf{dim} + 1 \leq 2^{|Q|}(|\mathcal{T}(\mathcal{A})| + |Q| + 4) + 1 \leq 2^{|Q|}(|\mathcal{T}(\mathcal{A})| + 5|Q|).$$

Since the state encodings for each predecessor semiautomaton $i' < i$ are $|Q|$-bounded integer vectors and $\Sigma$ has been assumed to be a $|\Sigma|$-bounded positive integer, it suffices to use $d_1 = 4d_{\mathrm{in}}\max\{|Q|, |\Sigma|\}$. The second hidden layer's width $d_2$ is the number of possible inputs $|\mathcal{X}|$, which is bounded by $|Q|^{2^{|Q|}} \cdot |\Sigma| > d_1$. The weights satisfy

$$\|W_1\|_\infty \leq 4, \quad \|b_1\|_\infty \leq 6\max\{|Q|, |\Sigma|\},$$
$$\|W_2\|_\infty \leq 1, \quad \|b_2\|_\infty \leq d_{\mathrm{in}}, \quad \|W_3\|_\infty \leq |\Sigma|, \quad b_3 = 0.$$

Between $i$ in the same layer, these routing constructions need to be executed in parallel, so this incurs another multiplicative factor in the width, bounded conservatively by $2^{|Q|}$.

The final construction concatenates these blocks, so that at the output of the last layer, every channel $i$ contains a representation of its corresponding component's semiautomaton $Q_i$. The $\mathcal{W}$ guaranteed by Theorem 7 suffices for the overall choice of $W$. $\qquad\square$

## C.4 Proof of Theorem 3: Even Shorter Shortcuts for Gridworld

Recall the gridworld semiautomaton in Example 3, where the state ($Q = \{0, 1, \dots, S\}$) either move to the adjacent state based upon seeing input token $L$ or $R$ (modulo boundary effects), or stay unmoved upon seeing $\perp$. More formally, the transition function is defined as:

$$\delta(q, L) = \max(q - 1, 0)$$
$$\delta(q, R) = \min(q + 1, S)$$
$$\delta(q, \perp) = q.$$

In this section, we will show how to implement gridworld simulation using only 2 Transformer layers. Here we restate the theorem in full generality:

**Theorem 3** (Even shallower shortcuts for gridworld). *For each positive integer $T$, Transformers can simulate the $(S+1)$-state gridworld semiautomaton with 2 attention layers, where the MLP has either (i) depth $O(\log S)$, width $O(T + S)$, or (ii) depth $O(1)$, width $O(T) + 2^{O(S)}$. The weight norms are bounded by $\mathrm{poly}(T)$.*

The depth in (i) can be reduced to $O(S)$ if we allow max pooling, and the dependence on $T$ in the width can be removed with sinusoidal activation. We discuss this in detail after the proof along with generalization to the $k$-dimensional gridworld case.

Note that, in order to find the current state, we need to only know the most recent time at which the semiautomata was at a boundary. It is not immediately obvious how to compute the most recent boundary, if one is not allowed to use the trivial sequential simulation algorithm. Our key insight is that this *boundary detector* can be computed without needing to parse the entire sequence, using the most recent $S + 1$ distinct values of the prefix sums in the sequence.

This algorithm is especially well-suited to the Transformer architecture since: (i) the prefix sum can be computed using one attention layer as in Lemma 6, and (ii) the identification of distinct values can be implemented by a sparse *value-dependent* lookup similar to the memory lookup in Lemma 7 with the help of the *self*-attention (context-dependent retrieval, as opposed to a static lookup), and (iii) the positional weight sharing and causal masking enable all of these computations to be performed in parallel. Overall, Theorem 3 consists of a concise implementation which executes all of these most-recent-boundary detectors in parallel.

In what follows, we first describe the algorithm (Algorithm 1) for computing the state of the semiautomata using the $S + 1$ distinct prefix sum values, and give a proof of its correctness. Subsequently, we formalize the Transformer construction that implements the algorithm. A consolidated list of notations used in the algorithm as well as the proofs is provided in Table 1 for the reader.

### C.4.1 The algorithm solving 1D gridworld

To convey the essence of the full construction, we first provide pseudocode (rather than Transformer weights) for computing the *final* state $q_T$ (rather than the entire state sequence).

We map actions $\sigma \in \{L, R, \bot\}$ to $\widetilde{\sigma} \in \{-1, 1, 0\}$, i.e. $L \mapsto -1$, $R \mapsto 1$, and $\bot \mapsto 0$. Let $\widetilde{\sigma}^{(:)}$ denote the sequence of mapped actions, and let 0 be the initial state. The algorithm (Algorithm 1) has two steps: first, we identify the last time the agent is at a boundary (wall) and the type of the boundary (i.e. state 0 or state $S$). The final state is then simply the sum of all actions in the subsequence, shifted by the last boundary, which is easily computable with 1 attention layer (Lemma 6). Our key insight is that we can identify the boundary using $O(S)$ attention heads in two attention layers, and therefore do not require a recursive computation from the start state (with depth $T$).

To show the correctness of Algorithm 1, it suffices to show that the boundary state is detected correctly, since after that there is no more boundaries and the only step remaining is to calculate the sum of the actions. Let $t_{\mathrm{uniq}}, t_{\mathrm{min}}, t_{\mathrm{max}}$ be as defined in Algorithm 1. Then:

**Lemma 13.** *If $t_{\mathrm{min}} > t_{\mathrm{max}}$, then state at $t_{\mathrm{min}}$ is 0, otherwise state at $t_{\mathrm{max}}$ is $S$.*

*Proof.* The proof follows from two observations:

1. $\min\{t_{\mathrm{min}}, t_{\mathrm{max}}\} = t_{\mathrm{uniq}}$,

2. Suppose $t_{\mathrm{min}} = t_{\mathrm{uniq}}$ (the argument is symmetric for $t_{\mathrm{max}} = t_{\mathrm{uniq}}$). Then $q_{t_{\mathrm{max}}} = S$.

First note that $\widetilde{\sigma} \in \{\pm 1\}$ which implies that the prefix sums increment or decrement by 1 at each index. Therefore, $z_{t_{\mathrm{max}}} - z_{t_{\mathrm{min}}} = S + 1$. This also applies that between (and including) $t_{\mathrm{max}}$ and $t_{\mathrm{min}}$, there must be indices such that they traverse the $S + 1$ distinct values. Since we take the shortest suffix satisfying this, $t_{\mathrm{uniq}} \geq \min\{t_{\mathrm{max}}, t_{\mathrm{min}}\}$. This proves Observation 1.

Assume $t_{\mathrm{min}} = t_{\mathrm{uniq}}$. We can break the analysis into the following 2 cases:

(a) *The $S + 1$ distinct values correspond to $S + 1$ distinct states (covering both boundaries).* This implies that the minimum and maximum out of these distinct prefix sums must correspond to the boundaries, that is, $q_{t_{\mathrm{max}}} = S$ and $q_{t_{\mathrm{min}}} = 0$.

| Notation | Definitions |
|---|---|
| $\sigma$ | An input token; $\sigma \in \{L, R, \bot\}$. |
| $\widetilde{\sigma}$ | A mapped input token; $\widetilde{\sigma} = -1$ if $\sigma = L$, $\widetilde{\sigma} = 1$ if $\sigma = R$, and $\widetilde{\sigma} = 0$ if $\sigma = \bot$. |

***Used in Algorithm 1:***

| | |
|---|---|
| $q_t$ | The state at position $t \in [T]$; $q_t \in \{0, 1, \cdots, S\}$. |
| $\boldsymbol{z} \in \mathbb{Z}^T$ | Prefix sums at all $T$ positions. |
| $t_{\text{uniq}}$ | The most recent position for which the prefix sums $\boldsymbol{z}_{t_{\text{uniq}}:T}$ contain $S + 1$ unique values. |
| $t_{\max}, t_{\min}$ | The positions corresponding to the max/min prefix sum among positions $t_{\text{uniq}}, \ldots, T$. |
| $t_{\text{final}}$ | The position of the last boundary state, defined as $t_{\text{final}} := \max\{t_{\max}, t_{\min}\}$. |

***Used in the Transformer construction:***

| | |
|---|---|
| $\gamma_t$ | The positional encoding for position $t \in [T]$, defined as $\gamma_t := \log(2T - t)$. |
| $x_{\text{attn}}^{(1)}[t]$ | The output of the first layer attention at position $t \in [T]$, defined as $x_{\text{attn}}^{(1)}[t] := \frac{1}{2T} \sum_{i \in [t]} s_i$. |
| $x_{\text{mlp}}^{(1)}[t]$ | The output of the first layer MLP at position $t \in [T]$, defined as $x_{\text{mlp}}^{(1)}[t] := [x_{\text{attn}}^{(1)}[t], \gamma_t, 1, \cos(x_{\text{attn}}^{(1)}[t]\pi), \sin(x_{\text{attn}}^{(1)}[t]\pi)]$. |
| $j_{\max}^{(s)}$ | The position which achieves the max attention score for the $s^{th}$ head at time $t \in [T]$ ($t$ is omitted for notational convenience), for $s \in [0, 1, \cdots, 2S]$. |
| $x_{\text{attn}}^{(2)}[t]$ | The output of the second layer attention at position $t \in [T]$, defined as $x_{\text{attn}}^{(2)} := [\gamma_{j_{\max}^{(0)}}, \gamma_{j_{\max}^{(1)}}, \cdots, \gamma_{j_{\max}^{(2S)}}]$. |
| $x_{\text{mlp}}^{(2)}[t]$ | The output of the second layer MLP at position $t \in [T]$, which gives the state at $t$. |

Table 1: Notations for the proof of Theorem 3.

(b) *The $S + 1$ distinct values correspond to fewer than $S + 1$ distinct states*. This implies that only one of the two boundaries is visited in the sequence starting from $t_{\text{uniq}}$. In order to get $S + 1$ distinct values, it must be that this boundary wall is hit, i.e., the sequence tries to make a move that the boundary blocks. If the sequence does not hit a boundary, then at every time the same state is revisited, the prefix sum must be the same, and we will not be able to get $S + 1$ distinct values. Since $t_{\min} = t_{\text{uniq}}$, we claim that the visited boundary must be $S$. Suppose this is not true, then the boundary visited is 0. This implies that $q_{t_{\min}} = 0$. Since the sequence does not hit $S$, at any position it is at state 0 before hitting the wall, the value will be $z_{t_{\min}}$. Thus, when the sequence first hits the wall at 0 (say index $\tau$), then $z_\tau = z_{t_{\min}} - 1$ which is not possible by definition of $t_{\min}$. Thus, the boundary must be $S$.

$\square$

Given the above, our algorithm identifies the boundary correctly and then can just use the prefix sum to evaluate the current state.

### C.4.2 TRANSFORMER CONSTRUCTION FOR ALGORITHM 1

In this section we will show how to simulate Algorithm 1 using a 2-layer Transformer with $2S$ attention heads.

*Proof of Theorem 3.* In our construction, the first attention layer will compute the prefix sums. This can be mapped to a cyclic group from Lemma 6, however for completeness, we will restate the main construction. The MLP in the first layer will map this prefix sum to a circular embedding

---

**Algorithm 1:** 1D gridworld: computing the final state

---

**Data:** $\widetilde{\sigma} \in \{\pm 1\}^T$, $S \in \mathbb{Z}^+$

**Result:** Final state $y_T \in \{0, 1, \cdots, S\}$

// Pad tokens so there are at least $S + 1$ distinct values

$\widetilde{\sigma} \leftarrow [\underbrace{-1, -1, \cdots, -1}_{S+1}, \widetilde{\sigma}]$

// Calculate the prefix sum for each index

$z \leftarrow \mathsf{prefix\_sum}(\widetilde{\sigma})$    (i.e. $z_t \leftarrow \sum_{\tau=1}^{t} \widetilde{\sigma}_\tau$)

// Find a substring containing $S + 1$ unique values

$t_{\mathrm{uniq}} \leftarrow \max\{t : |\mathsf{set}(z_{t:})| = S + 1\}$

// Find positions of the last max and min values

$t_{\min} \leftarrow \max\{t : z_t = \min_{\tau \geq t_{\mathrm{uniq}}} z_\tau\}$

$t_{\max} \leftarrow \max\{t : z_t = \max_{\tau \geq t_{\mathrm{uniq}}} z_\tau\}$

// Identify the type of boundary

**if** $t_{\min} > t_{\max}$ **then**
  | boundary $\leftarrow 0$
**else**
  | boundary $\leftarrow S$
**end**

// The final state is the sum of the substring after the last boundary

$t_{\mathrm{final}} \leftarrow \max\{t_{\min}, t_{\max}\}$

$y_T = z_T - z_{t_{\mathrm{final}}} + \text{boundary}$

---

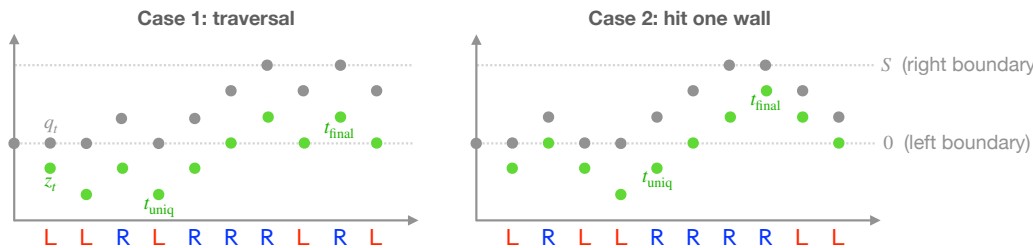

Figure 16: Illustrated examples for 4-state gridworld ($Q = \{0, 1, 2, 3\}$). The algorithms compute the prefix sums $z_t$ (as if there were no boundaries; shown as green dots); intuitively, it might seem like they have no direct relationship with the state sequence $q_t$ (gray dots). However, defining $t_{\mathrm{uniq}}$ as the start of the shortest suffix containing $S + 1$ distinct prefix sums, and $t_{\mathrm{final}}$ as the most recent minimum or maximum point within this suffix, our case analysis shows that $q_{t_{\mathrm{final}}}$ is always a boundary, resulting in a parallel simulation algorithm.

(see Proposition 5). The second layer attention will use the circular embedding structure to find $S + 1$ closest distinct values to the current value $z_t$ (suppose we are considering position $t \in [T]$) by identifying the positions for closest values in the set $\{z_t - S, z_t - S + 1, \ldots, z_t - 1, z_t + 1, z_t + 2, \ldots z_t + S\}$, i.e. $S$ closest distinct values smaller than $z_t$, and $S$ values larger than $z_t$. This closest distinct value construction can be viewed as a position dependent flip-flop monoid construction, where we need to identify the closest position with a particular action. Note that this set of values would contain the distinct $S + 1$ values needed by the Algorithm 1, hence the second layer MLP can implement the state computation using these values.

**Input representation:** We select input symbol embedding $E(\sigma) = \widetilde{\sigma} \cdot e_1 \in \mathbb{R}^d$ where $\widetilde{\sigma}$ is the action corresponding to $\sigma$, that is, $s = \begin{cases} -1 & \text{if } \sigma = L \\ 1 & \text{if } \sigma = R \\ 0 & \text{otherwise} \end{cases}$. We will use positional encoding

$P_{t,:} := \gamma_t e_3$, where $\gamma_t := \log(2T - t)$ is such that $\frac{1}{e^{\gamma_t + t}} = \frac{1}{2T}$. We will include an extra position $\perp$, with embedding $E(\perp) := e_2$ and position encoding $P_{\perp,:} := 0$. Think of this as padding at position 0; it is not masked by the causal attention mask.

**Prefix sum (Layer 1 attention):**    The attention construction for the first layer, in full detail:

- We select $d := 4, k := 1, H := 1$.

- Select $W_Q := e_3, W_K := e_2, W_V := e_1, W_C^\top := e_4$.

With this attention module, the $4^{\text{th}}$ channel of the output at position $t$ is $x_{\text{attn}}^{(1)}[t] = \frac{1}{2T} \sum_{i \in [t]} s_i$, which is the prefix sum scaled down by $1/2T$.

**Circular embedding (MLP 1):**    The first MLP maps $x_{\text{attn}}^{(1)}[i] \mapsto [\cos(x_{\text{attn}}^{(1)}[i]\pi), \sin(x_{\text{attn}}^{(1)}[i]\pi)]$, where $\cos, \sin$ are calculated up to $O(\log T)$ precision using the construction in Lemma 1 with width $4(2T + 1)$. [27] and weight norms at most $8T$. Together with the input using the skip connection (for $\gamma_i$) we get $x_{\text{mlp}}^{(1)}[t] := \left[ x_{\text{attn}}^{(1)}[t], \gamma_t, 1, \cos(x_{\text{attn}}^{(1)}[t]\pi), \sin(x_{\text{attn}}^{(1)}[t]\pi) \right]$ as the embedding to be input to the second attention layer.

**Finding closest $S + 1$ values (Layer 2 attention):**    Our goal is to find the shortest subsequence (looking back from the current position $t$) that contains $S + 1$ distinct values for $x_{\text{attn}}^{(1)}$; that is, we want to find the max $\tau \le t - S$ such that $\left| \{x_{\text{attn}}^{(1)}[i]\}_{i=\tau}^t \right| = S + 1$. We will do this by using $2S + 1$ heads such that $\forall s \in \{0, 1, \cdots, 2S\}$, the attention score for the $s_{th}$ head on position $i \in [T]$ satisfies

$$\tilde{\alpha}_{t,i}^{(s)} := \left\langle (W_Q^{(s)})^\top x_{\text{mlp}}^{(1)}[t], (W_K^{(s)})^\top x_{\text{mlp}}^{(1)}[i] \right\rangle$$

$$\begin{cases} = 1 - c \log(2T - i), & \text{if } x_{\text{attn}}^{(1)}[i] = x_{\text{attn}}^{(1)}[t] + \frac{s-S}{2T}, \\ \le 1 - c \log(2T - i) - \frac{\pi^2}{8T^2}, & \text{otherwise,} \end{cases}$$

where $c = \frac{\pi^2}{(16 \log 2)T^2}$. That is, for any $i, j$ s.t. $x_{\text{attn}}^{(1)}[i] = x_{\text{attn}}^{(1)}[t] + \frac{s-S}{2T}$ (matched) and $x_{\text{attn}}^{(1)}[j] \ne x_{\text{attn}}^{(1)}[t] + \frac{s-S}{2T}$ (unmatched), the difference in the unnormalized attention weights is lower bounded by $\tilde{\alpha}_{t,i}^{(s)} - \tilde{\alpha}_{t,j}^{(s)} \ge \frac{\pi^2}{16T^2}$. This can be achieved by letting $W_Q^{(s)} := \begin{bmatrix} \begin{bmatrix} 0 & 0 & 0 \\ 0 & 0 & -c \\ 0 & 0 & 0 \end{bmatrix} & 0 \\ 0 & \rho_{\theta(s)} \end{bmatrix} \in$ $\mathbb{R}^{5 \times 5}$ where $\rho_{\theta(s)}$ the rotation matrix of angle $\theta^{(s)} := \frac{(s-S)\pi}{2T}$, such that $(W_Q^{(s)})^\top x_{\text{mlp}}^{(1)}[t] = \left[ 0, -c, 0, \cos\left(\left(x_{\text{attn}}^{(1)}[t] + \frac{s-S}{2T}\right)\pi\right), \sin\left(\left(x_{\text{attn}}^{(1)}[t] + \frac{s-S}{2T}\right)\pi\right) \right]$. $W_K^{(s)}, W_C^{(s)}$ are simply the $5 \times 5$ identity matrix, and $W_V^{(s)} = e_1 e_1^\top + e_2 e_2^\top$.

Let $j_{\max}^{(s)}$ denote the position that achieves the max attention score for the $s^{th}$ head, then the output of the $s^{th}$ head [28] is $[x_{\text{attn}}^{(1)}[j_{\max}^{(s)}], \gamma_{j_{\max}^{(s)}}, 0, 0, 0]$. We can ignore the last three coordinates (which are 0) as well as $x_{\text{attn}}^{(1)}[j_{\max}^{(s)}]$, since we will only need the difference $x_{\text{attn}}^{(1)}[t] - x_{\text{attn}}^{(1)}[j_{\max}^{(s)}]$ which is $\frac{s-S}{2T}$ by definition. We then concatenate the outputs from all $(2S + 1)$ heads in a $(2S + 1)$-dimensional vector $x_{\text{attn}}^{(2)} = [\gamma_{j_{\max}^{(0)}}, \gamma_{j_{\max}^{(1)}}, \cdots, \gamma_{j_{\max}^{(2S)}}]$ as the input to the second layer MLP.

Intuitively, each head in the second attention layer is trying to identify the set of positions for which the prefix sums match a particular value specified by the head. Each head selects the last matching position if such positions exist, and selects $t$ if not. The following observation will be helpful for

---

[27] The width is 1 if we allow sinusoidal activation instead of relu; see the discussion after the proof.

[28] We assume hard attention here for ease of exposition of the proof; soft attention can be handled with Lemma 4 and Lemma 1 as in our previous constructions. This requires norm $\text{poly}(T)$ at max.

our subsequent MLP construction: the values of coordinates of $x_{\text{attn}}^{(2)}$ increase on both sides of the $S^{th}$ heads; that is, $x_{\text{attn}}^{(2)}$ satisfies the following:

**Lemma 14.** *There exist $a < b \in \{0, 1, \ldots, 2S\}$ such that*

$$x_{\text{attn}}^{(2)}[a] > x_{\text{attn}}^{(2)}[a+1] > \ldots > x_{\text{attn}}^{(2)}[S] < x_{\text{attn}}^{(2)}[S+1] < \ldots < x_{\text{attn}}^{(2)}[b]$$

*and all $s \in \{0, 1, \ldots, 2S\} \setminus \{a, a+1, \ldots, b\}$ we have $x_{\text{attn}}^{(2)}[s] = \log(2T - t)$.*

*Proof.* Note that $x_{\text{attn}}^{(2)}[s] := \log(2T - \gamma_{j_{\max}^{(s)}})$ which makes the ordering inverse of the position. Observe that the unmatched indices correspond to values that have not been reached. This can only happen for values on either the leftmost or the rightmost coordinates, since the prefix sums are continuous on integers. Now let's prove that the value will be decreasing moving away from index $S$ in both directions. Suppose this was not true, there indeed was $s \leq S$ such that $x_{\text{attn}}^{(2)}[s] \geq x_{\text{attn}}^{(2)}[s-1]$ ($s \geq S$ case is identical), then it implies that the closest index that achieved relative value $S - s$ is further away from $t$ than $S - s + 1$. However, since the moves can update the prefix sum by magnitude at most 1, then to get to relative value 0 from relative value $S - s + 1$, we would need to have crossed relative value $S - s$. This implies that there is another position closer to $t$ with this value, contradicting our assumption. This proves the result. $\square$

**Computing state (MLP 2):** To compute state from the positional information given by $x_{\text{attn}}^{(2)}$, we need to do the following computations:

- *Step 1*: consider $S + 1$ windows of size-$(S+1)$, each containing the $s^{th}$ to $(s+S)^{th}$ heads for $s \in \{0, 1, \ldots, S\}$, and identify the window that contains positions closest to the end (this would correspond to the closest $S + 1$ distinct values to $x_{\text{att}}^{(1)}[t]$);

- *Step 2*: identify the boundary state in the selected window by comparing the indices of the endpoints of the window;

- *Step 3*: output the final state based on the position of the boundary states and its value relative to the current position $t$.

We will show two constructions for implementing this, one of which will use $O(1)$ depth and $2^{O(S)}$ width, and the other will use $O(\log S)$ depth and $O(S)$ width. The trade-off essentially lies in how a min function is implemented and can be resolved if we allow a min-pooling layer, which we will discuss after the proof.

1. *$O(1)$-depth construction:* The idea is that we can first use $O(1)$ layers to construct "features" that contain all the information needed to determine the state, then a 3-layer MLP with $2^{O(S)}$ width can compute the state as a function of these features by Lemma 2. The features we need are the following (the labels underneath are to be consistent with Figure 17, *left*):

$$\underbrace{\{\mathbb{1}[x_{\text{attn}}^{(2)}[s] > x_{\text{attn}}^{(2)}[s+S]]\}_{s=0}^{S}}_{>}, \tag{C.3}$$

$$\underbrace{\{\mathbb{1}[x_{\text{attn}}^{(2)}[s-1] > x_{\text{attn}}^{(2)}[s+S]]\}_{s=0}^{S}}_{>_L}, \quad \underbrace{\{\mathbb{1}[x_{\text{attn}}^{(2)}[s] > x_{\text{attn}}^{(2)}[s+S+1]]\}_{s=0}^{S}}_{>_R}, \tag{C.4}$$

$$\underbrace{\{\mathbb{1}[x_{\text{attn}}^{(2)}[s] = \log(2T - t)]\}_{s=0}^{2S}}_{=}\}. \tag{C.5}$$

Here the feature in C.3 compares the end points of the $S + 1$ windows, the two features in C.4 compare the window with its adjacent windows on each side, and the last feature in C.5 will be used to eliminate the irrelevant window. Features in C.3 and C.4 can each be computed as a threshold function (at 0) on the difference between the two elements to be compared, which can be implemented using 2 layers by Lemma 3.

(a) First, we show that to compute *Step 1* we only need to compare between adjacent windows whose $S + 1$ heads are all matched, which can be computed using the features above. Consider any window starting at $s \in [0, 1, \cdots, S]$. On either side, if this window is closer to $t$ than its adjacent window on this side, then it is closer to the end of the boundary than all the windows on this same side, which would imply that we can ignore these non-adjacent windows. To prove this, let's consider the left side (the right side is analogous) and we want to show: For any $s \in [S]$, if $\max\{x_{\text{attn}}^{(2)}[s], x_{\text{attn}}^{(2)}[s + S]\} < \max\{x_{\text{attn}}^{(2)}[s - 1], x_{\text{attn}}^{(2)}[s + S - 1]\}$, then $\max\{x_{\text{attn}}^{(2)}[s], x_{\text{attn}}^{(2)}[s + S]\} < \max\{x_{\text{attn}}^{(2)}[s - i], x_{\text{attn}}^{(2)}[s + S - i]\}$ for $i > 1$: the *if* condition gives us $x_{\text{attn}}^{(2)}[s - 1] > x_{\text{attn}}^{(2)}[s + S]$, and we know from Lemma 14 that

$$x_{\text{attn}}^{(2)}[s - i] > x_{\text{attn}}^{(2)}[s - 1] > x_{\text{attn}}^{(2)}[s],$$
$$x_{\text{attn}}^{(2)}[s + S] > x_{\text{attn}}^{(2)}[s + S - 1] > x_{\text{attn}}^{(2)}[s + S - i].$$

Combining these inequalities together concludes the proof.

(b) Given the optimal window, we can use feature C.3 for the relevant window to identify the boundary, since the closer-to-$t$ index gives us the last boundary state (see Algorithm 1 for why this suffices).

(c) Now that we have identified the boundary state (suppose it is at position $i$), the final state can be computed as the last boundary state (0 or $S$) plus the difference $x_{\text{attn}}^{(1)}[t] - x_{\text{attn}}^{(1)}[i]$. The difference is built in to the ordering of the heads, hence we have all the information to compute the final state and we are done.

Therefore, we can compute this function using $4S + 3$ features each taking value in $\{0, 1\}$ and the output having $S + 1$ values. These features themselves can be constructed using Lemma 3 with $\Delta = 1/4T$ since the indices are separated by at least this gap. For the indicator index, we can compose two such constructions similar to Lemma 1. This gives us the first layer of MLP with width $O(S)$ and norms $O(T)$. After this, the rest of the function can be constructed using a 3-layer ReLU network with width $2^{O(S)}$ and norms bounded by $O(S)$ using Lemma 2.

2. $O(\log S)$-*depth construction:* An alternative solution to the above is to pay $O(\log(S))$ depth, but reduce the width to be $O(S)$. We will borrow features in equation C.3-C.5, but construct the MLP explicitly rather than calling Lemma 2 as a black box: the width and depth trade-off essentially correspond to two ways of implementing the min of $S$ numbers. We describe the corresponding MLP by components (Fig 17, *right*):

(a) Ignore the unmatched heads: as a preprocessing step for the cleanness of the proof, we use a 1-layer MLP to map $x_{\text{attn}}^{(2)}[s]$ for heads where $x_{\text{attn}}^{(2)}[s] = \log(2T - t)$ (i.e. $j_{\max} = t$) to $\log(2T)$ such that these unmatched heads can be ignored in the following steps. This can be done by multiplying $\log(2T)$ to the threshold function given by Lemma 3 (with $\Delta = \frac{1}{4T}$), where the network has 1 hidden layer with width $2S + 1$ and $\infty$-norm $4T$. Note that this map also changes $x_{\text{attn}}^{(2)}[S]$ (i.e. the center head that corresponds to position $t$) but this will not affect the correctness of the proof.

(b) Compute the function $f_1$ in Figure 17 (*right*), which computes $f_1(a, b) := (\max\{a, b\}, \mathbb{1}[a > b])$ for $a = x_{\text{attn}}^{(2)}[s]$, $b = x_{\text{attn}}^{(2)}[s + S]$, $\forall s \in \{0, 1, \cdots, S\}$. The first coordinate $\max\{a, b\}$ can be implemented using 1 hidden layer with width 1, and $\mathbb{1}[a > b]$ is the same as feature C.3 and can be implemented with 1 hidden layer by Lemma 3. There are $S + 1$ choices of $s$, hence the overall width is $O(S)$. For notational convenience, let's denote $f_1(s) := f_1(x_{\text{attn}}^{(2)}[s], x_{\text{attn}}^{(2)}[s + S])$ (with a slight abuse of notation).

(c) Find the min value of $f_1(s)[1]$, denoted as $f_{1,\min} := \min_s f_1(s)[1]$: This can be achieved using 1 min-pooling layer. If we allow ReLU only, then this can be implemented with pairwise comparison using a network with $\lceil \log S + 1 \rceil$ depth, $3S$ width and and constant weight norm. [29]

(d) Compute the function $f_2^{(s)}$ in Figure 17 (*right*): $f_2^{(s)}$ takes two inputs: 1) the second output of $f_1$, which we denote as $B_s := f_1(s)[2]$, and 2) $M_s := \mathbb{1}[f_{1,\min} \le f_1(s)[1]]$, which

---

[29]The log depth is conjectured to be unimprovable; see discussion after the proof.

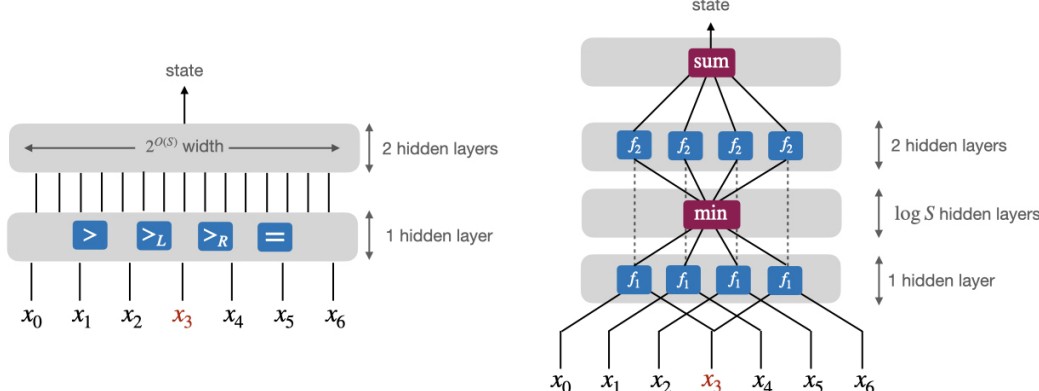

Figure 17: Illustration of the two constructions for second-layer MLP in Theorem C.4, with $S = 3$. For ease of readability, we replace $x_{\text{attn}}^{(2)}$ with $x$. *Left:* $O(1)$-depth solution where the first block compares the comparison and equality features (see equation C.3- C.5), and the second block computes the state from these features. *Right:* $O(\log S)$-depth solution where first block implements $f_1(a, b) = (\max\{a, b\}, \mathbb{1}[a > b])$, second block does a min-pooling operation, the third block implements $f_2$ which takes $f_1$ via a residual connection and output of $\min$-pool to compute the final state.

    indicates whether the $s^{th}$ window is the closest-to-$t$ window or not and can be computed using a 1-hidden-layer network with width 2. As in the previous construction, the difference $f_{1,\min} - f_1(s)[1]$ is built-in in the ordering of the head and hence does not need to be passed in explicitly. Then by Lemma 2, a 2-hidden-layer network of width $O(1)$ can take $B_s, M_s$ as input and compute $M_s \cdot \big[ B(S - s) + (1 - B)(2S + 1 - s) \big]$. The overall width is $O(S)$ for $S + 1$ choices of such $f_2^{(s)}$.

(e) Finally, the state is computed as $\sum_s f_2^{(s)}(B_s, M_s)$, which can be implemented with 1 layer of width 1.

□

**Improving the construction to remove $T$ width and $\log(S)$ depth.** Using standard architectural tools, such as max-pooling, we can improve our construction to get $O(1)$-depth and $O(S)$-width for the MLP.

- *Avoiding width $T$ in the MLP 1 using periodic activations.* As in the modular addition (Lemma 6) construction, we can use $\sin$ activations in the MLP to directly compute the circular embeddings that are used as input to the second attention layer. This would require only two hidden nodes in the MLP. Note that we do not need precision greater than $O(\log T)$ for these activations since we are embedding values only as close as $1/\mathsf{poly}(T)$.

- *Avoiding $\log(S)$ depth in the MLP 2 using max-pooling.* The $O(\log S)$-depth in MLP 2 is incurred by calculating the min of $S$ numbers and is conjectured to be necessary for ReLU networks (Goel et al., 2017; Mukherjee & Basu, 2017; Hertrich et al., 2021). However, the depth can be reduced to 1 if we allow max-pooling layers, which are commonly used in both theory and practice (Zhang et al., 2021b; He et al., 2016; Vaswani et al., 2021).

*Remark:* Yao et al. (2021) use layer-norm to compute $\cos$ and $\sin$ embedding with non-uniform angles. This could potentially alleviate the width $T$ concern; we leave this exploration to future work.

**Extending beyond 1 dimension.** Since a 2-dimensional gridworld is just the direct product of 1-dimensional gridworlds (by the construction in Lemma 8), we can implement both dimensions in

parallel by concatenating the network for each dimension. This can be done by doubling the dimensions, parallel attention heads, and parallel hidden units in the MLP. The attention head parameters for each dimension can be chosen to only focus on the relevant dimension and similarly the MLP can zero out dependence on the other dimension. We can extend this to higher dimension with a multiplicative increase in the size of the parameters.

### C.5 Proof of Theorem 4: Depth lower bound for non-solvable semiautomata

**Theorem 4** (Transformer Barrington). *Let $\mathcal{A}$ be a non-solvable semiautomaton. Then, for sufficiently large $T$, no fixed-precision Transformer with depth independent of $T$ and width polynomial in $T$ can simulate $\mathcal{A}$ at length $T$, unless $\mathsf{TC}^0 = \mathsf{NC}^1$.*

*Proof.* This follows straightforwardly from the fact that simulating $\mathcal{A}$ at length $T$ is $\mathsf{NC}^1$-complete under $\mathsf{NC}^0$ reductions: given any $O(\log T)$-depth bounded-fan-in $\mathsf{AND}/\mathsf{OR}/\mathsf{NOT}$ circuit $\mathcal{C}$, and a depth-$D$ circuit $\mathcal{C}'$ which simulates a semiautomaton whose transformation monoid contains a non-solvable subgroup, there is a procedure which generates a depth-$O(D)$ circuit to simulate $\mathcal{C}$; see (Barrington & Thérien, 1988). This in turn comes from the construction used in Barrington's theorem (Barrington, 1986), which characterizes $\mathsf{NC}^1$ as exactly the set of languages recognizable by *bounded-width branching programs*. For a closely related reference which follows almost exactly the same argument, see (Mereghetti & Palano, 2000).

Thus, it suffices to show that a constant-depth Transformer is in $\mathsf{TC}^0$. The details of manipulating floating-point numbers with discrete circuits are peripheral to the main results in this paper, so we provide a brief proof sketch. A similar argument is used by Merrill et al. (2021) to establish that *"saturated"* Transformers (a multi-index analogue of hard-attention Transformers), with $O(\log T)$ bit precision, can be represented with a $\mathsf{TC}^0$ circuit. We outline a proof (which applies to the formal setting considered by Merrill et al. (2021)) for the notion of Transformers defined in this paper.

With $O(\log T)$ bits of precision, all $n$-way (including unary) arithmetic operations mapping $\mathbb{R}^n \to \mathbb{R}$ can be represented with a constant-depth, $\text{poly}(T)$-width $\mathsf{AC}^0$ circuit, as long as $n$ does not depend on $T$. Although improvements are certainly possible, it suffices to consider the circuit which memorizes the $i$-th bit of the output, which has width $2^{n \log T} \leq O(\text{poly}(T))$. Thus, the position-wise non-interacting matrix operations (multiplication by $X \mapsto W_Q X$, etc., the feedforward MLP layers, and the encoding and decoding layers) can be simulated with $\text{poly}(T)$ width.

The only subtlety arises when there is a $T$-way summation over $O(\log T)$-bit numbers, which occur in the softmax and attention mixture layers. For this operation, we can use the construction from (Reif & Tate, 1992), which can even add $T \, \text{poly}(T)$-bit numbers in $\mathsf{TC}^0$. $\qquad\square$

