# OpenReview forum: "Transformers Learn Shortcuts to Automata"
_ICLR.cc/2023/Conference — ICLR 2023 notable top 5%_

### Official Review · Reviewer_Czkt · 2022-10-22

**Confidence:** 4
**Correctness:** 4
**Technical Novelty And Significance:** 4
**Empirical Novelty And Significance:** 3
**Recommendation:** 8

**Clarity, Quality, Novelty And Reproducibility:**

A further sense in which this paper is unusual is in the tidy, organized, and well polished presentation, which is at a level not typically seen in ICLR submissions. The paper is highly technical, and uses a considerable number of concepts that are likely new to the majority of the machine learning community, myself included, and yet the work remains relatively approachable to the extend that I am able to take away many useful points.

In terms of reproducibility, the authors commit to open sourcing their code. This is adequate for me.


**Strength And Weaknesses:**

This is a highly unusual ICLR paper. This is not to say that ICLR is an inappropriate venue for this work, since this is clearly not the case: the subject matter is of direct and high importance to the machine learning community. This work is unusual because of the breadth and depth of ideas and connections to different parts of the theoretical computer science literature, as well as very topical considerations to the ICLR community such as out-of-distribution generalization, spurious correlations, and neural algorithmic reasoning. For instance, the results presented in this paper are formulated in terms of the Transformer’s ability to simulate automata—-a now classical concept in AI (albeit being essentially conceptually the same as recurrent models). The analysis tools are also unusual, with many algebraic arguments used, and results from classical complexity theory are drawn upon.

A few brief strengths:

- A complete set of theoretical results, including general results for any arbitrary automata, sharper bounds for certain (solvable) automata, even sharper bounds for a particular class of automata (grid words), and lower bound results.
- Experimental results validating a number of these findings on a number of algorithmic reasoning tasks, taking pains to use the experiments as a chance to bridge some of the inevitable gaps between the theory and practice (such as checking whether gradient-based training actually finds the shortcuts the theory identifies).
- Taking care to at least discuss a number of questions a reader might have whilst reading this work, such as the connections to universal approximation,

No weaknesses come to mind, and I do not wish to work to artificially come up with problems. That’s not to say that the work is perfect, of course it isn’t, but I have no major concerns about the validity or significance of the results.

**Summary Of The Paper:**

This paper studies the relation between Transformer depth and its ability to solve algorithmic reasoning tasks. The authors formulate reasoning processes in terms of automata, and particularly focus on the question of why and if shallow Transformers are able to simulate automata with much longer horizons. A number of results are presented of the following flavor: Transformers of depth $o(T)$ can be used to simulate automata of length T so long as the width (of e.g., the MLP blocks in the transformer) is big enough, and enough attention heads are used. The width and number of heads needed is quantified in terms of the size of the state space. The shortcuts referred to in the title of the paper refers to the fact that these results show that Transformers can learn to simulate long processes without directly learning each step of the process explicitly.

**Summary Of The Review:**

In all, I am strongly in favor of this paper’s acceptance. I cannot pretend to have fully understood the depths of the ideas in this work - for instance I have not yet had the chance to study in detail how the attention mechanism is able to compete multiple automata steps in parallel (this is probably a good moment to make the disclaimer that I haven’t properly verified any of the sizably appendix to this paper, so all results are taken on face value). However, I have read and understand more than enough to conclude that a this paper makes a contribution that is easily sufficient to merit acceptance at ICLR. I will be interested to see if any points are raised by other reviewers that I wasn’t aware of, but I doubt anything will change my overall view. Congratulations to the authors on an immaculate piece of scholarship, I hope you continue to develop this line of work further.

---

> ### Author Response · Authors · 2022-11-10
> **Response to Reviewer Czkt**
>
> Thank you for the encouraging feedback. We were also surprised by how naturally these unusual mathematical tools showed up while pursuing these questions about Transformers.
>
> Regarding intuitions as to how attention can compute multiple steps in parallel: perhaps it would be helpful to point out informally that the crux of our paper is captured by the construction for “reversible car” example (see beginning of the proof of Lemma 9). Commutative operations can clearly be “summarized” in 1 layer via counting (implemented using uniform attention). This semidirect product construction shows how circuits for commutative operations can be glued together hierarchically to simulate certain **non-commutative** operations. And Krohn-Rhodes theory says that to handle non-invertible transformations, we only need one additional capability: sparse selection (which is easily implemented by attention).

---

### Official Review · Reviewer_cJyA · 2022-10-24

**Confidence:** 3
**Correctness:** 4
**Technical Novelty And Significance:** 4
**Empirical Novelty And Significance:** 4
**Recommendation:** 10

**Clarity, Quality, Novelty And Reproducibility:**

Very clearly written and gets strong results.  The analysis makes use of deep prior results (Krohn-Rhodes theory) and thus can go far.  Possibly a breakthrough in theoretical analysis of transformers.

A good computational model, much used for language processing (formal and natural), is the finite state machine (FSM or equivalently semiautomaton or deterministic Markov process).    Examples of computations naturally done by FSM's include group actions (say turning a sequence of Rubik's cube moves into the resulting sequence of face configurations) and processing regular languages (for example grouping characters into words).   Many more computations can be done with limitations.  Examples are checking properly nested parentheses with a maximum allowed nesting depth, or parsing sentences again with a limit on the parse tree depth.  There is a growing literature studying how and when transformers can learn to perform such tasks -- proposing circuits which can perform the tasks, empirically measuring the resources needed to learn the tasks, and reverse engineering the trained networks to understand their implementations.  Most of this work is referenced under "related work" and in section A.5.  A good part of it hypothesizes that the learned implementation is similar to a human designed circuit and looks for this structure in the trained network.  But this need not be the case and it is very interesting to have analyses based on other hypotheses or not presupposing a type of implementation.

The paper starts with the following observation.  An FSM can be implemented by a recursive neural network (RNN), which has state, in an obvious way.  However it is not as obvious how to implement a general FSM with a transformer model which does not have internal state and must compute each output as a function of the previous inputs.  A simple "unrolling" of the RNN leads to a circuit with depth T where T is the maximum length of an input sequence.  But transformers used in practice typically have depth 10-20, far smaller than the sequences being processed.  This is not a contradiction because attention can relate distant parts of the input, but there was no framework for understanding how this ability is actually being used.

The proposal of the paper is that one can start to understand this using Krohn-Rhodes theory, a branch of the mathematical theory of automata.  This reviewer had never heard of this theory and was not at first convinced that it was relevant, but after reading the paper finds that it makes a convincing case.  Intuitively, Krohn-Rhodes theory decomposes semiautomata into two basic units, simple group actions which are reversible, and memory units which are maximally irreversible.  The paper shows that these units and their combinations can be naturally implemented by the transformer architecture, and indeed one can see simpler (and presumably related) versions of these operations in the previous works on transformer circuits.  So this does seem to unify many previous results in a consistent framework.

In summary, this is a new and very original approach to a problem of great current interest, and which in my opinion could be developed much further.  As such it deserves to be highlighted at ICML.

**Strength And Weaknesses:**

Strengths: very interesting question and approach, solidly grounded in theoretical computer science, clearly written, strong results.

Weaknesses: the scope and complexity of the ideas make them hard to fully convey in a paper satisfying the length constraints, even with the supplementary material.  In particular a lengthier comparison with the related works (and those in section A.5) would be of great interest.  We hope the authors will write a longer treatment elsewhere.

**Summary Of The Paper:**

This is a theoretical work about how transformer models can perform symbolic tasks.  The class of tasks considered is semiautomata (deterministic finite state machines).  The focus is on how a shallow network can compute sequences of states of length L much greater than the network depth D.  It is shown that this is indeed possible, with D \sim log L at worst, and for a subclass of automata, constant D.
Experiments are done to validate these predictions and to give evidence that these networks are learnable from data.


**Summary Of The Review:**

A novel approach to understanding transformers which leads to significant results.

---

> ### Author Response · Authors · 2022-11-10
> **Response to Reviewer cJyA**
>
> We are grateful for the enthusiastic feedback.
>
> Space constraints were certainly hard to navigate for this manuscript. We are curious for which lines of related work the reviewer would like to see an expanded discussion. We would be happy to include more detailed technical comparisons.
>
> We agree that there is much more work to be done, in terms of exploring the implications for reverse engineering the implementations learned by deep learning. One comment is that our work can be viewed as quantifying an obstruction towards interpretability: even in this very general class of dynamics defined by state transitions, there are always solutions in which the activations do not correspond to states. In general, a trained model could contain a mixture of the different solutions (state emulation; memorization; function composition; holonomy decomposition) discussed in this paper, so interpreting such a model would necessitate disentangling this mixture.

---

### Official Review · Reviewer_zU3W · 2022-10-25

**Confidence:** 3
**Correctness:** 3
**Technical Novelty And Significance:** 3
**Empirical Novelty And Significance:** 2
**Recommendation:** 6

**Clarity, Quality, Novelty And Reproducibility:**

The paper is rather dense (9 pages for 60 pages with the Appendix) but it is clearly written. It should have been useful to position the study w.r.t. similar studies for LSTMs. For the theoretical part, as a non expert and because of the delay for reviewing,  I can not assert that the proofs are correct but the theorems seem sound to me. I would have liked also a discussion on the significance of simulation at length T while automata are designed as finite objects to deal with sequences of unbounded length.

For the second part of the paper, I would have liked to read more discussion with other contributions on learning automata such as the one by Bhattamishra et al. I am also always surprised that computational learning theory seems to be completely forgotten in papers along this trend of research.

It is clearly stated that the results are given by the best model but I would have liked to know how many experiments lead to good performance and how many to low performance. My main concern is about the evaluation protocol. It seems that the training set and the testing set are drawn with uniform distribution. I would have liked more on this. Indeed, when learning languages with very few positive examples such a protocol is meaningless. Also, for Dyck languages, what would be the meaning to evaluate on randomly drawn sequences of parentheses.

**Strength And Weaknesses:**

I am not expert in Computational Theory and Complexity Theory.

Pros

* The theoretical contribution is dense but clear
* The theorems seem sound
* Experimental results complement theoretical results

Cons

* Could be published at a conference but should be submitted at a review
* Experimental protocols should be made more precise
* I am not so convinced by the lessons drawn from the experiments

**Summary Of The Paper:**

The paper is a contribution to the understanding the capabilities of transformers to model languages. The first part of the paper provides results on the capabilities of (shallow) transformers to simulate computations of semiautomata at length T. The second part of the paper is an experimental study on whether learning allow to find such shallow transformers

**Summary Of The Review:**

I am clearly not expert enough to give a firm assessment of the paper. I tend towards acceptance because the theoretical results seem meaningful and sound to me. I would like to have more details to assert the significance of the experimental results.

---

> ### Author Response · Authors · 2022-11-10
> **Response to Reviewer zU3W**
>
> Thank you for the review. We appreciate your careful consideration of experimental details.
>
> **Input distribution and state coverage**. We begin by clarifying a factual misunderstanding, which seems to be a large part of the reviewer’s overall evaluation. We agree that the manuscript could have been clearer about this, and have revised accordingly.
>
> - We would first like to clarify that by “uniform distribution” in the case of Dyck languages, we mean that continuations are drawn uniformly from the set of valid tokens, not the full alphabet. In other words, we only consider sequences that are valid to the automata. For Dyck, when no closing bracket is a valid continuation, then the distribution is uniform over the open brackets. This is in accordance with [Yao et al. ‘21] (which falls under the partially-observable case). We apologize for the confusion and have updated Appendix B.1.1 to clarify this.
> - In all of the cases considered in our paper, except $(abab)^*$, this “uniform random walk” protocol can be verified to provide ample state coverage. Note that this does not in general imply coverage of shortcut features (e.g. outlier prefix sums in the parity task); indeed, one of our contributions is to uncover this mode of OOD brittleness.
>
> **Detailed comparison with prior literature**. As requested by this reviewer (as well as Reviewer cJyA), an expanded discussion:
> - While we obtain positive results on settings with the same underlying semiautomata as the negative results from Bhattamishra et al. (= [1]), there is no direct contradiction due to the following differences:
>   - Larger model: [1] considers Transformers with up to 4 layers, 4 heads and dimension up to 32, whereas we consider larger models with up to 16 layers, 8 heads, and dimension 512. We now clarify this in Appendix B.2.4.
>   - Full observations: A major difference in the experimental setup is that we supervise on the states (i.e. semiautomata, taking $|Q|$ values) whereas [1] supervises on the acceptance status (i.e. automata, taking binary values). The latter is considered partial supervision, which we discuss in Sec 5.1 and Appendix B.2.1. In B.2.1, we use one of the tasks in [1], namely $(abab)^*$.
>   - Evaluation protocol: we measure accuracy at the token level, whereas [1] measures accuracy at the sequence level. We have updated Appendix B.1.1 to clarify this difference.
> - In Appendix A.5, we have provided what we believe to be quite a comprehensive discussion of related work. The overall relation to studies involving RNNs/LSTMs is that those are the natural continuous relaxations of automata (see the discussion in Section 2.2). Our paper studies alternative relaxations, which are not at all obvious from the definitions of the state transitions, and can benefit from parallel computation.
>
> **Empirical novelty**. The reviewer also seems to be concerned with the novelty of the empirical results. Some comparisons:
> - We argue that unifying many of the results in past works is a contribution in itself.
> - We include previously unconsidered group/semigroup operations under this unified framework.
> - Similar to [1], we evaluate length generalization and have a similar conclusion that Transformers (by default) fail to generalize. However, unlike [1], we investigate a positive solution in this setting (i.e. scratchpad + recency bias). Our considerations for positional encodings are complementary to the results in [1].
> - We further evaluate incomplete/indirect supervision and OOD generalization under the same length.
> - We provide results on mechanistic interpretability, such as the boundary detector for gridworld and the relation to counts for parity.
>
> **Training instability**. Indeed, it was our goal to characterize some of the optimization challenges and obstructions towards learning “shortcuts” in neural algorithmic reasoning pipelines.
> - Closest to the reviewer’s suggestion, to quantify the random-seed-dependent variance, we supplemented the best-run accuracies with the **median** performance in Table 2 (Appendix B). We have added a pointer in the main text.
> - We find that the median is much lower than the max, which shows training instability (see Appendix B for more details). Addressing such instability issues is outside the scope of this work (mentioned in Section 4) but is an interesting future direction.
>
> **Significance of considering simulation at length $T$**. We didn’t completely understand this question and will answer it to the best of our ability. Our work focuses on automata precisely because of their ability to concisely represent unbounded-length sequences. We are interested in understanding how (and how efficiently) **non-recurrent** models can represent recurrent concepts. For these questions to be well posed, the size of the non-recurrent model must grow with $T$; our finding is that the **depth** only needs to grow modestly with $T$, and sometimes not at all.

---

### Author Response · Authors · 2022-11-10
**Revision to manuscript (post-reviews)**

We thank all reviewers for their thoughtful comments and encouraging feedback. We have uploaded a revised copy of the manuscript, with the following changes:
- Numerous clarity improvements, especially in the appendix.
- A few more related works that came to our attention after the submission deadline.
- More discussion on [Bhattamishra et al. ‘20] (and a direct experimental comparison, matching their sequence-level accuracy evaluation protocol), to address the concerns of Reviewer zU3W.
- We have added some more refined experiments to the empirical section.

---

### Decision · Program_Chairs · 2023-01-20

**Decision:**

Accept: notable-top-5%

**Justification For Why Not Higher Score:**

I am a bit uncertain on this. On the one hand, the paper is quite solid, on the other its scope can be of interest to a restricted number of people and the appendix is quite daunting. As one reviewer suggested, this work can be better suited in a journal.
After checking with the SAC, we agreed that the interest and potential impact of the work can be worthy a oral.


**Justification For Why Not Lower Score:**

A poster might be good, but authors deserve to communicate the results to a broader audience.

**Metareview: Summary, Strengths And Weaknesses:**

In this work authors try to address the problem of understanding both theoretically and empirically what reasoning tasks transformers can learn. They investigate this by simulating semiautomata with transformers of a certain depth.

All reviewers agree that this is a high-quality paper, that tackles an important question and frames it with rigor and soundness. Reviewers also praised the techniques used in the proofs and highlighting how this can inspire following theoretical investigations on transformers. Reviewers also highlighted that the appendix is quite daunting and suggested that this work (or its extension) can definitely find home in a journal.
I believe the contribution is still worth a conference publication.

During the reviewing phase some minor concerns over presentation and experimental details were raised. Authors were quick and precise into addressing them and revising the paper.

The paper is accepted.

**Note From Pc:**

if the above contains the word "oral" or "spotlight" please see: "oral" presentation means -> notable-top-5% and "spotlight" means -> notable-top-25%. As stated in our emails, we are disassociating presentation type from AC recommendations

**Summary Of Ac-Reviewer Meeting:**

N/A